# Sea-level rise in Venice: historic and future trends (Review article)

Davide Zanchettin[1], Sara Bruni[2*], Fabio Raicich[3], Piero Lionello[4], Fanny Adloff[5], Alexey Androsov[6,7], Fabrizio Antonioli[8], Vincenzo Artale[9], Eugenio Carminati[10], Christian Ferrarin[11], Vera Fofonova[6], Robert J. Nicholls[12], Sara Rubinetti[1], Angelo Rubino[1], Gianmaria Sannino[13], Giorgio Spada[2], Rémi Thiéblemont[14], Michael Tsimplis[15], Georg Umgiesser[11,16], Stefano Vignudelli[17], Guy Wöppelmann[18], Susanna Zerbini[2]

[1]University Ca' Foscari of Venice, Dept. of Environmental Sciences, Informatics and Statistics, Via Torino 155, 30172 Mestre, Italy

[2]University of Bologna, Department of Physics and Astronomy, Viale Berti Pichat 8, 40127, Bologna, Italy

[3]CNR, Institute of Marine Sciences, AREA Science Park Q2 bldg., SS14 km 163.5, Basovizza, 34149 Trieste, Italy

[4]Unversità del Salento, Dept. of Biological and Environmental Sciences and Technologies, Centro Ecotekne Pal. M - S.P. 6, Lecce Monteroni, Italy

[5]National Centre for Atmospheric Science, University of Reading, Reading, UK

[6]Alfred Wegener Institute Helmholtz Centre for Polar and Marine Research, Postfach 12-01-61, 27515, Bremerhaven, Germany

[7]Shirshov Institute of Oceanology, Moscow, 117997, Russia

[8]Istituto Nazionale di Geofisica e Vulcanologia, via di Vigna Murata 605, 00143 Rome, Italy

[9]ENEA C.R. Frascati, SSPT-MET, Via Enrico Fermi 45, 00044 Frascati, Italy

[10]University of Rome La Sapienza, Dept. of Earth Sciences, Piazzale Aldo Moro 5, 00185 Roma, Italy

[11]CNR - National Research Council of Italy, ISMAR - Marine Sciences Institute, Castello 2737/F, 30122 Venezia, Italy

[12] Tyndall Centre for Climate Change Research, University of East Anglia. Norwich NR4 7TJ, United Kingdom

[13] ENEA Casaccia, Climate and Impact Modeling Lab, SSPT-MET-CLIM, Via Anguillarese 301, 00123 Roma, Italy

[14]Bureau de Recherches Géologiques et Minières "BRGM", French Geological Survey, 3 Avenue, Claude Guillemin, CEDEX, 45060 Orléans, France

[15]City University of Hong Kong, School of Law, Tat Chee Avenue, Kowloon, Hong Kong

[16]Marine Research Institute, Klaipeda University, Klaipeda, Lithuania

[17]CNR, Institute of Biophysics, AREA Ricerca, Via Moruzzi 1, 56127 Pisa, Italy

[18]LIENSs, CNRS - La Rochelle University, 2 rue Olympe de Gouges, 17000 La Rochelle, France

[*]now at: PosiTim UG, Seeheim-Jugenheim, Germany.

*Correspondence to*: Davide Zanchettin (davidoff@unive.it)

**Abstract.** The City of Venice and the surrounding lagoon ecosystem are highly vulnerable to variations in relative sea level. In the past ~150 years, this was characterized by an average rate of relative sea-level rise of about 2.5 mm/year resulting from the combined contributions of vertical land movement and sea-level rise. This literature review reassesses and synthesizes the progress achieved in quantification, understanding and prediction of the individual contributions to local relative sea level, with focus on the most recent studies. Subsidence contributed to about half of the historical relative sea-level rise in Venice. The current best estimate of the average rate of sea-level rise during the observational period from 1872 to 2019 based on tide-gauge data after removal of subsidence effects is 1.23±0.13 mm/year. A higher - but more uncertain - rate of sea-level rise is observed for more recent years. Between 1993 and 2019, an average change of about +2.76 ±1.75 mm/year is estimated from tide-gauge data after removal of subsidence. Unfortunately, satellite altimetry does not provide reliable sea-level data within the Venice Lagoon. Local sea-level changes in Venice closely depend on sea-level variations in the Adriatic Sea, which in turn are linked to sea-level variations in the Mediterranean Sea. Water mass exchange through the Strait of Gibraltar and its drivers currently constitute a source of substantial uncertainty for estimating future deviations of the Mediterranean mean sea-level trend from the global-mean value. Regional atmospheric and oceanic processes will likely contribute significant interannual and interdecadal future variability of Venetian sea level with a magnitude comparable to that observed in the past. On the basis of regional projections of sea-level rise and an understanding of the local and regional processes affecting relative sea-level trends in Venice, the likely range of atmospherically corrected relative sea-level rise in Venice by 2100 ranges between 32 and 62 centimetres for the RCP2.6 scenario and between 58 and 110 centimetres for the RCP8.5 scenario, respectively. A plausible but unlikely high-end scenario linked to strong ice-sheet melting yields about 180 centimetres of relative sea-level rise in Venice by 2100. Projections of human-induced vertical land motions are currently not available, but historical evidence demonstrates that they have the potential to produce a significant contribution to the relative sea-level rise in Venice, exacerbating the hazard posed by climatically induced sea-level changes.

# 1 Introduction

This paper reviews the current knowledge about mean Relative Sea Level (RSL) changes in the Venice Lagoon on time scales from interannual to centennial and the associated contribution from oceanic, land and atmospheric processes. The assessment includes a paleo perspective, considering the Quaternary period. It encompasses an overview of available observed estimates of historical RSL changes in Venice (Sect. 2) and quantification of the individual contributions by the major underlying processes, including vertical land motions (Sect. 3) and climatic changes (Sect. 4). Estimates are supported by a review of downscaling mechanisms of global and large-scale oceanic and atmospheric signals to the Venice Lagoon (Sect. 5), with special focus on processes in the Atlantic and Euro-Mediterranean regions. Estimates of projected long-term future RSL changes based on state-of-the-art models of vertical land motions and of sea-level rise under different scenarios of

anthropogenic greenhouse gas emission are discussed, with emphasis on the associated major sources of uncertainty (Sect. 6).

The review primarily focuses on papers published in the past decade and also aims to define the overarching open research questions and possible approaches for progress (Sect. 7).

Given the multidisciplinarity of this review, it is useful to specify the meaning of terms and concepts associated with sea-level changes that are recurrent in this paper and often used inconsistently by different scientific communities (see also Gregory et al., 2019, for a broader discussion). Unless otherwise specified in the text, the following definitions apply:

● Relative Sea Level (RSL) change: change in local sea level relative to the local solid surface (Gregory et al., 2019). The acronym RSL is therefore used for tide-gauge data;

● Geocentric Sea Level (GSL) change: change in local sea level with respect to a geocentric reference, namely a Terrestrial Reference Frame or, equivalently, a reference ellipsoid (Gregory et al., 2019). The acronym GSL is therefore used for satellite altimetry sea-level data;

● Subsidence: land surface sinking (UNESCO, 2020; see also: Gregory et al., 2019);

● VLM-corrected RSL: local sea level derived from tide-gauge RSL data corrected for vertical land movements;

● Global-mean sea level (GMSL): spatially averaged RSL over the World Ocean.

A list of acronyms recurrently used in the paper is provided in Table 1.

The Reader is referred to Lionello et al. (2020a,b) and Umgiesser et al. (2020) in this special issue for details about the

geographical and historical setting of the Venice Lagoon, the linkage between RSL changes and the phenomenology of surges and extreme water levels affecting the lagoon, about their prediction and about broader implications for the ecosystems and the historical city.

## 2 Monitoring sea-level changes

The monitoring of sea-level changes in Venice relies on both in-situ data acquired by tide gauges (sect 2.1) and remote sensing

observations provided by satellite radar altimetry (sect. 2.2). Tide gauges record sea-level heights with reference to a permanent benchmark on land. Therefore, they provide measurements of RSL embedding the effects of vertical land motion (Sect. 3). Tide gauge data sets consist of local, long-term measurements acquired at high frequency and accuracy (Zerbini et al., 2017). Satellite radar altimetry, on the contrary, measures geocentric sea-level changes (Fu and Cazenave, 2001; Stammer and Cazenave, 2017). These are independent of variations of the local land levels (Gregory et al., 2019), hence unaffected by a

potentially key component of RSL (Wöppelmann and Marcos, 2016; Gregory et al., 2019). These measurements have a lower sampling rate (several days) and a lower accuracy than those provided by tide gauges, but they are representative of wider oceanic areas and have the potential to characterize the evolution of sea-level variability with an almost global coverage.

## 2.1 Tide Gauges

Tide gauges have been providing sea-level data in Venice for about 150 years. Historically, the establishment of tide gauges
was primarily dictated by navigational needs and tidal measurements, with an operational accuracy of a few centimeters. The
first self-recording tide gauge in Venice was installed at Palazzo Loredan, Campo Santo Stefano, in Rio San Vidal at a distance
of about 100 m from the Grand Canal (Fig. 1). Systematic measurements began on 27th November 1871. The observations
were performed under the responsibility of the Civil Engineering Office (Ufficio del Genio Civile) until 27 July 1896, when
the management was taken over by the Italian Military Geographic Institute (Istituto Geografico Militare), which was also in
charge of land levelling. Two additional tide gauges became operational in 1888 and 1906. The first one, owned and managed
by the Royal Italian Navy, was installed in the Venice Arsenal; the second one was installed in the Grand Canal, near Punta
della Salute. The tide gauges at Santo Stefano and in the Arsenal were decommissioned in 1911 and 1917, respectively. In
1923, the gauge on the Grand Canal was moved to the Giudecca Canal side of Punta della Salute. This gauge is still active
under the management of the "Istituto Superiore per la Protezione e la Ricerca Ambientale" (ISPRA, Venice branch,
www.venezia.isprambiente.it). Since 2002, a gauge on the Grand Canal side is again operational on the site of the previous
installation, thanks to the Venice municipality ("Centro Previsione e Segnalazione Maree",
www.comune.venezia.it/content/centro-previsione-e-segnalazione.maree).

Further details on the tide gauges installed in the Venice Lagoon up to the early 20th century are reported by Magrini et al.
(1908). Dorigo (1961) reviews the sea-level observations in Venice and summarizes the main development stages of the
observational network in the Venice Lagoon, including lists of active and decommissioned tide-gauge stations. Battistin and
Canestrelli (2006) provide the most recent review of tide-gauge data for Venice and collect quality-checked published and
unpublished records of high and low waters since 1872.

Linking the data from the various tide gauges to provide one continuous dataset of long-term sea-level change requires an
accurate knowledge of the corresponding reference levels (or datums) on land. Before the 1910s, the most common vertical
reference level in Venice was the so-called 'comune alta marea' or 'comune marino' (CM). The CM represents the upper edge
of the green belt formed by algae on quays and walls and corresponds to mean high water. It was often indicated by an engraved
horizontal segment and/or a 'C' (Rusconi, 1983; Camuffo and Sturaro, 2004). According to Dorigo (1961), Mati established
the tide gauge datum at Santo Stefano at 1.50 m below the CM of 1825. In 1910, a new datum was adopted, namely the mean
tide level (MTL) of 1884-1909 (central year 1897), computed from the high and low waters measured at Santo Stefano.
According to Dorigo (1961), it turned out to be 1.2754 m above the tide gauge datum and 0.2246 m below the CM of the year
1825. This new reference was named the "Zero Mareografico Punta Salute" (ZMPS). Since 1910, the ZMPS has been the
standard reference for RSL observations in Venice. The benchmarks of the two tide gauges at Punta della Salute were also
connected to the levelling network in 1910 and 1923, respectively. The heights of the various benchmarks and vertical
reference levels are shown in the inset of Figure 2. The record of high and low waters since 1872 allowed a composite 148-
year RSL time series to be developed from 1872 to 2019, with very few gaps (Fig. 2). Note that these data consist of MTL

which, in principle, differ from the MSL computed from, e.g., hourly data, because MTL does not account for shallow water tidal effects. However, the difference is negligible in Venice. In fact, from observations covering the period 1940-2012, Zerbini et al. (2017) obtained MTL-MSL = -0.1±0.1 cm, and the Permanent Service for Mean Sea Level provides a value of 0.0 cm, estimated according to Woodworth (2017) (www.psmsl.org/data/obtaining/stations/168.php).

Estimates of centennial rates of Venetian RSL and VLM-corrected RSL rise based on tide-gauge data are summarized in Sect. 4.1.

## 2.2 Altimetric data

Since the first satellite altimetry missions in the mid-1970s, the accuracy of sea-surface height estimates has increased considerably until high-precision and routinely measured altimetric data were made available in the early 1990s with the launch

of the TOPEX/Poseidon mission. Since then, satellite radar altimeters have been providing an operational global monitoring of the GSL (Cazenave et al., 2019). The spatial resolution of these data is controlled by the orbital parameters selected for each mission, as the radar altimeters acquire narrow threads of measurements along those portions of the ocean surface that are directly overflown by the satellite (along-track data). Depending on the orbital period, these tracks can be separated by hundreds of km, limiting the actual spatial coverage provided. For example, the Jason-3 mission that continues the climatic

sea-level record started in 1993 with TOPEX/Poseidon has only four tracks crossing the Adriatic Sea and does not cover the Venice area. To improve the monitoring, it is possible to take advantage of the data collected by various radar altimeters flying at the same time. However, this requires a characterization of the inter-mission biases and the development of suitable interpolation schemes of the independent ground tracks. Multimission datasets are typically distributed over regular grids.

An additional potential limiting factor for the Venice area is the degradation of the technique performance towards the coast

resulting from the contaminating presence of land in the satellite footprint and from the enhanced inhomogeneity of the local ocean surface. Limitations and possible perspectives of coastal altimetry in the Adriatic Sea have been discussed in several studies since the late 1990s (Cipollini et al., 2008; Fenoglio-Marc et al., 2012; Vignudelli, 1997; Vignudelli et al., 2011, 2019a). This has motivated further investigations (Cipollini et al., 2013; Passaro et al., 2014) based on the latest coastal altimetry datasets (e.g., CTOH, see Birol et al., 2017) and/or reprocessing initiatives (e.g., ALES, see Passaro et al., 2014). In the

Northern Adriatic, these studies analyzed GSL data around Venice and Trieste, including their validation against tide-gauge measurements. The results show that a reasonable increase in quantity and quality of data can be achieved compared to standard products up to a few kilometers from the coastline. Figure 3 illustrates the example of the Gulf of Trieste, where three missions cross the area and a data gap exists with standard products. In this case, the number of outliers along the Jason-1 and Jason-2 tracks is almost always less than the standard product and the improvement is clear until 6 km from the coast. The comparative

assessments with tide gauges confirm that the correlation of coastal altimetry products is always higher than that of standard products and that the difference in sea-level estimates provided by the two techniques is typically below 10 cm in proximity of the point of closest approach to the tide gauge. Among the most relevant reprocessing efforts of the last years, we should mention the Sea Level Climate Change Initiative (SLCCI) of the European Space Agency, that encompassed nine satellite

radar-altimetry missions over the period 1993-2015 (Legeais et al., 2018). The SLCCI product, distributed over a homogenous grid of 0.25°, contains data close to the coast, e.g., 10 km near Trieste, and was used for the assessment of coastal sea-level trends (Rocco, 2015; Vignudelli et al., 2019b).

The sufficient maturity of the algorithms and processing in coastal altimetry offered the opportunity to extend the SLCCI product to the coastal zone. During the bridging phase in 2018 a new product with an along-track spacing of about 350 m for estimating GSL trends has been developed in selected regions, including the Adriatic Sea. The experimental dataset only covered the period from July 2002 to June 2016 and the Jason-1 and Jason-2 missions. It combines the post-processing strategy of X-TRACK (Birol et al., 2017) and the advantage of the ALES re-tracker (Passaro et al., 2014). The product was tested along track 196 in the Gulf of Trieste. The improvement is particularly good in the entire Gulf of Trieste (ESA CCI, 2019), confirming what was found by Passaro et al. (2014).

Altimetry-based assessments of multidecadal trends of Venetian GSL are summarized in Sect. 4.2.

## 3 Vertical land movement

Vertical land movement is a critical component of Venetian RSL variability. Therefore, the phenomena that control the vertical land movement are presented in the following together with their relevant time scales and estimated trends. This section includes a paleo perspective on vertical land movements and considers processes whose characteristic time scales extend in some cases largely beyond the observational RSL period. We consider information on such time scales essential in the context of this review to understand ongoing processes and frame them within the correct time scale. The aim is therefore to provide the Reader with an overview of the main characteristics of the local vertical land movement, of the methods that allow quantifying it over different time intervals and of the resulting uncertainties. The joint consideration of all these elements determines the constraints on our current ability to make predictions on the future evolution of the local vertical land movement.

The vertical velocity of a given area results from the sum of different velocity components due to tectonics, sediment loading, sediment compaction, Glacial Isostatic Adjustment (GIA), and anthropic activities (Carminati and Di Donato, 1999; Pirazzoli, 1996).

In the Venice area, all the components listed above induce non-negligible displacements, even though their magnitude and relative importance have changed over time. The net result is a time-dependent land lowering (subsidence) that enhances RSL. Natural and anthropogenic components are assumed to act on different time scales: millions to thousands of years and hundreds to tens of years, respectively. This assumption allows a separation of the factors controlling sea-level changes, if the estimates of vertical land movements over different time spans are available (Carminati and Di Donato, 1999).

### 3.1 Natural land movements

The Venice area is naturally subsiding. This process is characterized by a long-term component controlled by tectonics/geodynamics and sedimentation, active on time spans of about $10^6$-$10^4$ yr, and a short-term component controlled by glaciation cycles and due to GIA processes acting on periods of $10^3$-$10^4$ yr (Antonioli et al., 2017; Cuffaro et al., 2010; Stocchi et al., 2005).

Depending on the time interval considered, different datasets are available for investigating the rate of vertical land movement. Subsidence rates up to 2 Myr ago can be inferred from the thickness of the different layers of Quaternary sediments. Over this time frame, sedimentation rates are equivalent to subsidence rates, since the entire sedimentary sequence was deposited in shallow marine to continental environments (Massari et al., 2004). Investigation further in the past is made through seismic lines, which indicate buried interfaces between materials of different acoustic impedances, and drilled cores. Deposition rates can be computed using sedimentological indicators (Antonioli et al., 2009, and references therein; Carminati and Di Donato, 1999; Favero et al., 1973), nannofossil biostratigraphy, paleomagnetic polarity and magnetic susceptivity (Kent et al., 2002). Additional techniques are available for more recent epochs. Radiocarbon dating allows investigating organic sediments, mainly peats, up to ~50,000 years ago (Bortolami et al., 1985), while the depth of archaeological remains and historical data provide information for the last few thousand years (Flemming, 1992). Finally, information on the natural component of the contemporary land subsidence is provided by tide-gauge and leveling measurements made before the 1930s, when human activities impacting land subsidence started to develop (Gatto and Carbognin, 1981). The following sections illustrate the evolution of the natural component of subsidence using the Marine Isotope Stage (MIS) 5.5 event (~130-120 kyr ago) as a reference to separate geologically older and newer RSL changes. Due to its relevance within geophysical studies on sea-level variations, a dedicated section on GIA is also provided.

### 3.1.1 Pleistocene up to MIS 5.5

The natural subsidence of Venice on timescales from tens of millennia to millions of years is controlled by sedimentary and tectonic/geodynamic processes. Venice is located at the northeastern border of the Po plain (Figs. 1 and 4), which is the foreland basin of two fold-and-thrust belts: the N-NE vergent Northern Apennines and the S vergent Southern Alps (Carminati et al., 2003). Figure 4 shows the geometry of the foreland regional monocline related to the subduction of the Adriatic plate (that includes the Po plain) below the Northern Apennines from the southern Po plain to the Friuli Region, as reconstructed from seismic reflection profiles. The dip of the regional monocline gradually decreases from about 22° to close to 0°. This geometry is consistent with the southward increasing thickness of Quaternary sediments, found in borehole stratigraphies (Carminati and Di Donato, 1999). These data imply that the long-term component of subsidence in the Po Plain and in Venice is almost entirely controlled by the retreat and flexure of the Adriatic plate subducting underneath the Apennines (Carminati et al., 2003; Cuffaro et al., 2010).

Kent et al. (2002) derive a more complex evolution for the Venice area from integrated magneto-bio-cyclo-stratigraphy and palynofloral analyses on the VENEZIA-1 borehole, drilled in 1971 by the Consiglio Nazionale delle Ricerche (CNR, 1971) down to a total depth of 950 m. They concluded that the region collapsed at about 1.8 Myr ago and was characterized by slow marine sediment accumulation until around 0.8 Myr ago, shoaling rapidly in subsequent times. The initial transition to continental sediments occurred during a glacioeustatic low-stand dated at 0.43 Myr or 0.25 Myr. From the VENEZIA-1 record, Kent et al. (2002) calculated a total long-term subsidence rate of less than 0.5 mm/year, about half of that proposed earlier on less refined data, and a mean subsidence rate of 0.36 mm/year for the last 600 kyr. The latter value is considerably lower when compared with rates obtained for the Holocene and the upper MIS 5.5. The most reasonable interpretation is that the mid-Pleistocene rates are unavoidably averaged over many cycles of quiescence and rapid motion, thus they cannot be readily compared to shorter periods, which could experience phases of rapid change induced by both natural and anthropogenic factors. Concerning natural variations acting on shorter time scales ($10^3$-$10^4$ yr), several transgressive/regressive Pleistocene cycles are recorded in well-core stratigraphies consisting of alternating shallow marine and continental deposits. In the Late Quaternary, the evolution of the Venetian–Friulian plain was strongly influenced by glacial cycles and a general regressive trend is apparent (Massari et al., 2004). The coastal to shallow-marine deposits assigned to MIS 5.5 can be tracked in borehole logs up to 30 km west of the present shoreline. South of the Po Delta, the base of the Tyrrhenian coastal deposits lies at about 125 m below sea level, but its depth rapidly increases toward the south along the Romagna coastal plain (Amorosi et al., 2004; Bondesan et al., 2006). This pattern may reflect the northeasterly retreat of the Adriatic slab (Cuffaro et al., 2010; Ferranti et al., 2006).

The MIS 5.5 markers allow calculating reliable rates because compaction is negligible, the basal MIS 5.5 unconformity is widely distributed and the overlying lagoonal paralic sediments in cores are easy to recognize. Several sites related to sea-level position during MIS 5.5 are considered in Antonioli et al. (2009) and Lambeck et al. (2011). These have a fairly good W–E distribution along the distal sector of the Venetian plain. The stratigraphic data were obtained from boreholes mainly drilled for the Geological Map of Italy (CARG-Veneto Region) and for the mobile barriers-based protective system (so-called "MOdulo Sperimentale Elettromeccanico" or MOSE, see Lionello et al., 2020a) project by the Venice Water Authority. An error bar of ±2 m was assigned because the sediments are lagoonal. The northwestern Adriatic coast (Friulian and Venetian plain) shows homogeneous subsidence, with rates ranging between 0.58 and 0.69 mm/year. The MIS 5.5 data from the VENEZIA-1 core provides a rate of 0.69 mm/year (Ferranti et al., 2006).

**3.1.2 Late Pleistocene and Holocene**

After the Last Glacial Maximum several lagoons developed along the Adriatic Sea, formed by the rivers flowing into the sea. Only two of them, the Grado and Venice lagoons, still exist today, while the rest have been infilled by sedimentation (Tambroni and Seminara, 2006). Recent stratigraphic information about Holocene sea levels (2-6 kyr Cal BP) were obtained from lagoonal deposits found in boreholes between the Tagliamento River and the city of Venice. Other data were derived from archaeological markers reported in the abundant literature available for the Venice Lagoon and its mainland (Antonioli et al., 2009; Fontana et al., 2017; Lambeck et al., 2011). The shell base of the lagoon indicates a subsidence rate over the last 7.3 kyr

of 1.6±0.3 mm/year (Antonioli et al., 2009, their Tab. 1 average of H/G values for sites 26 and 28). The higher Holocene subsidence with respect to the MIS 5.5 is possibly due to sediment compaction, which does not contribute to the long-term rate (Gatto and Carbognin, 1981; Tosi et al., 2009).

Subsidence rates up to 1.2-1.3 mm/year were calculated by radiocarbon dating on late Pleistocene and Holocene deposits of the Venice Lagoon (e.g., Bortolami et al., 1985; Gatto and Carbognin, 1981). This estimate is interpreted as the average of a time-varying trend related to periods of excess sedimentation alternating with periods without deposition or even with erosion (Bortolami et al., 1985). Indeed, over relatively short periods, different rates can be observed. For instance, the largest rate, ~5 mm/year, occurred during the Last Glacial Maximum which induced the maximum effect of isostatic lowering.

Finally, it is generally assumed that natural subsidence of Venice is continuous in time. However, abrupt catastrophic pulses of subsidence cannot be ruled out as suggested by the sudden disappearance of the island of Malamocco at the beginning of the XII century. Carminati et al. (2007) investigated the potential effects of earthquakes on the subsidence of Venice by means of numerical models. The authors concluded that, while the coseismic effects of a single event are unlikely to be detectable, *a priori* they cannot be considered as negligible given the number of seismogenic sources within a 100 km distance from the

town. These authors, however, suggest that earthquake-induced liquefaction may cause or have caused local acceleration of subsidence in Venice. For example, the destruction and sinking of ancient Malamocco is roughly coincident with a strong earthquake cycle that was associated with phenomena possibly explained by liquefaction of sandy layers.

A summary of the rates of natural subsidence discussed in this and in the previous sections is presented in Table 2. The values reported in the literature are often presented without indicating the corresponding uncertainty level. In some cases, it is even

explicitly stated that the data available to the study did not allow for a quantification of uncertainty (e.g., Carminati and Di Donato, 1999). Uncertainty estimation is further complicated by the fact that subsidence does not only vary with time, but also in space, depending on the local conditions of the subsoil (Brambati et al., 2003). For what concerns uncertainties of geomorphological and historical markers, Antonioli et al. (2009, 2017) proposed a strategy based on archaeological metadata and on standard bathymetric corrections for the Holocene and late Pleistocene (Ferranti et al., 2006; Lambeck et al., 2004).

The resulting median uncertainty for the Venice area is 0.2 mm/year (markers 24-30 in Antonioli et al., 2009, their Table 1).

### 3.1.3 Glacial Isostatic Adjustment

GIA describes the response of the solid Earth to the growth and decay of continental ice sheets because of climate variations (for recent reviews, see: Spada, 2017; Whitehouse, 2018). GIA stems from interactions between the cryosphere, the solid Earth and the oceans, involving sluggish deformations of the crust driven by surface mass redistribution, mutual gravitational

attraction and rotational variations (Melini and Spada, 2019; Spada and Melini, 2019). The GIA-induced RSL variations are characterized by a strong regional imprint reflecting such interactions. They can be modeled by means of the Sea Level Equation first introduced by Farrell and Clark (1976), which is an implicit equation that accounts for variations of the Earth's topography in response to sea-level change, consistently with changes in the gravity field (Peltier, 2004). Among the processes contributing to present-day RSL change (e.g., Milne et al., 2009), GIA is the only one that is sensitive to the solid Earth

rheology. Because changes in the Earth system observed by geodetic methods would be unfeasible without taking GIA properly into account (e.g., King et al., 2010), GIA modeling plays an important role in the study of the impacts of contemporary and future climate change.

Due to the widespread evidence of past RSL variations since the late Holocene across the Mediterranean Sea, much work has been done to reconcile field observations of past RSL variations with GIA modeling predictions (Antonioli et al., 2009, 2017 and references therein). In two recent contributions, attention has been paid to the history of sea level in the Northern Adriatic, also providing GIA modeling predictions for the city of Venice. The first one (Lambeck et al., 2011) is based on the ice-sheets history "K33_j1b_WS9_6"; it assumes a 65-km thick elastic lithosphere and one order of magnitude viscosity increase across the 670 km depth seismic discontinuity. The second one, proposed by Roy and Peltier (2018) and named "ICE-7G_NA(VM7)", is characterized by a 90-km thick lithosphere and by a comparatively milder viscosity increase (by a factor of ~3). The two models predict distinctly different histories for the GIA-induced RSL variations during the Holocene: the first shows ~2.2 m of RSL rise in the last 5,000 years, whereas the second indicates essentially unvaried RSL during the same period. Note that in previous work (Lambeck et al., 2004), the GIA predictions for the Northern Adriatic had a larger uncertainty, with a range of RSL rise between ~5 and ~2 m in the last 5,000 years. This shows that GIA models are constantly being updated due to improvements in the constraining RSL datasets and of modeling techniques.

Based on the work quoted above, the rate of long-term RSL change in Venice due to the melting of the late-Pleistocene ice sheets does not appear to be tightly constrained (Tosi et al., 2013). Further uncertainties arise from the effects of the melting of the Würm Alpine ice sheet, whose chronology remains uncertain regarding several aspects (Spada et al., 2009). Nonetheless, the long-term rate of Venetian RSL change due to GIA can be assessed in the range between -0.2 and +0.5 mm/year based on the published works cited above. Estimates by Carminati and Di Donato (1999) and Stocchi and Spada (2009) broadly fall within this range, although these works are more pertinent to the Po Plain scale. Note that since the GIA acts on timescales of millennia, these natural contributions to total RSL change will remain constant over the 21$^{st}$ century.

## 3.2 Anthropogenic subsidence

Anthropogenic land subsidence mainly occurs due to extraction of subsurface fluids causing compaction of unconsolidated sediments. This is a process that is widespread in susceptible areas (e.g., Gambolati et al., 2006; Galloway and Riley, 1999; Erkens et al., 2015; Galloway et al., 2016). Measurements of piezometric level and of vertical land movements are fundamental to constrain quantitatively these processes. Numerical modeling is often used to link the flow of subsurface fluids to the corresponding geomechanical response of the porous medium, although caution is needed. In fact, the paucity of geological data, the imperfect knowledge of forcing processes and the geomechanical and hydraulic properties generally require significant modeling assumptions and approximations. These techniques have been used to analyze and control the effects of human activities on subsidence in the Venice area.

Prior to 1930, subsidence rates in the Venice region were similar to Holocene rates, suggesting limited anthropogenic contribution. This is confirmed by both leveling measurements (Dorigo, 1961; ISPRA, 2012; Wöppelmann et al., 2006) and differences in RSL trends between Venice and Trieste. The post-1930 period is now considered in more detail.

### 3.2.1 The 1930-1970 period

In the Po and Veneto Plains, anthropogenic activities affecting natural land subsidence mainly began in the 1930s due to the overpumping of groundwater and natural gas to support intense civil and industrial development, as shown by geodetic data and reproduced by numerical models (Gambolati et al., 1974; Gambolati and Gatto, 1975; Carbognin et al., 1976). Between World War II and 1970, anthropogenic subsidence was a problem common to the whole Northern Adriatic coastline (Tosi et al., 2010). However, the nature of the withdrawn fluids varied: artesian water in the Venice area, gas-bearing water in the Po

Delta and both groundwater and gas in the Ravenna region. Anthropogenically driven subsidence rates of 10 to 20 mm/year and even higher occurred in certain locations (Carminati and Di Donato, 1999; Teatini et al., 2005), dominating there the RSL change.

In the Venice area, large quantities of groundwater were pumped to develop the industrial zone of Marghera after 1930. Groundwater pumping was most intensive after World War II during the period of greatest industrial growth. The six aquifers

found in the upper 350 m were progressively exploited (Fig. 5a); the most intensively used aquifer was between 200 and 250 m depth due to its productive character (Carbognin et al., 1976).

Between 1950 and 1970, human-induced subsidence reached 14 cm at Marghera and averaged 10 cm at Venice (Fig. 5d). The dramatic effects of this loss of elevation became apparent in the exceptional flood ("acqua alta") of November 1966. The ground beneath Venice is more sensitive to changes in the hydraulic head because of the occurrence of a larger amount of clay

in the subsurface (Zezza, 2010): the ratio between subsidence and piezometric decline was 1/100 at Marghera and 1/50 at Venice (Gatto and Carbognin, 1981).

### 3.2.2 The post-1970 period

After the 1966 flood, the problem of subsidence in Venice received more attention and drastic measures were taken after 1970 to reduce both industrial and other groundwater extraction. Groundwater consumption in the Marghera area decreased from

340 500 l/s in 1969 to 170 l/s in 1975 (Gatto and Carbognin, 1981). A corresponding rapid piezometric rise occurred (Fig. 5a): in 1978 the hydraulic head rose to ground level, re-establishing the levels existing in 1950 (Gatto and Carbognin, 1981). Land subsidence slowed concurrently and stopped; by 1975, a surface rebound of about 2 cm was recorded (Fig. 5d), equal to 15% of the total anthropogenic subsidence experienced. This result is consistent with mathematical model results and was interpreted as the elastic response of cohesive soils after recovery.

In recent years, Global Navigation Satellite System (GNSS) and Synthetic Aperture Radar (SAR) measurements confirm that the city of Venice is no longer sinking due to groundwater pumping (Tosi et al., 2013). However, at the local scale, ground movements are still impacted by anthropogenic activities such as new construction and conservation works dedicated to

preserve the Venetian architectural heritage. Tosi et al. (2018) estimated that about 25% of the city experienced movements attributable to anthropogenic causes. In most cases (15%) these displacements induced an increase in local subsidence, but in some areas (10%) the short-term sinking rate was found to be smaller than the natural one. The measured displacement rates range between -10 and 2 mm/year.

### 3.3 Monitoring land subsidence

Over the 20[th] Century, high-accuracy geodetic techniques became available for monitoring land subsidence with unprecedented temporal and spatial resolution. These new data were key to reveal the increasing impact of human activities on the subsidence rate.

The first direct measurements of changes in land elevation were obtained through leveling campaigns based on both local and national networks (Salvioni, 1957; Gambolati et al., 1974; Gatto and Carbognin, 1981; Arca and Beretta, 1985; Carbognin et al., 1995a, 1995b). Additional information was derived by comparing the tide-gauge records acquired in Venice with those available in neighboring areas subjected to the same absolute sea-level changes (Carbognin et al., 2004; Zerbini et al., 2017). These techniques allowed an unambiguous detection of the impact of anthropic activities on land lowering. It was possible to identify the increase in subsidence between the 1950s and the 1970s caused by severe groundwater extraction (Gambolati et al., 1974). The maximum rate was observed in Mestre in 1968-69 when the local subsidence reached 17 mm/year (Brambati et al., 2003). Leveling lines also provided information on the spatial variability of subsidence over a few tens of kilometers. The cone of land depression was found to spread from Marghera, where most of the pumping occurred, towards the Venice area (Fig. 5).

During the following decades, leveling measurements performed in 1973 and 1993 recorded the slowdown in subsidence rates, and even a small uplift, which followed the dismissal of artesian wells and the diversification of water supply (Carbognin et al., 1995b, 1995a).

The monitoring capabilities further improved during the 1990s with the development of space techniques such as GNSS and SAR (Teatini et al., 2012; Tosi et al., 2013, 2018; Zerbini et al., 2017). The latest measurements provided by the integrated use of these techniques confirmed that, in Venice, the anthropogenic subsidence due to activities characterized by large-scale and long-term effects ended a few decades ago. However, relevant trends are still observed locally. Subsidence up to 70 and 20 mm/year is found around the inlets where the MOSE is being constructed and in artificial salt marshes, respectively (Tosi et al., 2018). In addition, spatial patterns in subsidence have been identified at different scales. The average ranges of subsidence rates observed over the lagoon are 3-4, 1-2 and 2-3 mm/year for the northern, central and southern parts, respectively (Tosi et al., 2018). This reflects both the increase in the thickness of Holocene deposits from the Venice mainland to the lagoon extremes and residual groundwater pumping in the northeastern sector (Tosi et al., 2013). In the historic center of Venice, the ancient areas are more stable than those urbanized over the last centuries. This is consistent with the older settlements being developed on well-consolidated sand layers, while recent land claims occurred over areas where consolidation processes are still ongoing. Thanks to the high spatial resolution of SAR images, Tosi et al. (2018) were able to

detect the impact of restoration work and new construction down to the single-building scale. This variability in displacement correlates with the nature of the shallow subsoil, the different phases of growth of the city, and the load and foundation depth of the buildings.

Table 3 presents the evolution of subsidence rates in the historical center of Venice, as measured by geodetic instruments over the last century. The estimates refer to different periods of time as well as different locations, and they result from different techniques, for each of which associated uncertainty is available in the literature. Precise leveling allows measuring height differences with a mean error ranging from 0.3 to 1 mm in a line of 1 km (Torge and Müller, 1980). The average uncertainty for the vertical component of the GNSS trends is in the order of 0.3 mm/year for time series spanning a decade or more (Santamaría-Gómez et al., 2017). This estimate increases to 0.4 mm/year when reference frame uncertainties are considered in the error budget (Santamaría-Gómez et al., 2017). Finally, Tosi et al. (2013) propose to present the results of the SAR technique over a selected area in terms of average rate and standard deviation of the spatial variability. By doing so, the technique provides insights on the representativeness of the estimated trend at the investigated spatial scale. Therefore, the two GPS-based trend estimates reported in Table 3 are both consistent with the average value obtained by the SAR technique over the whole city.

## 4 Estimation of sea-level changes

This section reviews the estimates of centennial average rates of sea-level rise and recent multidecadal trends, and interannual-to-interdecadal variations identified in historical sea-level records for the Venice Lagoon and its surroundings, and puts them in the context of observed sea-level changes in the Mediterranean Sea and the global ocean. Average rates of sea-level rise are often calculated using some linear fit to the available data. However, given the variety and non-linearity of the processes known to contribute to sea-level trends on the considered time scales, such estimates should not be intended as necessarily representing a linear process (see Section 6).

### 4.1 Average rates of sea-level rise over centennial periods

One of the first estimates of the long-term RSL trend at Venice Punta della Salute was made by Polli (1953), who obtained 2.3±0.2 mm/year performing a least-square fit of the annual means from 1872 to 1941. Since then, several authors have proposed updated estimates by progressively considering newly acquired data and different approaches. A summary of the long-term RSL trends from tide-gauge data proposed during the last 15 years is presented in Table 4; estimates derived from data collected over the considered altimetric period (i.e., since 1993) are pointed out. Analogous VLM-corrected RSL estimates are presented in Table 5. Marcos and Tsimplis (2008), Wöppelmann and Marcos, (2012) and Vecchio et al. (2019) used a linear fit to analyze the RSL data from about 1910 to 2000, obtaining trends between 2.4 and 2.5 mm/year. Vecchio et al. (2019) also modelled the time series by means of the superposition of a straight line and three Empirical Mode Decomposition components, suggesting a slightly larger trend of 2.78 mm/year. Zerbini et al. (2017) isolated the effect of subsidence on the

Venetian RSL time series by deriving an empirical curve from levelling data of benchmarks close to the tide gauge, GPS and InSAR heights (Fig. 5c). After removing the estimated subsidence from the tide-gauge data, the centennial average rate of sea-level rise of the corrected time series was 1.23±0.15 mm/year for the period 1872-2012 (see Table 5). It should be stressed that the trend analysis has little meaning without the correction of subsidence effects because the linear model is otherwise inadequate to represent the Venice time series. In Zerbini et al. (2017), the application of the same procedure to the neighboring tide gauge of Marina di Ravenna, also located in a rapidly subsiding area, provided a consistent estimate of 1.22±0.32 mm/year (period 1896-2012).

These corrections allow for a proper comparison with other tide-gauge records in the Mediterranean Sea that span the whole 20th century and are not affected by significant vertical land motions, namely Trieste in the Adriatic and Marseille and Genoa in the northwestern Mediterranean (Carbognin et al., 2009; Wöppelmann et al., 2014; Zerbini et al., 2017; Sánchez et al., 2018) (Fig. 1). The RSL time series of Trieste, Marseille and Genoa exhibit centennial trends between 1.2 and 1.3 mm/year (Marcos and Tsimplis, 2008; Wöppelmann and Marcos, 2012; Zerbini et al., 2017). The estimates agree with the 20th century trend of sea-level rise of 1.2±0.1 mm/year reported for the same stations by Marcos et al. (2016). The 1-sigma errors are around 0.1 mm/year according to Marcos and Tsimplis, (2008) and Wöppelmann and Marcos, (2012), and between 0.10 and 0.24 mm/year (90% confidence) in Vecchio et al. (2019). Zerbini et al. (2017) obtained uncertainty values between 0.13 and 0.22 mm/year at 95% confidence level considering a reduced number of degrees of freedom due to time autocorrelation. Therefore, the 20th century average rate of sea-level rise in Venice is consistent within uncertainty with those of Marseille, Genoa and Trieste. Accordingly, EOF analysis on annual means from 1901 to 2012 of the corrected time series of Venice and Marina di Ravenna, and those of Marseille, Genoa and Trieste yields a leading mode explaining 62% of variance and corresponding to coherent sea-level variability of the long time series from Mediterranean tide gauges (Zerbini et al., 2017). Scarascia and Lionello (2013) estimated a trend of 1.3 mm/year for the period 1905-2005 for the Adriatic Sea using a combination of Adriatic tide gauges and manually removing the land subsidence in Venice.

Tables 4 and 5 include an update on the RSL trend calculation to the period 1872-2019 including a comparison between estimates with and without the subsidence contribution following Zerbini et al. (2017). The subsidence curve by Zerbini et al. (2017) was updated to 2019 by applying a 1 mm/year trend since 2013 based on the SAR estimate by Tosi et al. (2013) and on the trend exhibited by the PSAL GPS from 2014 onward (Table 3). Our updated estimates agree with previous results concerning the magnitude of the full-period average rates of sea-level rise in RSL (2.53 ± 0.14 mm/year) and VLM-corrected RSL (1.23 ± 0.13 mm/year). Subsidence therefore contributed to about half of the total RSL rise in the period 1872-2019 and explains discrepancies between published Venetian sea-level trends.

As illustrated in Sections 5 and 6, Mediterranean, hence Venetian, sea-level variations are tightly connected to sea-level variations in the midlatitude eastern North Atlantic, whose underlying processes differ from those in other oceanic basins. Therefore, any statistical consistency between historical Venetian RSL/VLM-corrected RSL and GMSL rise should not give the false impression that both variables are interchangeable and that any similarity in the historical period necessarily holds in the future. Still, it is relevant to compare estimates of Venetian RSL/VLM-corrected RSL and GMSL rise during the 20th

Century to put local changes in the context of global mean changes, since some of the available projections of future Venetian sea level are directly based upon estimates of the GMSL rise (see, for instance, Troccoli et al., 2012, and Carbogning et al., 2010).Venetian sea-level trends are smaller than GMSL trends reported in the fifth assessment report of the Intergovernmental Panel on Climate Change (IPCC-AR5), quantified as 1.7 [1.5 to 1.9] mm/year (likelihood >90%, period from 1901 to 2010, see: Church et al., 2013). They are, however, consistent with revisited estimates of historical GMSL rise that include significantly slower rates than reported by the IPCC-AR5 for the pre-altimetry period, e.g., 1.2±0.2 mm/year (90% confidence interval, Hay et al., 2015), $1.1 \pm 0.3$ mm/year (99% confidence interval, Dangendorf et al., 2017) and $1.56 \pm 0.33$ mm/year (90% confidence interval, Frederikse et al., 2020). Figure 6 revisits Venetian RSL/VLM-corrected RSL and GMSL trends on time scales ranging from interannual to centennial. Clearly, the significant difference between centennial trends in Venetian RSL and GMSL is strongly damped when the contribution of subsidence is removed, confirming the critical role of vertical land motions in determining local RSL variations. Nonetheless, the Venetian VLM-corrected RSL appears to rise at a lower rate than the GMSL over the second half of the 20[th] century (Fig. 6a). The GMSL-Venetian sea-level discrepancy observed in the first portion of the record is resolved when uncertainty in GMSL estimate is considered (not shown).

## 4.2 Rates of sea-level rise during the satellite altimetry era

Sea level measurements acquired with satellite radar altimetry are available since 1993, allowing, together with tide gauges, to estimate recent multidecadal trends from two independent sources. An overall global-mean GSL trend of about 3 mm/year during the satellite altimetry period is consistently reported by several studies (Hay et al., 2015; Chen et al., 2016; Dangendorf et al., 2017; Quartly et al., 2017). Regional trends can deviate considerably from the global mean (e.g., Scharroo et al., 2013; Legeais et al., 2018; Cazenave et al., 2019). This is also the case of the Mediterranean Sea that is subject to pronounced spatial and temporal variability (Figure 7a), with the entire area of the Adriatic Sea exhibiting positive GSL trends that peak in the northern part of the basin.

Several altimetric datasets have been used to estimate sea-level trends in the Venice area. Fenoglio-Marc et al. (2012a) estimated a trend of 5.9±1.4 mm/year over the period 1993-2008 for an along-track point about 80 km away from Venice (see their Table 2). Rocco (2015) obtained trends of 4.18±0.92 mm/year (period 1993-2014) and 3.40±0.99 mm/year (period 1993-2013) for the closest grid point to the Venice tide gauge in the AVISO and SLCCI V1 products, respectively, with both estimates consistent with each other within errors. A reprocessing of the SLCCI V2 data set over the period 1993-2015 yielded a trend of 4.25±1.25 mm/year (Vignudelli et al., 2019b), further reduced to 4.03±1.27 mm/year after removing the seasonal signal (ESA CCI 2019). Explanations for the differences between trend estimates in these studies include the different time spans, especially for Fenoglio-Marc et al. (2012), different methodological aspects in the spatial characterization of the study area (e.g., closest point vs. area with a certain radius), and the recurrent reprocessing and continuous improvement of the satellite radar altimetry products.

The altimetric trends derived for Venice are typically consistent with those estimated around Trieste over corresponding time spans (Fenoglio-Marc et al., 2012). This evidence is supportive of a rather uniform sea-level trend in the Northern Adriatic (Fig. 7a, see also Bonaduce et al., 2016).

A thorough comparison between tide-gauge and altimetric data in Venice is made possible by the availability of independent information on the evolution of the vertical land motion (Fenoglio-Marc et al., 2012; Wöppelmann and Marcos, 2016). For consistency with post-processed altimetry data (Carrère and Lyard, 2003), the tide-gauge time series need to be preprocessed for removal of the local effect of atmospheric forcing (see Sect. 5.2). The most recent trend estimates by Vignudelli et al. (2019b) provide values of 6.17±1.50 mm/year from in situ data at the Acqua Alta Platform (AAPTF), 14 km offshore the

Venice coast, and +5.81±1.47 mm/year at Punta della Salute (inside the city center) during the overlapping altimetry period. After subtracting altimetry and AAPTF tide-gauge time series, the residual time series shows a trend of -2.14±0.65 mm/year. This estimate agrees with the trend of 2.17 mm/year extrapolated from Figure 3 in Zerbini et al. (2017) that represents a best fitting of the benchmarks, GPS and PS InSAR normalized heights.

Table 4 and Figure 6 also provide updated RSL trend estimates for the period 1993-2019 based on the Punta della Salute tide-

490 gauge data. Our estimates confirm that for the satellite altimetry period the total RSL trend from the tide gauge (5.01 ±1.75 mm/year, including subsidence) is consistent with uncertainties with some satellite estimates and the tide-gauge estimate by Vignudelli et al. (2019b) over similar periods. Our estimate for the VLM-corrected RSL trend for the same period is 2.76±1.75 mm/year, again confirming the results by Vignudelli et al. (2019b).

## 4.3 Interannual-to-interdecadal variability

In addition to long-term changes, the tide-gauge time series of Venetian RSL is characterized by variability on several interannual-to-interdecadal periods. Significance of observed fluctuations was tested against a red-noise hypothesis, namely a lag-1 autoregressive model characterized by high power spectral density at lower frequencies. It is the most widely used model for geophysical purposes since a large class of geophysical processes produce output statistically compatible with the red noise hypothesis (Allen and Smith, 1996, and references therein). According to Grinsted et al. (2004), significance is computed

through the distribution of an ensemble of surrogate series describing a red noise process with the lag-1 parameter and variance estimated from the analysed time series. Hereafter, we indicate detection of a statistically significant spectral component around a period of XX years with $O_{XX years}$, where O means order of magnitude. Based on detrended seasonal Venetian RSL for the period 1872-2003, Zanchettin et al. (2009) report spectral peaks in the autumn (OND) time series at $O_{22 years}$ and at larger multidecadal time scales, at around $O_{2.4 years}$ and at around $O_{3.4 years}$, with secondary peaks at around $O_{5 years}$ and $O_{8 years}$. In the

winter (JFM) time series, they report significant multidecadal variability at $O_{50 years}$ and larger, at $O_{3.4 years}$, $O_{8 years}$ and, less apparent, $O_{5 years}$. Carbognin et al. (2010) also identify an $O_{8 years}$ component in Venetian RSL variability. An updated spectral analysis based on the Fourier transform applied on autumn and winter raw detrended (second order polynomial fit) Venetian RSL indicates that the dominant variability modes contained in the time series over the time interval 1872-2019 are the interannual components at $O_{2.4 years}$ and $O_{5 years}$, for both autumn and winter series (95% confidence level), that account for about

20% of the total variance of the records. Moreover, the winter time series features the $O_{8years}$ (~6% of the total variance) and $O_{50years}$ (~9% of the total variance) spectral peaks as highly significant. A secondary peak at $O_{16years}$ is detected at 90% confidence level in the autumn series (~7% of the total variance). Removal of subsidence does not change the spectral features of the series, except for the $O_{50years}$ component in the winter series, whose significance then only reaches the 90% confidence level. Lionello (2005), Barriopedro et al. (2010), Troccoli et al. (2012) and Martínez-Asensio et al. (2016) consistently identify significant decadal variability in the time series of autumn Venetian surge events for the period 1948-2008 (for an updated analysis see Lionello et al., 2020b, in this special issue).

Continuous wavelet transform analysis on updated and detrended seasonal time series of the Punta della Salute tide-gauge record confirms the presence of statistically significant interdecadal fluctuations in autumn ($O_{20years}$, period 1960-2000), and interannual ($O_{5years}$ and $O_{8years}$, periods 1930-1950 and 1970-1990, respectively) and multidecadal ($O_{50years}$, since 1950) fluctuations in winter (Figure 8). Such fluctuations, however, appear only over limited periods, typically for a few decades or even less. This intermittent recurrent interdecadal variability can significantly impact on sub-centennial trend estimates and contribute to explaining associated spatial features. For instance, in the period between the mid-1960s and the early 1990s, the RSL time series of Venice and Trieste appear almost stationary (Figs. 2 and 6). Marcos and Tsimplis (2008) estimated RSL trends to be zero (within the errors) in the 1960-2000 period at the tide-gauge stations of Trieste, Genoa and Marseille. So, stationary sea level characterized the whole Mediterranean Sea during this period but not the Atlantic Ocean, and proposed explanations include an atmospheric contribution mainly consisting of persistent high pressure and an oceanic contribution due to steric changes in deep water masses (Tsimplis and Baker, 2000; Tsimplis et al., 2005; Gomis et al., 2008). Scarascia and Lionello (2013) conclude that the mean sea-level rise of 1.3 mm/year in the Northern Adriatic in the period 1940-2005 is primarily due to ice cap melting, with no trends in atmospheric drivers, upper ocean temperature and surface layer salinity. The different estimates and attribution of multidecadal trends by Marcos and Tsimplis (2008) and Scarascia and Lionello (2013) reflect the strong interdecadal variability of Mediterranean sea level and the variety of its driving mechanisms. The former study computes the expansion of the water column missing the contribution of the redistribution of mass and produces negligible values of sea-level rise in shallow water areas, while the latter accounts for it and explicitly considers the Adriatic Sea. Figure 6c confirms that often bidecadal trends, but occasionally also longer ones, in annual-mean Venetian VLM-corrected RSL are negative and can differ in sign from the GMSL trend. Accordingly, the linear trend can yield a local sea-level anomaly in Venice from the GMSL of about 10 cm (but up to about 20 cm occasionally) over bidecadal and shorter periods, and a rather small anomaly (generally <5 cm) over interdecadal and longer periods (not shown).

## 5 Climatic drivers of Venetian sea-level variations

Variations of Venetian RSL closely depend on sea-level variations in the Adriatic Sea, which in turn are linked to sea-level variations in the Mediterranean Sea. It is well established that basin-wide sea-level fluctuations in the Mediterranean Sea are

associated with sea-level variations in the eastern boundary of the North Atlantic Ocean on a broad range of time scales (e.g., Fukumori et al., 2007; Calafat et al., 2012; Landerer and Volkov, 2013; Marcos et al., 2016; Volkov et al., 2019). Such linkage is largely determined by the water-mass exchange between Atlantic Ocean and Mediterranean Sea through the Strait of Gibraltar (e.g., Brandt et al., 2004; Fukumori et al., 2007; Menemenlis et al., 2007; Gomis et al., 2008; Calafat et al., 2012;

Landerer and Volkov, 2013; Tsimplis et al., 2013; Adloff et al., 2018; Volkov et al., 2019). Section 5.1 provides further details on the lateral boundary forcing at the Strait of Gibraltar and its drivers. The water-mass signal at Gibraltar then propagates within the Mediterranean Sea almost uniformly, yielding basin-wide sea-level variations (e.g., Fukumori et al., 2007; Calafat et al., 2012) and overwhelms steric changes as far as basin-average sea-level trends are concerned (Jordà and Gomis, 2013).

Spatial heterogeneity in Mediterranean sea-level variations is determined by a combination of local steric and ocean circulation

changes as well as local atmospheric mechanical forcing, which are described in Section 5.2. Another potential contributor to local sea-level changes is freshwater horizontal advection (Jordà and Gomis, 2013), primarily linked to river runoff. At the basin scale, riverine input to the Mediterranean Sea and other freshwater fluxes resulting in net loss of freshwater are regarded as negligible for sea-level variability, as they are quickly compensated by changes in the mass transport through the Strait of Gibraltar (Adloff et al., 2018). However, the conspicuous and noticeably time-varying inflow of freshwaters from the Italian

Po River, whose delta is located only 90 km south of the Venice Lagoon (Fig. 1), contributes shaping haline properties in the Adriatic Sea (Taricco et al., 2015) with detectable imprint on Venetian sea levels (Zanchettin et al., 2009).

Venetian RSL variability has been investigated in relation to modes of large-scale climate variability, especially concerning teleconnections during wintertime and with the dominant mode of atmospheric variability over the North Atlantic on interannual timescales, known as North Atlantic Oscillation or NAO. The geostrophic winds caused by the large-scale

atmospheric pressure anomalies linked to the NAO drive a barotropic sea-level response in Venice, especially in winter. Given its relevance in the scientific literature, the connection between NAO and Venetian as well as Mediterranean sea level is reviewed in detail in Section 5.2. In addition to the NAO, statistical connections identified in the literature between Venetian RSL and climatic modes include the atmospheric patterns known as Scandinavian and East Atlantic Western Russia (Zanchettin et al., 2009), showing prominent variability at interannual-to-decadal time scales in the autumn and winter seasons,

and the Atlantic Multidecadal Oscillation or AMO (Scafetta (2014), describing multidecadal fluctuations in North Atlantic sea-surface temperature and influencing atmospheric variability over the Euro-Mediterranean region (e.g., Mariotti and Dell'Aquila, 2012; Maslova et al., 2017).

## 5.1 Lateral boundary forcing at the Strait of Gibraltar

Water mass exchange through the Strait of Gibraltar constitutes the critical lateral boundary forcing to determining how sea-

level signal propagates from the midlatitude eastern North Atlantic into the Mediterranean basin. The exchange at Gibraltar consists of a strong surface current of relatively fresh and warm water from the ocean and a deep-water current of salt and cold Mediterranean water, outflowing into the ocean and sinking in the North Atlantic in the form of gravity current. Water mass exchanges across the Strait critically depend on the number and location of its hydraulic controls, being sub-maximal if subject

to only one control in the western part, or maximal if the flow is also controlled in the eastern part, with different implications for the characteristics of the circulation. Local dynamics are strongly influenced by tides, which are responsible for the modulation of the water transport and the hydraulic control (Armi and Farmer, 1988) as well as for the substantial vertical mixing that has been observed (Garcìa-Lafuente et al., 2013).

Nonseasonal mass exchanges through the Strait up to the decadal time scale are modulated by fluctuations in the winds over and near the Strait (Fukumori et al., 2007; Landerer and Volkov, 2013; Volkov et al., 2019). On longer time scales, oceanic contributions mainly including steric changes and mass addition/removal as well as geological processes such as GIA become predominant. These contributions are specific for each basin of the World Ocean and especially so for the subpolar and eastern North Atlantic due to the presence of an active thermohaline circulation and to its proximity to the Greenland ice sheet. There, GIA was predominant over the ocean-mass contribution to determine the upward sea-level trend over the 20[th] Century; since the late 1950s unbiased estimates of steric changes are also available, indicating a contribution to the sea-level rise comparable to GIA (Frederikse et al., 2020). Steric changes particularly contribute to multidecadal sea-level variability in the different ocean basins (Frederikse et al., 2020), with the Atlantic Ocean accumulating heat at a higher rate than the Pacific and Indian Oceans (e.g., Zanna et al. 2019). So, despite the overall sea-level trend for the subpolar and eastern North Atlantic over the 20[th] Century is comparable to GMSL trend, primarily driven by changes in the ocean mass, it is caused by a different combination of processes (Frederikse et al., 2020).

Recent progress in the simulation of water mass exchange through the Strait of Gibraltar led to a major improvement in the simulation of Mediterranean circulation. Sannino et al. (2015) demonstrated that the inclusion of explicit tidal forcing in an eddy resolving Mediterranean model has important effects on the simulated circulation, in addition to the expected intensification of local mixing processes. Marcos et al. (2016) decompose the Mediterranean sea-level signal into two components: first, variations in the eastern North Atlantic sea level, estimated through global coupled climate models, second, relative variations in the Mediterranean sea level with respect to the eastern North Atlantic, estimated through regional climate models. More recently, Adloff et al. (2018) provide an overview of current methods to implement Atlantic sea-level forcing at the lateral boundary of state-of-the-art regional ocean models for the Mediterranean Sea, concluding that the quality of such forcing is essential for appropriate modelling of Mediterranean sea level.

**5.2 Air-sea interaction within the Mediterranean basin**

Local atmospheric mechanical forcing is primarily exerted through local pressure anomalies, associated with the so-called Inverse Barometer Effect (IBE), and wind anomalies, the latter exerting a dominant effect on Mediterranean coastal sea levels especially by inducing a barotropic oceanic response (e.g., Calafat et al., 2012; Jordà et al., 2012a, 2012b). The IBE is quantified by the hydrostatic equation in about 1 cm of sea-level rise per 1 hPa of sea-level pressure drop. Calafat et al. (2012) quantify in 25% the IBE contribution to decadal winter sea-level variability in Trieste for the period 1950-2009. The highest IBE contributions to seasonal Venetian sea-level variability over the period 1872-2003 in autumn (about 32%) and winter

(41.5%) estimated by Zanchettin et al. (2009) can be accounted for by the regression model between local sea-level and local sea-level pressure, which could embed also other contributions than IBE alone (e.g., Woodworth et al., 2010).

As far as wind forcing is concerned, the morphology of the Adriatic basin displaying a NW–SE elongation and a shallow northern portion (Figure 1) enhances the effect of the remarkably strong and frequent meridional Sirocco wind: Prevailing Sirocco-favorable conditions are associated to a strengthened northeastward flow over the Central Mediterranean, which then favors the piling of Ionian surface waters toward the northern Adriatic, resulting in an increase of Venetian RSL. This Scirocco situation is illustrated for autumn and summer by maps of seasonal anomalies of near-surface wind and sea-level pressure over the Euro-Mediterranean region obtained by compositing the detrended Venetian VLM-corrected RSL series around years characterized by high and low values (Figure 8a,b). The analysis is performed on data from the version 3 of the NOAA-CIRES-DO Twentieth Century Reanalysis, which contains objectively analyzed 4-dimensional weather maps and their uncertainty from the early 19[th] century to the 21[st] century (Silvinski et al., 2019). For each considered season, the years were identified using the 10[th] and 90[th] percentiles of the seasonal sea-level time series as thresholds. Results indicate that in both autumn and winter relatively high sea level in Venice is associated with anomalously low sea-level pressure over the Mediterranean region, which intensifies over the northern portion of the basin, and with extensively significant north-eastward wind anomalies over the Ionian Sea – corresponding to blowing meridional Sirocco wind - as well as with an eastward wind anomaly over the Strait of Gibraltar. The morphology of the Adriatic basin displaying a NW–SE elongation and a shallow northern portion (Fig. 1) enhances the effect of the Sirocco, as prevailing Sirocco-favorable conditions favors the piling of Ionian surface waters toward the northern Adriatic, resulting in an increase of Venetian RSL. The anomalous wind patterns agree with those identified by Zanchettin et al. (2009) obtained with a different set of data.

Fluctuations of surface heat and freshwater fluxes are important drivers of steric changes, through variations in thermal and haline oceanic properties, as well as of ocean circulation changes by affecting processes such as the intermediate or deep water formation and transformation (e.g., Calafat et al., 2012; Jordà and Gomis, 2013; Cusinato et al., 2018). An example of ocean circulation changes on local sea-level variability are the negative trends observed in the Ionian Sea and south-east of Crete shown in Fig. 7a (Bonaduce et al., 2016).

## 5.3 Linkage with the NAO and other teleconnection patterns

The NAO is the dominant mode of large-scale interannual-to-decadal atmospheric variability over the North Atlantic. It is commonly characterized by a teleconnection between tropospheric pressure anomalies over the subpolar Arctic and over the subtropical North Atlantic (e.g., Marshall et al., 2001). Despite the NAO can be identified throughout the year, its anomalies are more stable in winter. The pressure dipole associated to the NAO is tied to changes in the Euro-Mediterranean weather since it produces a meridional displacement of the primary trajectories of the perturbations originating over the Atlantic and affects the frequency and intensity of blocked regimes in the Euro-Atlantic region. Statistical analysis of atmospheric pressure demonstrates coherent large-scale patterns covering the North Atlantic, Europe and the Mediterranean Sea, which explain significant parts of the atmospheric signal variability at interannual and interdecadal scales, particularly during winter. The

large-scale coherency of the atmospheric pressure fields means that several of the local atmospheric parameters as well as the oceanic circulation driven by this forcing becomes correlated. Such linkage drives coherent sea-level changes within the whole Mediterranean basin and, ultimately, in the Venice Lagoon. Despite being studied so far mainly within the framework of interannual to multidecadal climate variability (e.g., Zanchettin, 2017; Han et al. 2019), the same connections can be relevant for longer-term trends as well, and we therefore include them in this review.

Numerous studies attribute a significant fraction of interannual-to-decadal winter sea-level variability in the Mediterranean Sea to the NAO (e.g., Tsimplis et al., 2006; Gomis et al., 2008; Tsimplis and Shaw, 2008; Calafat et al., 2012; Tsimplis et al., 2013; Martínez-Asensio et al., 2014; Ezer et al., 2016; Rubino et al., 2018). The NAO exerts its influence on Mediterranean sea level in multiple ways. First, the larger-scale circulation changes associated to the NAO contribute to set the wind forcing over and near the Strait of Gibraltar, thereby influencing the Atlantic-Mediterranean water-mass exchange on decadal time scales (Fukumori et al., 2007; Landerer and Volkov, 2013; Volkov et al., 2019). Then, the NAO modulates sea-level pressure, freshwater and buoyancy fluxes and riverine inputs within the Mediterranean Sea and, in particular, the Adriatic Sea (e.g., Zanchettin et al., 2008; Josey et al., 2011; Tsimplis et al., 2013; Criado-Aldeanueva et al., 2014; Cusinato et al., 2018).

More specifically for Venetian sea level, Zanchettin et al. (2009) estimate that about half of the variability of detrended winter Venetian RSL can be explained linearly by the NAO. An updated analysis between seasonal time series of NAO (Jones et al., 1997) and detrended (second order polynomial fit) raw Venetian RSL for the period 1872-2019 confirms large values of the correlation statistics ($r_{JFM}$=-0.68, p~0 accounting for autocorrelation in the series; $r_{OND}$=-0.50, p<0.0001). Results on VLM-corrected RSL confirm the strong connection between NAO and detrended Venetian RSL during the cold semester, particularly in winter.

We update previous results about the statistical connection between NAO and Venetian RSL by performing also a wavelet analysis on the 1872-2019 autumn and winter time series of the NAO index and Venetian sea level. The NAO signal is most stable and yields the clearest teleconnections in winter; the autumn season is considered here due to its importance for the occurrence of surges in Venice (Lionello et al., 2020a,b). The wavelet coherence spectra (Grinsted et al., 2004) in Figure 9 illustrate the link between NAO and Venetian VLM-corrected RSL fluctuations over a broad range of timescales in the decadal to multidecadal range, when both variables are robustly in rough antiphase in both seasons. In autumn inclusion of subsidence in the RSL series disturbs the multidecadal coherent signal (see the red contours in bottom panels of Fig. 8), confirming the importance to account for it in studies on the climatic component of local sea-level variability based on tide-gauge data. Temporal variations in amplitude and phase of wavelet coherence and cross-wavelet spectra have been proposed as evidence of lacking robust connection between geophysical series (e.g., Zanchettin et al., 2016; Peicuch et al., 2019). For instance, both characteristics are evident in the wavelet coherence spectrum of the autumn series at the interannual time scales, posing caveats on the truthfulness of the connection. Similarly, in winter coherent interannual variations in anti-phase is only observed intermittently.

We therefore complement the statistical connection between NAO and Venetian sea-level time series with maps of seasonal anomalies of near-surface wind and sea-level pressure over the Euro-Mediterranean region obtained by compositing the data

around years characterized by positive and negative values of the NAO index (Fig. 8c,d). As for compositing around Venetian sea level, for each season, the years were identified using the 10[th] and 90[th] percentiles of the seasonal time series of the NAO as thresholds. To ease the comparison with the anomalous patterns around sea level, the maps show differences between values under negative NAO minus values under positive NAO. In both autumn and winter, the patterns of sea-level pressure and wind anomalies under different NAO states superpose well on those that correspond to variations in Venetian sea level, including large-scale low pressures enhanced over the northern Mediterranean Sea and Sirocco-like northeastward wind anomalies over the Ionian Sea. However, amplitude and spatial extent of the significant anomalies in the atmospheric forcing fields are larger in winter compared to autumn, confirming the weak imprint of the NAO on Venetian sea level in the latter season.

Overall, the seasonal difference in the strength of the NAO imprint on Venetian sea level confirms previous results (Zanchettin et al., 2009) and supports the investigation of other large-scale precursors of atmospheric forcing of Venetian sea level in autumn. For instance, Zanchettin et a. (2009) identify the SCA and EAWR to be related with the interannual-to-decadal variability of meridional atmospheric flow over the Adriatic Sea in autumn (Zanchettin et al., 2009).

## 6 Projections

Estimates of the future RSL change at Venice require that all the different components described in the previous sections are considered over the coming decades and combined (Nicholls et al., 2021). It is useful to distinguish GMSL changes which can be derived from the SROCC ("Special Report on Ocean and Cryosphere in a changing climate", Oppenheimer et al., 2019), regional deviations from GMSL in the Mediterranean and Northern Adriatic, and local vertical land movement contributions.

### 6.1 Vertical land movements

Projections of future contribution of vertical land motion are available only for the GIA (see section 6.2 for a quantification). For the other components of vertical land movements, it is only possible to consider their historical variations to contemplate their potential to affect future RSL changes. Estimates of subsidence at sub-regional scale can be constrained based on observations of past evolution. The sum of sediment compaction, tectonics and GIA is estimated to be about 1.0 mm/year (Antonioli et al., 2017; Tosi et al., 2013) with a constant rate on centennial time scale. At a local scale, subsidence trends of a few cm/year are still observed in restricted areas such as at the lagoon inlets interested in the construction of mobile barriers against high tides (Tosi et al., 2018). Controls on groundwater extraction should prevent a return to the large subsidence rates that occurred between 1930 and 1970, but shallow natural processes, notably consolidation, and anthropogenic activities are observed to continue contributing at rates spatially varying from 10 to -2 mm/year (Tosi et al., 2013). Hence, human activities can contribute significantly to RSL rise, but detailed scenarios are currently not available in the literature and this is a generic problem (Nicholls et al., 2014). All these natural and anthropogenic contributions have the potential to increase RSL in Venice, exacerbating the negative impact of local sea-level rise, with contribution of order of ten centimeters per decade at sub-regional scale, and potentially much larger locally.

## 6.2 Sea-level projections for the Northern Adriatic and Venice

Sea-level anomalies linked to GMSL rise propagate from the mid-latitudes of the eastern North Atlantic to the Venice Lagoon through the Gibraltar and Otranto straits. Sea-level variations along this path is further determined by the mechanical action of the atmosphere and by steric effects associated with changes of temperature and salinity of the water masses. Therefore, beyond local effects on RSL, the projection of sea-level evolution in the Venice Lagoon depends crucially on processes acting on GMSL, sea-level variations across the World Ocean and regional patterns of sea-level change inside the Mediterranean basin.

The recent SROCC report summarizes projected GMSL rise estimates suggesting a likely range from +29 cm to +110 cm for the year 2100 with respect to the 1986-2005 average depending on the future emission scenarios, with the low RCP2.6 and the high RCP8.5 providing the lower and the upper limit, respectively. Here "likely" corresponds to the IPCC uncertainty language, meaning that the probability of future sea-level change within this range is estimated from ≥66% to 100%, and therefore does not exclude values outside this range. The contribution of ice sheets, especially Antarctica, and the underlying dynamical processes that have been intensively debated recently (e.g., Kopp et a., 2014; DeConto and Pollard, 2016; Edwards et al., 2019), provide the main source of uncertainty in future sea-level change projections (Bakker et al., 2017; Oppenheimer et al., 2019). The high uncertainty and strong scientific debate on the contribution of Antarctic ice-sheet melting to the future rate of sea-level rise generates the so-called 'deep uncertainty' (Lempert, 2019), i.e., a condition where experts lack sufficient knowledge or parties to a decision cannot agree upon the system processes and futures (see also Haasnoot et al., 2020). While high-end GMSL rises are unlikely, they cannot be excluded (e.g., Nicholls et al., 2014; Kopp et al., 2017) and are particularly important for decision-making applications and adaptation planning, especially in decision contexts with low tolerance to uncertainty (Hinkel et al., 2019). According to Slangen et al. (2017), the sea-level rise at the subtropical and mid-latitudes of the eastern North Atlantic (and the Mediterranean Sea itself) will only have small deviations (less than 10%) from the global-mean change.

Analysis of the possible deviations of future mean sea level in the Mediterranean basin from the GMSL has been attempted using dynamical and statistical models. Dynamical models of the circulation inside the Mediterranean Sea allow direct estimates of the mechanical effect of the atmosphere on the circulation (wind stress and IBE). There is consensus that this is a small contribution with changes generally less than 10 cm (Tsimplis and Shaw, 2008; Tsimplis et al., 2008; Jordà et al., 2012b; Jordà, 2014; Adloff et al., 2018). The steric effects are computed from temperature and salinity changes using a diagnostic offline computation. This computation obviously depends on the water depth and tends to zero at the coastline. Therefore, for producing a realistic estimate of the actual sea-level change, they should be integrated with the computation of the associated water mass redistribution. In fact, the spatial variations of the resulting sea-level change follow the bathymetry of the Mediterranean Sea and are rather small (< 5 cm) over shallow water areas such as the northern Adriatic Sea (e.g., Tsimplis et al., 2008). The overall steric sea-level change is the consequence of two contrasting effects: thermosteric expansion (associated to warming of water masses) and halosteric contraction (associated to increasing salinity of water masses). Most studies agree

that the former is larger, and there is also a freshening effect of the Atlantic inflow across the Strait of Gibraltar, whose

magnitude is poorly constrained. Several studies contributed to an assessment of the overall steric effects at the Mediterranean scale, with differences depending on periods, models, scenarios and the representation of water exchange between Mediterranean Sea and Atlantic Ocean (Adloff et al., 2018). There are various basin-wide Mediterranean steric sea-level projections. For instance, a range of 2 to 7 cm for the mid-century sea-level anomaly (2050 with respect to 2001) under the SRES-A1B scenario was reported (Carillo et al., 2012). By the end of the 21$^{st}$ century (i.e. 2070-2099), for the steric component

alone, Tsimplis et al. (2008) found a 13 cm sea-level anomaly (with respect to 1960-1990) under the A2 scenario, while Adloff et al. (2015) reported sea-level anomalies over the same periods that range between 34 cm and 49 cm considering a 6-member ensemble and the A1, A1B and B1 scenarios (see Fig. 7b). The magnitude of the steric sea-level rise for the Mediterranean Sea is similar to estimates of the steric GMSL rise, e.g., 29-45 cm (67% confidence interval) for the RCP8.5 scenario between 2100 and 2000 by Kopp et al. (2014).

Estimates of future sea-level rise in the Northern Adriatic Sea from a statistical model are provided by Scarascia and Lionello (2013). The computation is based on a linear regression linking the deviation of the sub-regional sea level to changes of mean sea-level pressure, water temperature and salinity, meant to represent the mechanical atmospheric forcing, steric effects and the redistribution of mass inside the basin (Jordà and Gomis, 2013). Scarascia and Lionello (2013) concluded that regional effects, at the end of the 21$^{st}$ century for the A1B scenario, result in a deviation in the range from 2 to 14 cm from the sea level

of the Eastern Atlantic outside the Strait of Gibraltar; they also concluded that the main contribution to local sea-level rise is caused by remote effects, such as mass inflow across the Strait of Gibraltar, which was conservatively estimated without considering a likely future acceleration and information from climate projections.

Adopting the "mid-range" and "high-end" climate change scenarios defined by Spada et al. (2013), Galassi and Spada (2014) have evaluated the effect of contemporary and future terrestrial ice melting on future regional RSL evolution across the

Mediterranean Sea by solving the Sea Level Equation including the contribution of glaciers, ice caps and of the Greenland and Antarctic ice sheets. They found that terrestrial ice melting will be responsible for a sea-level rise in the Northern Adriatic of ~8 and ~17 cm to 2040–2050 relative to 1990–2000, in the two considered scenarios, respectively. Since the sources of terrestrial ice melting are mostly located in the far-field of the Mediterranean Sea, these contributions shall be largely uniform across the whole Adriatic Sea.

Northern Adriatic RSL projections informed by climate projections can be obtained by summing up the future regional contributions of sterodynamic effects - which corresponds to changes in ocean density and circulation corrected from the IBE -, melting of mountain glaciers and ice sheets, land water and vertical land motions (e.g., Slangen et al., 2012, 2014; Kopp et al., 2014; Gregory et al., 2019). Kopp et al. (2014) provide probabilistic sea-level projections for Venice as part of a global set of local sea-level projections for three different representative concentration pathways. Their projections build on a

combination of expert community assessment (the IPCC-AR5), expert elicitation (e.g., Bamber and Aspinall, 2013), and process modelling (e.g., the 5th phase of the Coupled Model Intercomparison Project or CMIP5) for most sea-level contributors. The "background non-climatic local sea-level change" corresponding to GIA, tectonics, and other non-climatic

local effects was derived by applying a Gaussian process model to tide gauge records. This background linear estimate and its uncertainty (0.72±0.33 mm/year for Venice, see supplementary material, Table 8 in Kopp et al., 2014) is then included in the projections, together with the other components. The 5th-95th percentile range of the resulting sea-level change projections at year 2100 is 29-79 cm for RCP2.6 and 41-107 cm for RCP8.5.

However, two critical aspects are needed to obtain reliable sea-level projections for Venice. First, a full characterization of the vertical land motions in Venice requires an explicit local focus beyond what is provided by a mere extrapolation on the site from regional patterns of GIA, tectonics, and other non-climatic contribution alone. In fact, the subsidence estimate by Kopp et al. (2014) lies in the lower range of the available instrumental estimates, especially as far as the most recent period is considered (see Table 4). Then, the sterodynamic component is derived from the outputs of the coupled climate-model simulations performed within CMIP5. The rather coarse resolution of coupled climate models prevents an accurate representation of small-scale processes (e.g., water exchange at Gibraltar, see: Parras-Berocal et al., 2020), which in turn affects regional sea-level estimates (Marcos and Tsimplis, 2008; Slangen et al., 2017). Another important caveat on multi-model assessments is their reliability on coastal regions where the contributing models may differently resolve the coastline and bathymetry peculiarities, thus yielding local anomalies in the gridded multi-model output that may reflect a bias originated by heterogeneous spatial resolutions across models rather than a physical process (e.g., Landerer et al., 2014).

On this premise, we propose probabilistic projections of Northern Adriatic RSL for two climate scenarios (RCP2.6 and RCP8.5) and one high-end scenario following Meyssignac et al. (2017) and Thiéblemont et al. (2019). The method allows to inflate the uncertainty in projections of the stereodynamics component by accounting for the low confidence in projections of coastal sea-level rise obtained from the limited number of global circulation models participating in CMIP5 and covering the Mediterranean Sea (see Figure 2 in Thièblemont et al., 2019). Specifically, the Mediterranean sterodynamic sea-level projections are estimated by relying on those of the Atlantic area near Gibraltar. Other mass contributions to sea level (i.e., glaciers, ice sheets, land water) have a global effect due to the addition of water mass to the ocean (barystatic sea-level rise) and a regional effect through instantaneous changes in the Earth gravity, Earth rotation and solid-Earth deformation (GRD-induced RSL change). Both contributions are combined into a geographical pattern called barystatic-GRD fingerprint (Mitrovica et al., 2009; Gregory et al., 2019) and are proportional to the land water mass change. For each scenario, uncertainties correspond to the combined uncertainty of each sea-level component calculated as the square root of the sum of the squares of each component uncertainty. Note, however, that contributions that correlate with global-mean air temperature, namely the sterodynamic and ice-sheet surface mass balance components, have correlated uncertainties and are therefore added linearly according to the following equation: $\sigma_{tot}^2 = (\sigma_{sterodynamic} + \sigma_{smb-a} + \sigma_{smb-g})^2 + \sigma_{Glac}^2 + \sigma_{LW}^2 + \sigma_{dyn-a}^2 + \sigma_{dyn-g}^2$ where *smb* stands for Surface Mass Balance, *Glac* for Glaciers, *LW* for Landwater and *dyn-a/dyn-g* for Dynamic Antarctic and Greenland (see Church et al., 2013, for more details). Considering an annual subsidence rate of 1 mm/year, Northern Adriatic RSL is projected to rise by 47 centimeters (likely range 32-62 cm) for the RCP2.6 scenario and by 81 centimeters (likely range 58 cm-110 cm) for the RCP8.5 scenario by the end of the 21st century with respect to the reference period 1986-2005. These projections start

to diverge after 2050 and, excluding the contribution of subsidence of 10.5 cm, yield a total range of local sea-level rise (VLM-corrected RSL) by 2100 between 21 and 100 centimeters, i.e., ~10% lower than GMSL rise (Figure 7c). The high-end scenario – obtained by selecting, for each sea-level component, the highest physical-based estimate found in the literature (see Thiéblemont et al., 2019 for details) – shows that, by 2100, Northern Adriatic sea level could unlikely but possibly rise by
more than 1.8 m. Note that for this high-end scenario, the Antarctic component contributes nearly half of the sea-level change in 2100. This estimate agrees with the analysis by Scarascia and Lionello (2013). Our projections largely overlap with those provided by Kopp et al. (2014) for both scenarios, indicating that they are broadly consistent with each other. Differences are likely due to differences in the methods, models and assumptions employed by both studies, which requires a dedicated investigation to be fully understood.

In conclusion, while our understanding of past RSL change in Venice has improved, the large uncertainty in the magnitude of future GMSL rise remains a major scientific challenge (Oppenheimer et al., 2019). Hence, public policy needs to recognize this uncertainty and monitor sea-level change as part of the management of this uncertainty, including drawing on relevant experience from elsewhere (e.g., Ranger et al., 2013; Environmental Agency, 2021). Regional effects can determine differences in the order of 10 cm between the mean Mediterranean sea level and the GMSL and within different parts of the
Mediterranean basin itself as seen in the historical record. Changes over interdecadal periods can also distort the detection of forced trends over rather long periods of time (e.g., Jordà, 2014). However, scientific literature provides no evidence for a future deviation, on a centennial timescale, of the local sea level at the Venetian coastline from the GMSL that is larger than the abovementioned order of magnitude. RSL can differ by more as land movements and regional atmospheric patterns could provide additional and important contributions.

**7 Gaps of knowledge and opportunities for progress**

This literature review identified several issues where progress is needed, including: (1) improving satellite observations of sea-level change and integrating them with tide-gauge measurements, (2) improving monitoring and prediction of vertical land motions, (3) improving the simulation of Mediterranean oceanic circulation, (4) determining the shape of the historical RSL and VLM-corrected RSL trends, and (5) reducing uncertainty on estimates of regional effects on future climate-induced sea-
level rise.

Concerning observations, satellite GSL data measure the open sea and therefore do not capture coastal variations. In addition, they are insensitive to the vertical land-movement component of RSL. Altimeter data are nonetheless fundamental for providing a regional perspective and reaching robust conclusions on observed local sea-level rise, as they provide an independent source of information from the local tide-gauge data. The contribution of coastal altimetry is considered essential
to within 0–10 km to link the sea-level changes derived from satellites with those measured at tide-gauge locations (Ponte et al., 2019). The extension of the satellite-based sea-level record toward the coast with measurement quality comparable to the open ocean allows detection of differences in the rate of coastal sea-level rise from the open ocean (e.g., Marti et al., 2019)

and their implications for coastal hazards (Ablain et al., 2016; Benveniste et al., 2019). It has been shown that by improving processing, it is possible to make more accurate sea-level measurements in coastal zones (Cipollini et al., 2017). Progress has

840 been made in fitting the radar signal (the so-called re-tracking) to extract a robust estimate of the distance between satellite and sea surface. The ALES retracker, with a proper threshold on error, recovers significantly more data in the 10 km near the coast (Passaro et al., 2014). There are several satellite radar altimetry operational products (along-track and gridded) dedicated to the monitoring of open-ocean sea level, whose quality is constantly improving. Various experimental coastal altimetry products are also now available and validated in some regions, allowing sea-level research in the coastal zone (Gómez-Enri et

al., 2019). An updated table is accessible at www.coastalt.eu/datasets. Importantly, merging altimeter data from different missions requires homogenous re-processing and minimization of drifts and systematic biases between missions. Overall, progress is expected in terms of improving the altimetry estimates closer to the coast, perhaps with different instruments, and understanding the relationship between the coastal and close-to-the coast sea-level changes in a complex environment (De Biasio et al., 2020).

For the Northern Adriatic, an opportunity for progress is provided by the ESA SLCCI extension (CCI+), which will process along-track data from additional satellite missions using re-tracked data, dedicated coastal geophysical corrections, and improved editing that will then be combined in a global grid with higher resolution near the coast (Anny Cazenave, personal communication). Also, the novel GNSS-derived Path Delay Plus (GPD+) correction now provides accurate wet tropospheric delays (Fernandes and Lázaro, 2016). The Wide-swath interferometry will be deployed for the first time on the Surface Water

Ocean Topography mission to provide for the first time sea-level imaging that will solve the limitation of existing satellite sea-level measurements being available only along tracks (Vignudelli et al., 2019a).

Concerning the monitoring of vertical land motions, integrated systems have been shown to offer the best approach to the study of subsidence (Wöppelmann and Marcos, 2016; Zerbini et al., 2017; Tosi et al., 2009): GNSS provides point-wise continuous positioning with respect to a global reference frame; SAR offers spatially dense measurements of surface

displacements relative to a ground target selected as reference point. However, while these techniques can support the investigation of present subsidence patterns with unprecedented detail (i.e., at the scale of individual buildings), future scenarios remain difficult to construct, with the anthropogenic component of vertical land movements being the most difficult to assess (Nicholls et al., 2021). Historical observations showing the potential of anthropogenic subsidence in Venice to be of order of tens of cm per decade demonstrate the need of continued careful regulation of land and groundwater use, and

monitoring of local subsidence. Together with subsidence process understanding and simulation, this might be used to develop high and low subsidence scenarios, respectively.

Concerning the simulation of Mediterranean oceanic circulation, considerable efforts have been invested over recent years into developing and applying regional climate and ocean circulation models approaching the issue of dynamical downscaling from different perspectives (e.g., Somot et al., 2008; Sannino et al., 2009; Artale et al., 2010; Naranjo et al., 2014; Sein et al., 2015;

Turuncoglu and Sannino, 2017; Androsov et al., 2019; Palma et al., 2020). This included coordinated international regional climate modeling activities (e.g., MedCORDEX) and regional reanalysis for the Mediterranean Sea obtained through

numerical simulations with data assimilation (Simoncelli et al., 2014). However, despite recent progress in the representation of lateral boundary forcing at the Strait of Gibraltar, there are several aspects that remain poorly understood or worth deeper investigation. For example, small changes in the salinity difference between Mediterranean and Atlantic waters around a threshold of 2 psu can determine shifts in the simulated hydraulic regime within the Strait of Gibraltar, from sub-maximal to maximal (e.g., Artale et al., 2006). Accordingly, a scenario involving a positive trend in the salinity difference can result in a partial isolation of the Mediterranean Sea from the rest of the World Ocean (Tsimplis and Baker, 2000). How non-linear interaction between large-scale ocean variations and local strait phenomena may sustain an abrupt change in the salt/freshwater transport between the Mediterranean and the Atlantic, and a shift in the Mediterranean mean circulation, remains to be investigated within a comprehensive modelling framework. It is also possible that the large-scale ocean circulation in the North Atlantic can impact the Mediterranean sea level through ocean–atmosphere feedbacks (e.g., Marshall et al. 2001; Volkov et al., 2019), but this requires further investigation (Piecuch et al., 2019). Other exchanges are also relevant: for instance, water-mass exchanges with the Black Sea through the Turkish Straits remain idealized in current simulations, and their effect underrepresented in future projections. Water mass exchange between shelf and ocean is performed through cascading processes, which are hardly reproduced by both regional and global solutions (e.g., Polyakov et al., 2012; Holt et al., 2017). In this context, the unstructured approach adopted by Ferrarin et al. (2018) is promising where the system of inter-connected basins formed by the Mediterranean, the Marmara, the Black and the Azov seas was numerically investigated using a unique computational mesh allowing for a seamless transition between different spatial scales, from narrow straits to open sea (see also Umgiesser et al., 2020, in this special issue).

Poorly simulated internal ocean variability also provides potential weakness to projected circulation changes in the Mediterranean Sea, which calls for a stronger focus on the validation of regional ocean models regarding interior and abyssal dynamics linked to fundamental oceanographic processes. The Adriatic Sea is the only Mediterranean sub-basin in which the evaporation-precipitation-runoff budget is negative: the buoyancy flow at the Otranto Strait is either positive or negative depending on the predominance of production of dense water within the Adriatic or of the inflow of the Levantine Intermediate Water, respectively. Numerical simulations indicate that a nonlinear convection-mixing feedback can favor hysteresis in the Adriatic Sea with multiple equilibria encompassing estuarine and anti-estuarine circulation (Pisacane et al., 2006; Amitai et al., 2017). Such behavior could have important implications for future sea-level variability in the Venice Lagoon. Overall, even under accurate representation of global steric and mass addition from the Atlantic, projections of Mediterranean sea-level change from current regional ocean models would be reliable only in the basin mean tendencies. Further, comparison of the regional simulations with satellite-derived data highlights local biases in the historical sea-surface height patterns and trends, as well as large inter-model heterogeneity in projected changes at the local scale driven by differences in simulated circulation changes. Simulation-data comparison using updated hydrographic datasets, such as EN4 (Good et al., 2013) or Ishii et al. (2017), could be extremely valuable.

The higher rate of RSL rise observed in recent decades compared to the longer-term estimate (Table 4) raises the practical question of how different statistical models used to extract the trend for time series analysis affect the results. Here, trends

refer to the long-term movement in a time series, which may be regarded, together with the oscillation and random component, as generating the observed values (ISI, 2003). We have evaluated the performance of a linear and a quadratic regression model on the RSL times series, including the raw series and the climatic component alone, and for the periods 1872-2019 and 1993-2019. According to several skill metrics including $R^2$, AIC and FPE (Alessio, 2016, and references therein) (Table 6) the quadratic model only slightly outperforms the linear model for both periods and both series. As far as the higher rates of Venetian RSL rise observed in recent decades are concerned, the simple acceleration expressed statistically in terms of quadratic fitting seems to be insufficient and further methods could be explored (but as a note of caution see, e.g., Chambers, 2015). The presence of substantial variations in multidecadal trends of sea level in Venice, together with methodological differences, explains the wide range of estimates of average sea-level rise obtained by different authors considering different periods. Still, certain aspects of the forcing of Venetian sea-level variability requires improved quantifications and characterization (for instance regarding the atmospheric forcing of local sea-level variability), and this will better constrain the range of projected future changes.

Concerning understanding the future evolution of Venetian RSL, we identify two critical gaps of knowledge. First, the significant interannual to multidecadal variability observed in Venetian sea level requires new studies to address their implications for near- and mid-term sea-level prediction. A possible focus could be, for instance, the impact on Venetian sea level of a multiannual period of positive phase of the NAO as observed in the early 1990s. Second, on the long term, there is no consideration of the implications of high-end GMSL rise for RSL in Venice. Research on both aspects would contribute to improved understanding of forcing mechanisms of Venetian sea-level changes, risk assessment and management, and to more effective scientific communication.

Finally, this literature review builds on recent studies on local and regional sea-level change projections (e.g., Slangen et al., 2012; Kopp et al., 2014; Oppenheimer et al., 2019; Thiéblemont et al., 2019) to develop explicit local RSL change scenarios for Venice by combining the uncertainty ranges of future projections for the individual contributing processes. Understanding the causes of the differences between the projections obtained in this study and those obtained by Kopp et al. (2014) can help clarifying the possible implications in the choice of methods, models and assumptions to build future scenarios of Venetian sea level and by implication for any site. Further approaches to objectively combine such uncertain estimates are hoped to be tested, based on qualitative criteria (e.g., considered process, statistical and numerical framework) or quantitative metrics, such as relative or absolute model skills in representing relevant physical features (e.g., boundary forcing at the Strait of Gibraltar). It will be also important to compare our estimates with updated scenarios of Venetian sea-level future change that are expected from the 6[th] phase of the Coupled Model Intercomparison Project (Eyring et al., 2016).

## 8 Conclusions

The City of Venice and the surrounding lagoon ecosystem are critically affected by variations in RSL height driven by a host of diverse processes. These encompass oceanic processes driving sea-level variations from diurnal astronomical oscillations

to climatic interannual-to-multi-centennial fluctuations, and vertical land movements causing RSL variations on time scales ranging from a few decades due to, e.g., anthropic activities, to multi-millennial trends due to tectonic activity. This review has summarized and reassessed recent progress in the estimation, understanding and prediction of the individual contributions to RSL by exploiting new observational datasets, improved statistical methods and more realistic numerical simulations of ocean and Earth system components achieved in the past decade.

Our estimate of the historical long-term linear trend of Venetian VLM-corrected RSL is 1.23±0.13 mm/year (from 1872 to 2019). Inclusion of subsidence gives a centennial RSL rise of about 2.5 mm/year. Looking to the future, the effects of both subsidence and climate-induced sea-level rise will have profound implications for Venice. By 2100, natural local subsidence is expected to result in a RSL rise of about 10 cm relative to the late 20$^{th}$ century. Projected climatically-induced Venetian sea-level rise from estimates for the GMSL corrected for the regional redistribution of mass contribution components such as glacier, ice-sheet and groundwater is in the range from 21 to 52 (from 48 to 100) centimeters by 2100 for the RCP2.6 (RCP8.5) socio-economic scenario. Ice-sheet melting provides a highly uncertain contribution. A RSL rise in Venice exceeding 180 cm by 2100 is an unlikely but plausible high-end change under strong melting of Greenland and Antarctica. These estimates neglect the effect of atmospheric forcing of local sea level, which potentially contributes additional uncertainty, estimated here in the range of about ten centimeters. The uncertainty range of the RSL rise by 2100 in Venice across all emission scenarios obtained here is thus very large: a minimum low-end scenario is about 10 centimeters, corresponding to unlikely combinations such as either low emission scenario with no subsidence and regional atmospheric interaction producing a negative effect or subsidence continuing at an historical rate and all other effects compensating each other; the upper limit of the likely range considering RCP8.5 projections is about 120 centimeters. Including unlikely high-end scenarios would rise this limit to 180 centimeters, which could approach two meters under adverse regional atmospheric forcing. An additional contribution could be produced by anthropogenically-driven subsidence. While, in general, the resulting effect of regional climatic processes could either attenuate or increase regional RSL with respect to GMSL, local subsidence will necessarily exacerbate it. Further, because of the differential rates of subsidence observed across the Venice area, the land movement estimates at the tide gauges may underestimate the risk for other parts of the city.

Among the important advances highlighted in the review are: Centennial RSL variations are now known to be spatially heterogeneous within the lagoon and the city due to differential vertical ground movements, hence local trend estimates are not expected to be representative for the city or lagoon as a whole; due to the non-linear variations caused by ground subsidence, a linear detrending of Venetian RSL time series is unsuitable unless data are preliminarily corrected for the effect of vertical land motion; remote climate forcing from the Atlantic sector via atmospheric and oceanic processes critically contributes to interannual-to-multidecadal Venetian RSL variability; Atlantic hydrographic boundary conditions are a major source of uncertainty for future projection of Mediterranean sea levels: uncertainty in water mass flows at the Strait of Gibraltar yields an ensemble spread between simulations comparable to that determined by uncertainty in greenhouse gas emissions.

We confirm the existence of a strong link between interannual and interdecadal variability observed in Venetian sea levels and in the large-scale atmospheric circulation over the North Atlantic during the cold semester, particularly with the North Atlantic Oscillation (about 46% of shared variability for the detrended winter average time series in the period 1872-2019).

The review has highlighted several major gaps of knowledge as well. Among these: altimetry data are recorded rather far from Venice and may not represent the lagoon RSL variability; uncertainties in geologic trends remain difficult to assess; a reliable framework is lacking to combine uncertain future estimates of RSL change due to individual contributions, which provides for a major opportunity for progress to better constrained ranges of future projections; and historical evidence demonstrates that subsidence can be temporarily dominated by the anthropic component. This last point shows the importance of sustaining the management regime that brought this anthropogenic component under control across the lagoon, and possibly strengthen it to bring the small-scale ongoing anthropic subsidence under control. Finally, whereas several studies explored scenarios of RSL changes in Venice at the end of the 21$^{st}$ century under global climate change, near-term and transient predictions have not been attempted yet, and high-end scenarios have not been a subject of explicit focus.

**Acknowledgments**

The authors want to thank the European Space Agency (ESA) that funded the Climate Change Initiative to produce a climate quality record of sea level from satellite altimetry. This work is partially funded by ESA under Phase 2 Bridging Phase to CCI (cont. 4000109872/13/I-NB – Contract Change Notice 6). The Centro Previsioni e Segnalazioni Maree of the Venice Municipality is acknowledged for providing the tide gauge data. Thanks are due to Francesco De Biasio (CNR-ISP) for the joint work during the CCI project. Wavelet coherence spectra are calculated using the cross wavelet and wavelet coherence package by Aslak Grinsted, John Moore and Svetlana Jevrejeva. Linear regressions are performed with the "regress" function of Matlab. The code to compute regional sea-level projections was formerly developed by Gonéri Le Cozannet. 20th Century Reanalysis V3 data provided by the NOAA/OAR/ESRL PSL, Boulder, Colorado, USA, from their Web site at https://psl.noaa.gov/. Scientific activity by DZ, AR, GU and SR performed in the Research Programme Venezia2021, with the contribution of the Provveditorato for the Public Works of Veneto, Trentino Alto Adige and Friuli Venezia Giulia, provided through the concessionary of State Consorzio Venezia Nuova and coordinated by CORILA. GS is funded by a FFABR (Finanziamento delle Attività Base di Ricerca) grant of MIUR (Ministero dell' Istruzione, dell'Università e della Ricerca)

**Author contribution**

DZ and SB coordinated the paper with help from FR. Specific contributions to the sections are as follows (LA = leading author, CA = contributing author). Section 1: LA: DZ; CA: SB. Section 2: LA: FR, SZ; CA: MT, SV, GW, SZ. Section 3: LA: SB; CA: FA, EC, GS, GW, SZ. Section 4: LA: DZ, FR. CA: PL. Section 5: LA: DZ, PL; CA: MT, GS, FA, AA, VF, AR, CF. Section 6: LA: DZ, PL; CA: RT, FR, SB, RJN. Section 7: LA: DZ; CA: SB, PL, RJN, CF. Section 8: LA: DZ, PL, RJN; CA: all authors. SR created figure 1, 8. DZ created figures 2,4,6,7,9. SB created figure 5. SV created figure 3.

**Competing Interest**

The authors declare that they have no conflict of interest.

**Code/Data availability**

Relevant data used in this study will be made available in a public repository upon publication.

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

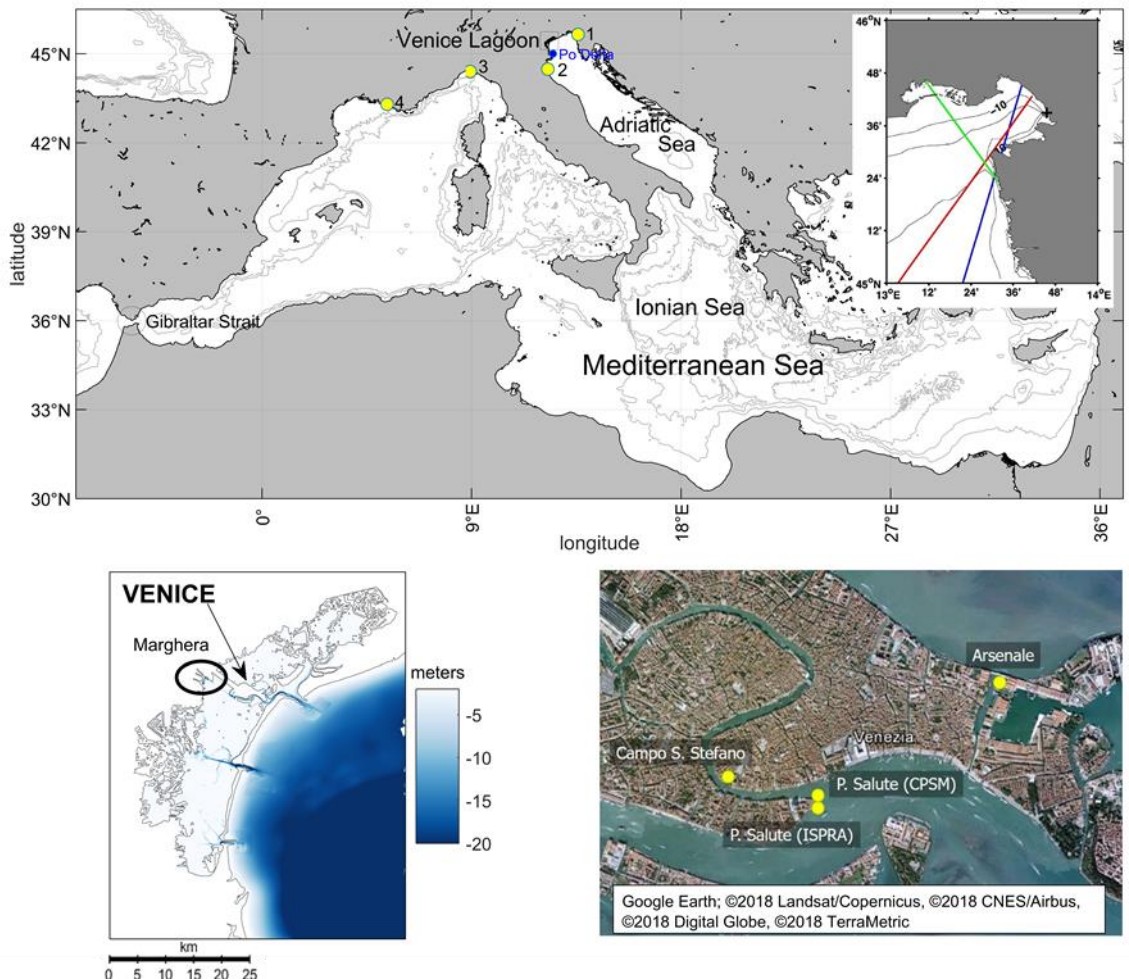

**Figure 1: Maps of the study area and major locations and geographical features mentioned in the paper. Top: the Mediterranean Sea (main panel) and satellite altimetry tracks over the northern Adriatic Sea (blue: Envisat 416; red: Jason1-151; green: Jason2-196) (inset); bottom left: the Venice Lagoon; bottom right: the historical city of Venice. Tide gauge stations are indicated with yellow dots (1-Trieste, 2-Marina di Ravenna, 3-Genoa, 4-Marseille). The map for bottom right panel is extracted from Google Earth; ©2018 Landsat/Copernicus, ©2018 CNES/Airbus, ©2018 Digital Globe, ©2018 TerraMetric.**

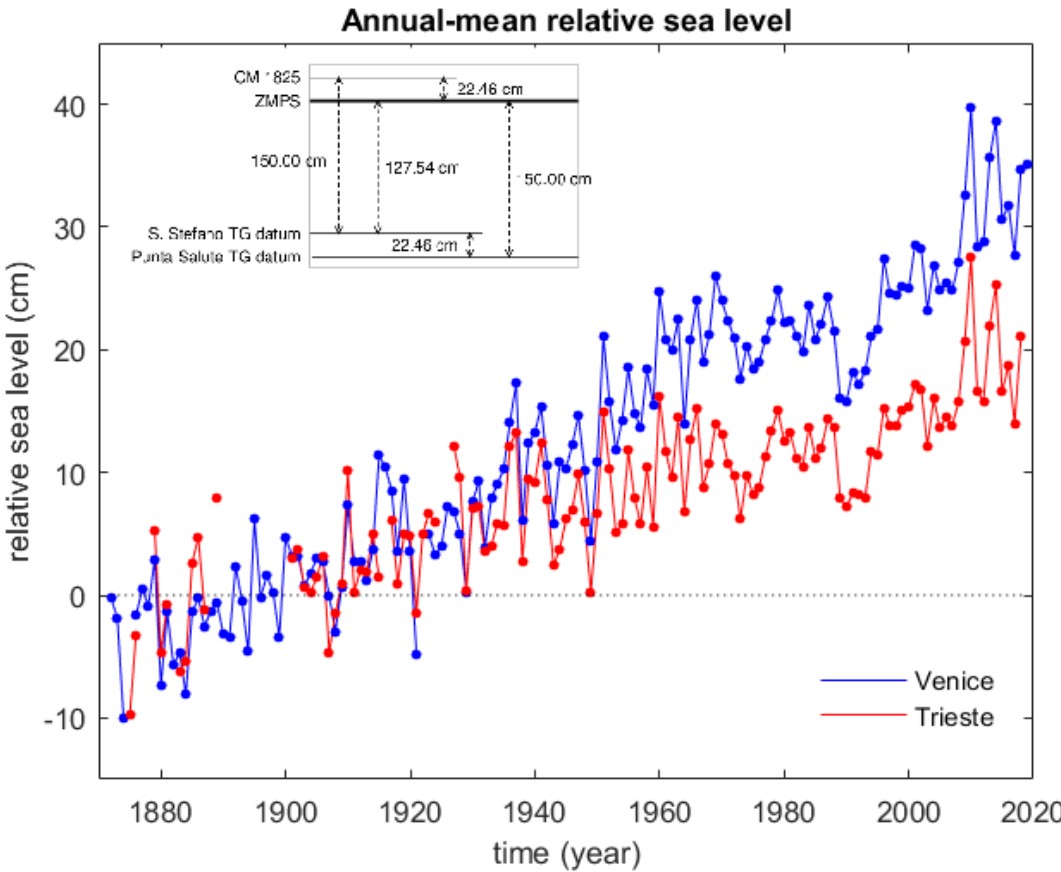

**Figure 2: Time series of annual-mean RSL measured by tide gauges in Venice and Trieste. Venice data are referred to ZMPS, Trieste data are offset for illustrative purposes. The top-left inset defines the reference planes of the tide gauges at Santo Stefano and Punta della Salute (redrawn from Battistin and Canestrelli, 2006).**

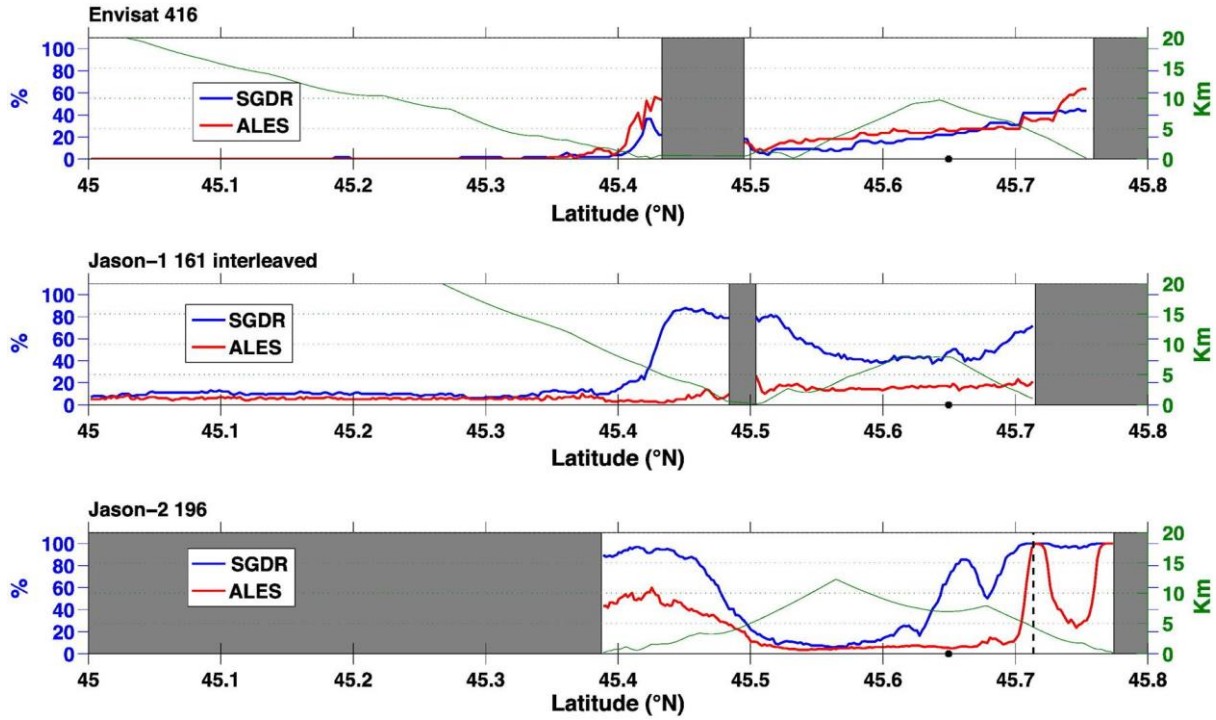

**Figure 3: Percentage of outliers along track for Envisat track 416 (top right panel), Jason-1 track 161 (mid right panel) and Jason-2 track 196 (bottom right panel) from SGDR (blue line) and ALES (red line) products. Land is shaded in grey. The green line represents the distance from the closest coast (adapted from Passaro et al., 2014).**

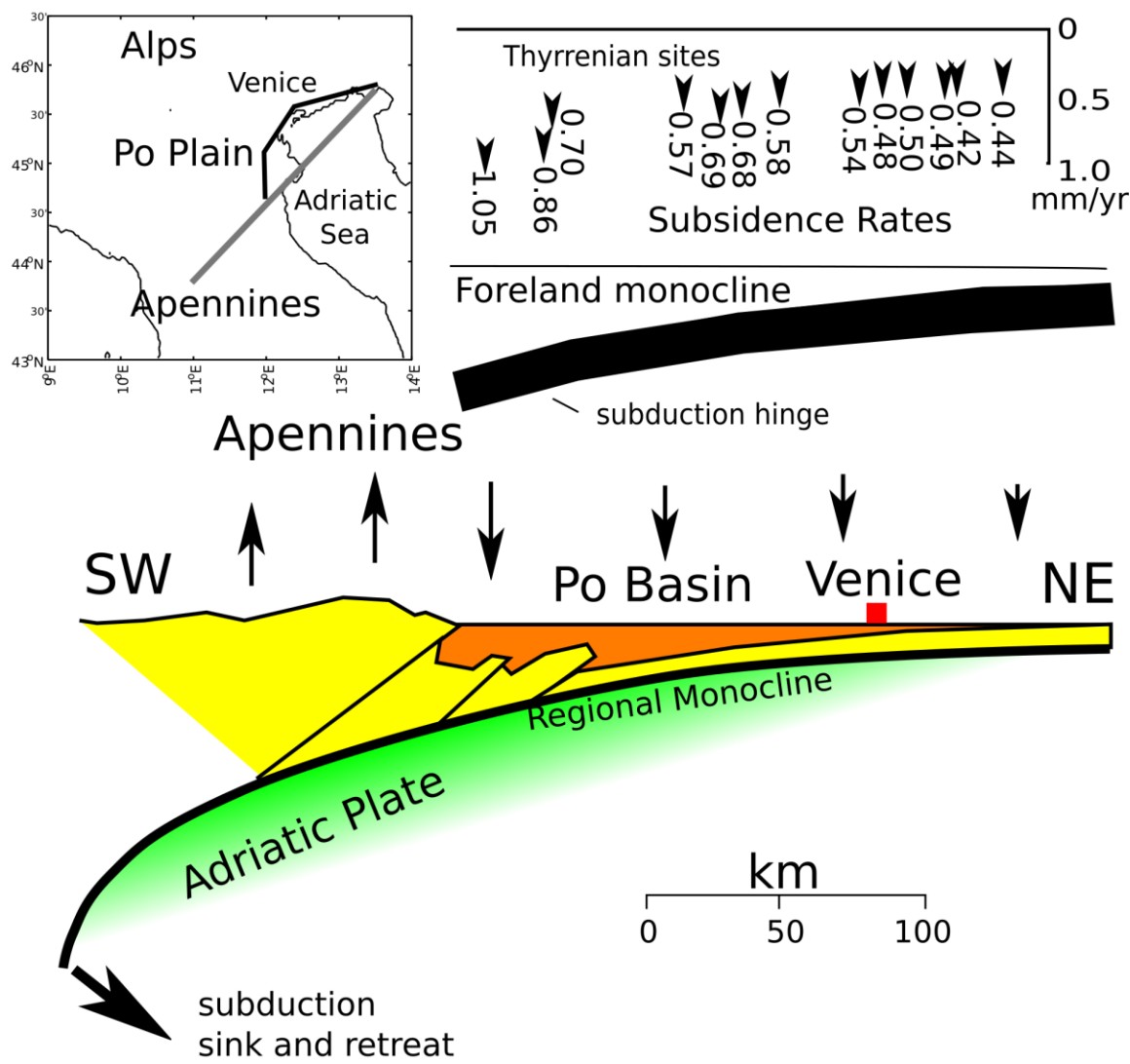

Figure 4: Geological setting of the Po plain area with dominant tectonic features (adapted from Cuffaro et al., 2010).

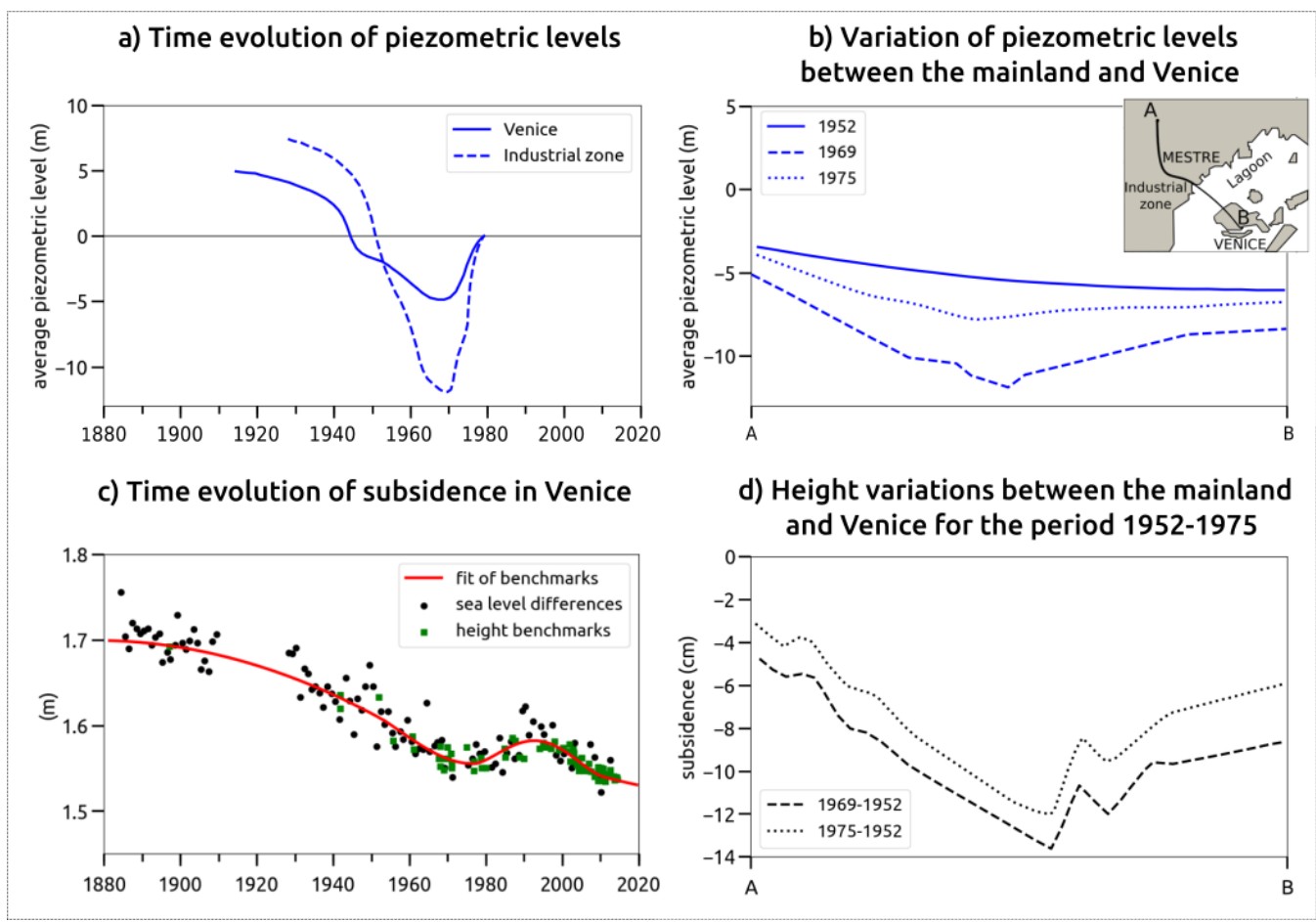

**Figure 5: (a)** Comparison of the piezometric level in the Marghera industrial area and in Venice from 1910 to 1980 (redrawn after Gatto and Carbognin, 1981); **(b)** Average piezometric level between 1952 and 1975 along a levelling line from the mainland to Venice (redrawn after Carbognin et al., 1976). **(c)** Empirical curve (red line) accounting for subsidence in Venice (updated from Zerbini et al., 2017; from 2013 onward, the subsidence trend shown in the figure is the one derived from the GPS data of the station PSAL, see Table 3). Black dots represent the annual sea level difference between Genoa and Venice. Green dots represent the height of various benchmarks; **(d)** same as (b) but for land subsidence. Height differences were estimated by means of 3 levelling campaings performed in 1952, 1969 and 1975, respectively. Note the rebound observed in the early 1970s, in correspondence with the partial recovery of the aquifer.

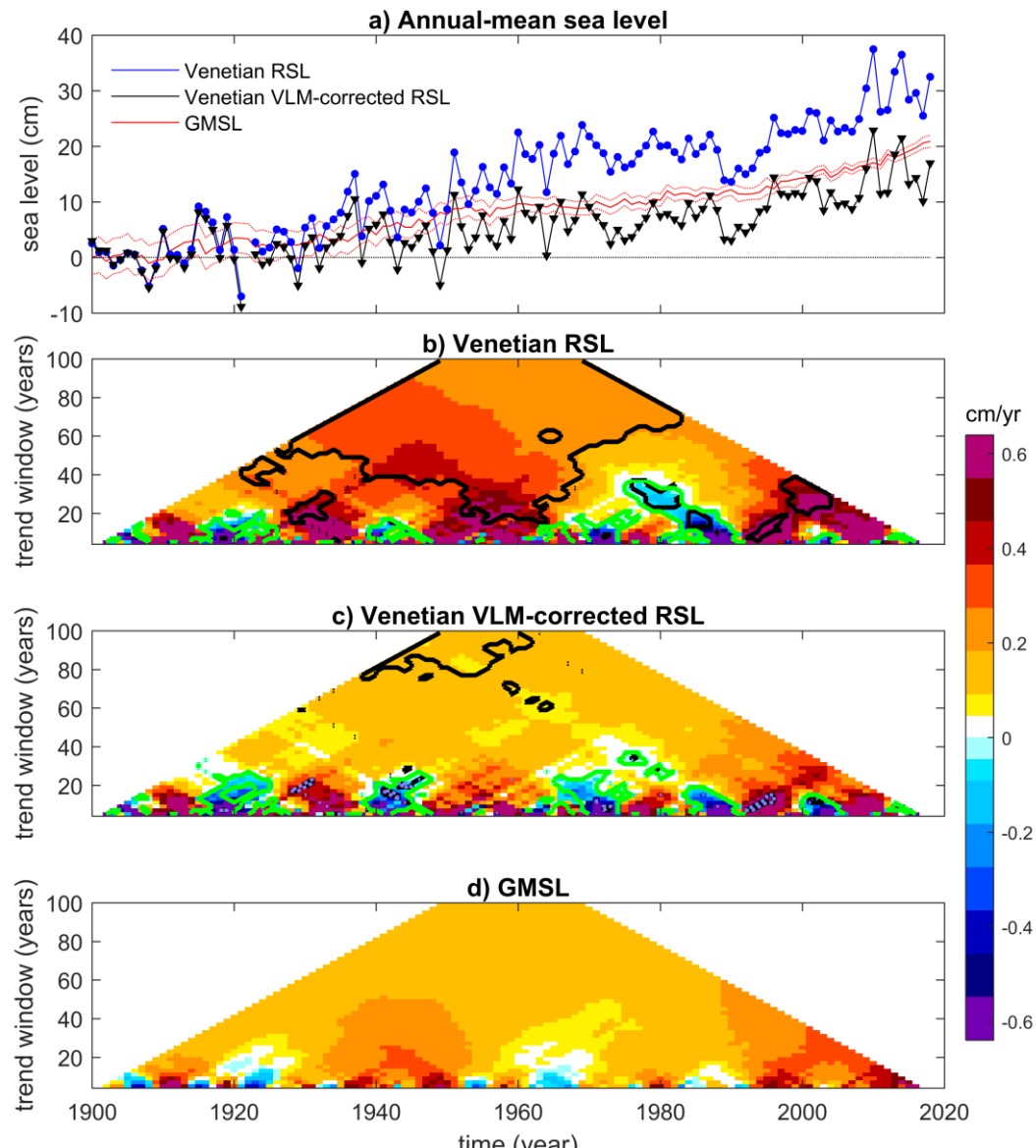

**Figure 6: Comparison between trend estimates of GMSL and Venetian sea levels (RSL and VLM-corrected RSL). (a) Temporal evolution of annual-mean sea levels, including GMSL (from Frederikse et al., 2020; dashed lines are upper and lower estimates) and Venetian RSL and VLM-corrected RSL. All anomalies with respect to the 1900-1910 mean. (b-d) Maps illustrating linear sea-level trends, estimated via linear regression, for running windows of variable width along the observation period. In panels b and c, black contours illustrate where the GMSL and Venetian sea-level trend estimates do not overlap within 95% confidence intervals obtained from the linear regression, while green contours indicate where they differ in the sign.**

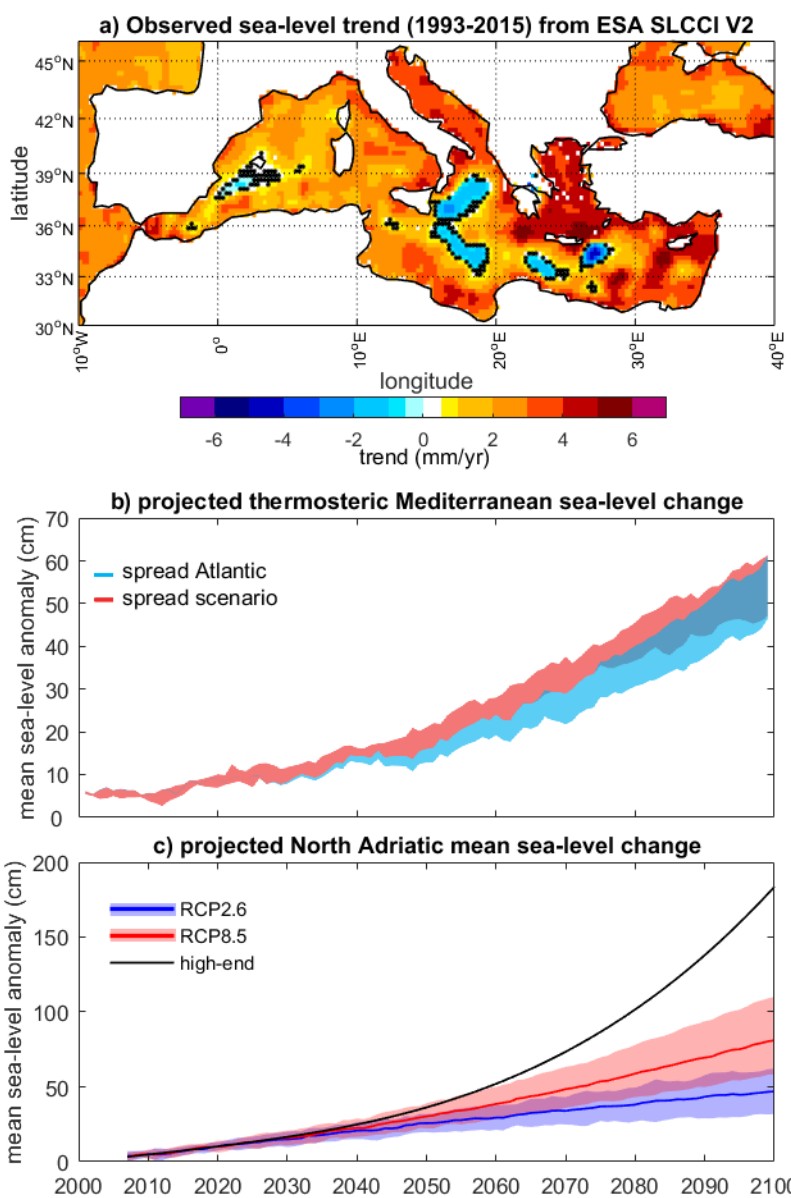

**Figure 7: Observed and projected trends of Mediterranean sea-level variations. (a) Sea-level trends in the Mediterranean Sea obtained using the ESA SLCCI V2 product over the period 1993-2015. Dots indicate grid points where the trend is not different from zero within the associated error range, estimated as the 90% confidence interval from linear least-squares regression analysis. (b) Projected thermosteric basin-average sea-level anomalies for the Mediterranean Sea and associated uncertainties related to the Atlantic hydrographic boundary conditions (blue) and to the socio-economic scenarios based on the Special Report on Emissions Scenarios (red) with the regional ocean model NEMOMED8, for the 2000-2100 period (vs. 1961-1990). Uncertainties are the spread among the simulations with only differing Atlantic boundary conditions (blue) and the spread among the simulations with differing socio-economic scenarios (red). Adapted from Slangen et al. (2016). (c) Projected Northern Adriatic RSL anomalies and associated uncertainties related to socio-economic scenarios (shading: 5-95 percentile range, line: median). See main text for details on the calculation of uncertainties. Adapted from Thiéblemont et al. (2019).**

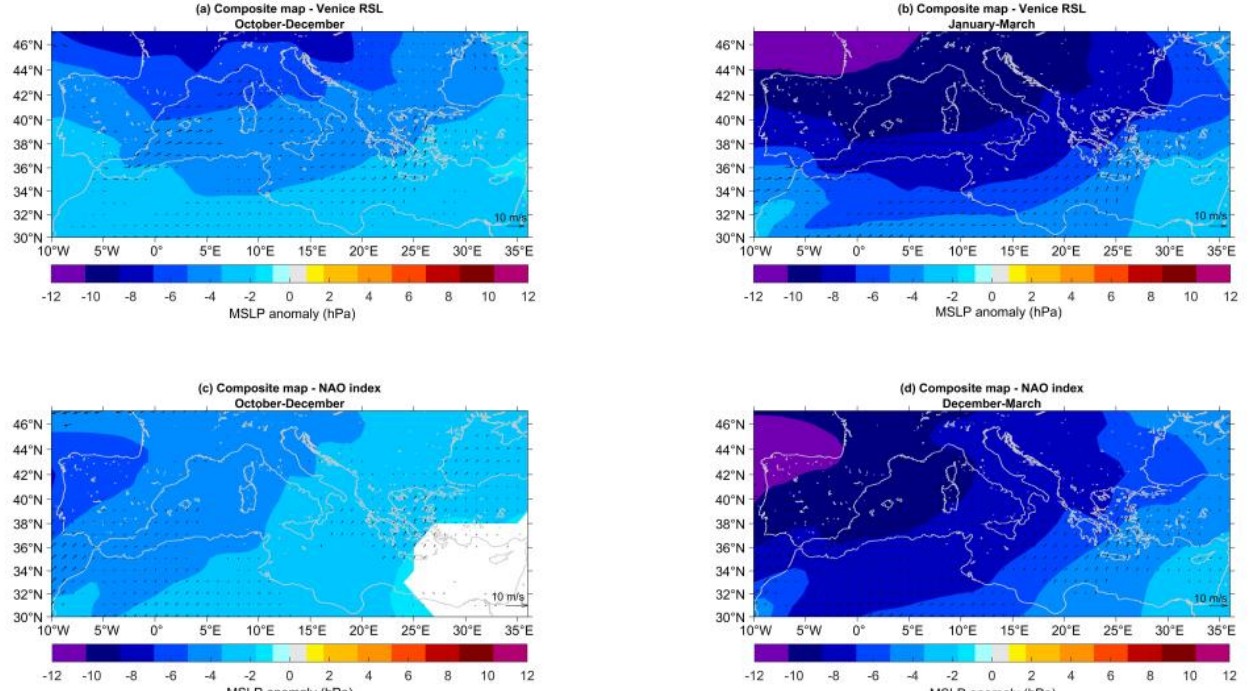

**Figure 8: Composite analyses of autumn (OND, panels a,c) and winter (JFM, panels b,d) reanalysed sea-level pressure (shading) and 10-meter wind (arrows) around Venetian sea-level (a,b) and NAO anomalies (c,d). Composite years are determined based on detrended seasonal values of Venetian VLM-corrected RSL (second order polynomial trend) and of the Jones' NAO index (linear trend) below the 10th percentile and above the 90th percentile. Shown differences are for high minus low sea level (a,b) and for negative minus positive NAO (c,d). Analysis is for the period 1872-2015 and for linearly detrended sea-level pressure and wind data. Only grid points where differences are significant at 90% confidence according to the Wilcoxon rank test are shown.**

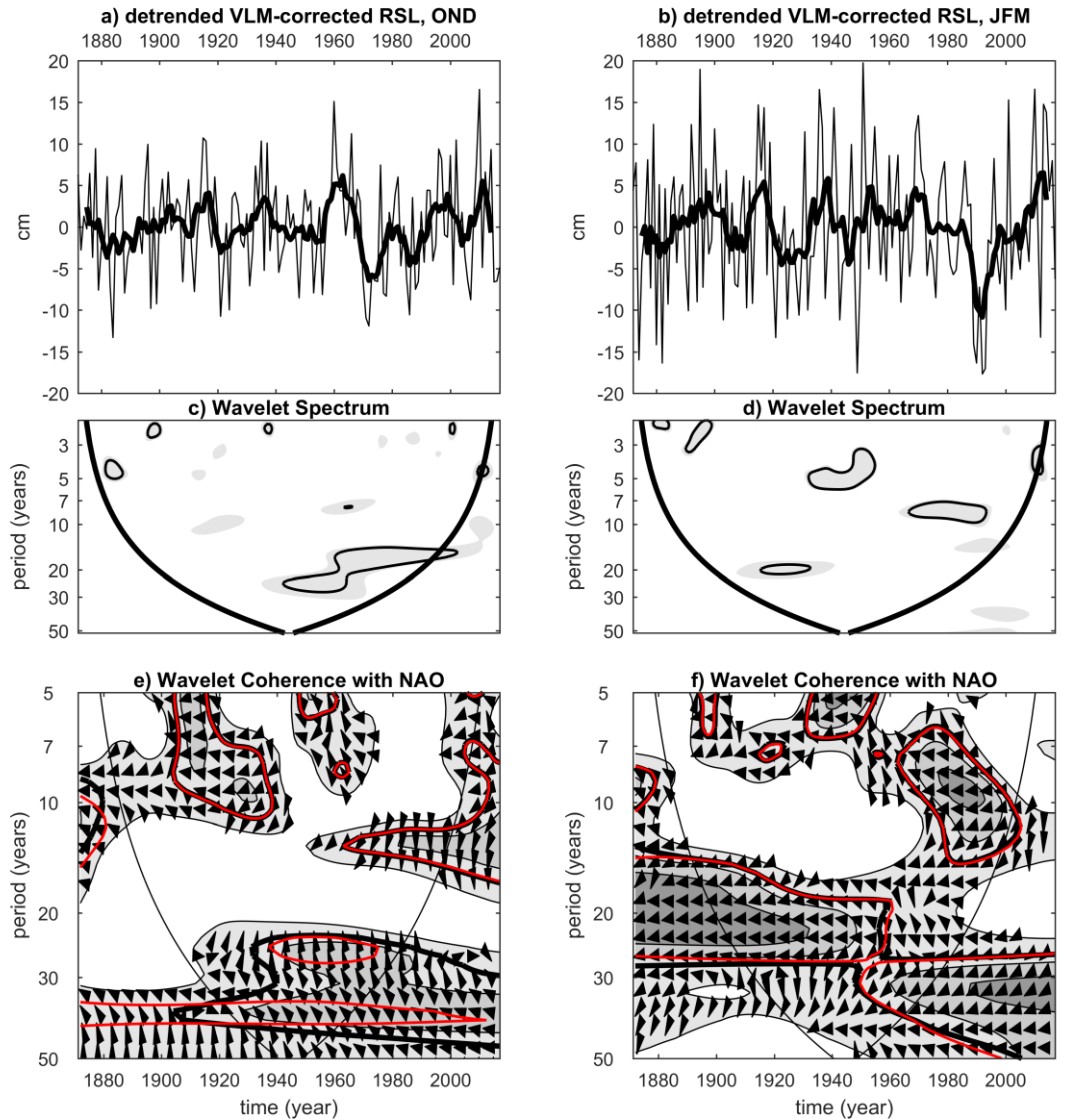

Figure 9: Interannual to interdecadal autumn (left) and winter (right) Venetian sea-level variability since 1872 and its link with the NAO. a,b: detrended (second order polynomial fit) autumn and winter time series of the VLM-corrected RSL from the Punta della Salute gauge record (VLM estimated from Zerbini et al., 2017); c,d: Continuous Wavelet Spectra. Shading (thick black contour) is the portion of the wavelet spectrum exceeding 90% (95%) confidence against red noise (lag-1 autoregressive model) hypothesis (see

Grinsted et al., 2004, for details); black lines: cone of influence where edge effects occur; e,f: wavelet coherence spectra between Punta della Salute data and the Jones NAO index (arrows indicate the phase, with co-phase pointing to the right; thick black contour: 95% confidence, in red for Punta della Salute RSL data without removal of VLM and detrended as in the main analysis; black lines: cone of influence).

**Table 1 - List of acronyms used in the paper.**

| | |
|---|---|
| BP | Before Present |
| O | Order of |
| NAO | North Atlantic Oscillation |
| RSL | Relative Sea Level |
| EAWR | East Atlantic Western Russia pattern |
| SCA | Scandinavian pattern |
| GIA | Glacial Isostatic Adjustment |
| GSL | Geocentric Sea Level |
| CM | Comune Marino/Comune Alta Marea |
| MTL | Mean Tide Level |
| GMSL | Global-Mean Sea Level |
| ZMPS | Zero Mareografico Punta Salute |
| ALES | Adaptive Leading Edge Subwaveform |
| RMS | Root Mean Square |
| RADS | Radar Altimeter Database System |
| ESA - SLCCI | European Space Agency - Sea Level Climate Change Initiative |
| CTOH | Centre of Topography of the Oceans and the Hydrosphere |
| SHYFEM | Shallow water HYdrodynamic Finite Element Model |
| AIC | Akaike Information Criterion |
| FPE | Final Prediction Error |

 **Table 2 Time evolution of the natural component of land subsidence in the Venetian region over geological time scales**

| Period | Subsidence rate [mm/yr] | Data source | Reference(s) |
|---|---|---|---|
| Last 2 Myr | ~0.5 | Nannofossil biostratigraphy, paleomagnetic polarity, magnetic susceptivity and sedimentologic facies of a drilled core | Kent et al., 2002 |
| Last 1.43 Myr | 0.7-1.0 | Thickness of Pleistocene sediments from seismic lines and boreholes | Carminati et al., 2003 |
| Last 125 kyr | 0.58-0.69 | MIS 5.5 paralic deposits in drilled cores | Antonioli et al., 2009 |
| Last 40 kyr | 1.2-1.3 | Radiocarbon dating on organic remains, mainly peats and shells | Bortolami et al., 1985 |
| 4-5 kyr | 1 | Same as previous line | Bortolami et al., 1985 |
| | 1.1±0.3 | Geomorphological and archaeological markers | Antonioli et al., 2009 (From their Table 1: average of H/G values for sites 17, 27 and 30 ). Original data from Antonioli et al. (2009); Lezziero (2002); Serandrei Barbero et al. (2001); |

**Table 3 - Recent evolution of land subsidence in the historical city center of Venice as measured using geodetic techniques. Tosi et al. (2013) point out that the uncertainties associated with their SAR estimates represent the ground motion variability at the city scale and are not related to the measurement accuracy.**

| Period | Subsidence rate [mm/year] | Data source | Reference(s) |
|---|---|---|---|
| At the turn of the 19[th] and 20[th] Century | 0.9 | Long-term interpolation of height benchmarks | Zerbini et al., 2017 |
| 1931-1970 | 2.3 | Difference of tide gauge records (with reference to Trieste) | Carbognin et al., 2004 |
| | 2.3 | Long-term interpolation of height benchmarks | Zerbini et al., 2017 |
| 1953-1973 | 5 | leveling | Gatto and Carbognin, 1981 |
| 1973-1993 | -0.02 (uplift) | leveling | Carbognin et al., 1995a, 1995b |
| 1992-2002 | 0.8±0.7 | SAR | Tosi et al., 2013 |
| 2003-2010 | 1.0±0.7 | SAR | Tosi et al., 2013 |
| 2008-2020 | 1.7±0.5 | GPS station VEN1 (Riva dei Sette Martiri) | daily solutions provided by NGL (Blewitt et al., 2018); velocity estimated assuming a white + power-law noise model of a priori unknown spectral index |

| | | | (CATS Software, Williams, 2008) (consistent with Santamaría-Gómez et al., 2017) |
|---|---|---|---|
| 2014-2020 | 0.9±0.6 | GPS station PSAL (Punta della Salute) | daily solutions provided by NGL (Blewitt et al., 2018); velocity estimated assuming a white + power-law noise model of a priori unknown spectral index (CATS Software, Williams, 2008) |

**Table 4 - Linear trends of Venice RSL from tide gauge measurements estimated by various authors in the last 15 years. Errors are STD (68% confidence) except where noted in the Confidence column. The linear fit of observed sea level is used except where annotated. Estimates are grouped based on the period of analysis: long-term and satellite altimetry period.**

| Period | Source | Trend (mm yr$^{-1}$) | Confidence | Notes |
|---|---|---|---|---|
| *Long-term* | | | | |
| 1909-2000 | Marcos and Tsimplis, (2008) | 2.5±0.1 | | |
| | Wöppelmann and Marcos, (2012) | 2.45±0.09 | | |
| 1914-2000 | Vecchio et al., 2019 | 2.43±0.23 | 90% | |
| | | 2.78±0.04 | | trend derived from fit using straight line plus Empirical Mode Decomposition components |
| 1872-2019 | this study | 2.53 ± 0.14 | 95% | deseasoned data |
| *Altimetry period* | | | | |
| 1993-2015 | Vignudelli et al. (2019b) | 6.29±1.53 | 99% | Punta della Salute tide-gauge data |
| | | 5.29±1.27 | 99% | Punta della Salute tide-gauge data with IBE correction |
| 1993-2019 | this study | 5.01±1.75 | 95% | deseasoned data |

1630

**Table 5 - Linear trends of Venetian sea level from tide gauge data (VLM-corrected RSL) and from satellite altimetry (GSL) estimated by various authors in the last 15 years. Errors are STD (68% confidence) except where noted in the Confidence column. The linear fit of observed sea level is used except.**

| Period | Source | Trend (mm/year) | Confidence | Notes |
|---|---|---|---|---|
| *Long-term* | | | | |
| 1890-2007 | Carbognin et al. (2010) | 1.20 ± 0.01 | | deseasoned data |
| 1872-2012 | Zerbini et al. (2017) | 1.23±0.15 | 95% | deseasoned data |
| 1934-2012 | | 1.20±0.35 | 95% | deseasoned data |
| 1905-2005 | Scarascia and Lionello (2013) | 1.3 | N.A. | comparison among adriatic tide gauges |
| 1872-2019 | this study: VLM-corrected tide-gauge data; subsidence estimated after Zerbini et al. (2017), updated to 2019 | 1.23 ± 0.13 | 95% | deseasoned data |
| *Altimetry period* | | | | |
| 1993-2008 | Fenoglio-Marc et al., 2012a | 5.9±1.4 | 95% | altimetry data with dynamic atmospheric correction |
| | | 5.6±1.6 | 95% | tide-gauge data with dynamic atmospheric correction |
| 1993-2015 | Vignudelli et al. (2019b) | 4.25±1.25 | 99% | altimetry point near Venice, IBE removal |
| 1993-2019 | this study | 2.76 ±1.75 | 95% | VLM-corrected deseasoned tide-gauge data |

**Table 6 - Performance of the linear and quadratic regression models applied on the raw annual time series of Venetian RSL and the corresponding climatic component alone (i.e., VLM-corrected RSL) for the periods 1872-2019 and 1993-2019. $R^2$ is the coefficient of determination, AIC is the Akaike Information Criterion, and FPE is the Final Prediction Error.**

|  | RSL | | VLM-corrected RSL | |
|---|---|---|---|---|
|  | Linear | Quadratic | Linear | Quadratic |
| 1872-2019 | | | | |
| $R^2$ | 0.89 | 0.89 | 0.67 | 0.69 |
| AIC | 400 | 398 | 570 | 570 |
| FPE | 15 | 15 | 48 | 47 |
| 1993-2019 | | | | |
| $R^2$ | 0.58 | 0.59 | 0.30 | 0.30 |
| AIC | 70 | 69 | 76 | 76 |
| FPE | 14 | 14 | 18 | 18 |