# Peer review of "Sea-level rise in Venice: historic and future trends (Review article)"

_Natural Hazards and Earth System Sciences, 2020_

## Referee Comment (RC1) · Anonymous Referee #1 · 17 Dec 2020

This is a comprehensive paper reviewing the present knowledge of sea level changes in Venice, including observations and mechanisms and also with some discussion about regional projections in the Northern Adriatic. The manuscript is complete and includes the state of the art in regional sea level in the Mediterranean basin, in addition to provide new computations that include the most recent data for Venice. One strength is the detailed description of vertical land movements, a major driver of relative sea level changes, at many different temporal scales and with emphasis of the distinct acting mechanisms. Another one is the summary of some of the main results in table format. In my opinion, this manuscript deserves publication and will likely become a main reference of sea level variations in Venice. Some parts of the paper are, however, confusing and I think should be reorganised. I am giving details on this latter comment

below.

General: The way section 5 is organised is confusing and, in my opinion, the separation into subsections is misleading. I am giving more details below but, essentially, I do not think that if the section is discussing climate forcing of MSL variability, it is convenient to separate the physical mechanisms from numerical modelling. I would suggest focussing on the mechanisms (namely the effect of atmospheric pressure and winds, the water mass exchanges through Gibraltar, surface fluxes) and discuss their spatial and temporal scales, together with the origin of the data, whether observations or models. Note that barotropic models are mentioned in 5.1.1 but separated from the section on numerical modelling. Overall, I think that this section would benefit from rewriting.

Section 7 discusses many of the gaps of knowledge in Mediterranean physical oceanography. In particular, around lines 766-794, the focus is on the limited knowledge on ocean circulation and the impacts that thermohaline changes in the lateral Atlantic boundary. However, the impact of these processes on basin-scale sea level and in Venice is small in comparison to other effects. This is especially true when climate projections are concerned. Indeed, in section 6.2, the authors include a nice summary of regional projections and uncertainty ranges which are by far much larger than the impacts of regional circulation changes.

Throughout the paper, MSL is used to refer to geocentric MSL in contrast to RSL. But MSL can be computed from RSL from tide gauges as well as from geocentric sea surface height, following the definition here and in Gregory et al (2019). I suggest to add geocentric when referring to RSL corrected for vertical land movements (e.g. section 4.1) to avoid confusion.

Specific comments: - Line 90: "from the open ocean to the coastal zone": strickly speaking this depends on the definition of coastal zone, as altimetry measurements are often only valid tenths of km offshore, where the land signal does not contaminate

the observations.

- l. 118-119: how was the vertical datum determined to an accuracy of 0.1 mm?

- l. 171: determine->determines

- l. 225: b.s.l. -> below sea level (I guess)

- table 2: shouldn't GIA rates from section 3.1.3 be included in this table, for completion?

- fig. 5: please add a-b-c-d labels to subplots

- l. 357: wrong reference to fig. 3

- table 3: the value of subsidence reported for the period 2008-2020 is notably larger than 2003-2010 and 2014-2020 (the latter also from a GPS station), but no comments are provided in the text. Given the differences and the likely overestimation of the GPS VEN1 I think it is worth to mention it in the paper.

- l. 402-404: please provide a reference for the value in Ravenna.

- Fig. 8 and l. ~550-555: it is surprising the differences between fall and winter wavelets and coherence RSL-NAO. Generally, extended winter (Dec-March) NAO is used to correlated with RSL, since it is during these months when the signal is stronger. It is probably worth to comment on the differences found here.

- section 5.1.1 discusses the effect of atmospheric pressure and winds. Yet, the title "atmospheric forcing" is misleading as it might also include heat/water flux exchanges. These processes are then discussed in section 5.1.2 instead. I suggest merging both sub-sections.

- section 5.2 on numerical modelling of the Mediterranean sea level is definitely too short. The authors should also describe other numerical simulations that are available and that provide sea surface height as an output, as
well as the community effort developed by MedCORDEX in which the ocean component is very strong. For the former case, one prominent example is the recent 3D reanalysis b y Simoncelli et al (2017) available through CMEMS (https://resources.marine.copernicus.eu/?option=com_csw&view=details&product_id=MEDSEA_REANALYSIS_PHYS_00 Also, section 5.2 refers only to 3D numerical modelling, while does not mention vertically integrated numerical models that have contributed in the past to unveil the role of atmospheric pressure and wind. Some of these works are discussed in section 5.1.1. The term coastal sea level in the title is not particularly addressed either.

- L. 632-634: GMSL and regional deviations from the global mean, instead of regional sea level changes. This distinction would come up naturally if the mechanisms are described in terms of their spatio-temporal variability in section 5.

- l. 641: where does the range 0.6-1 mm/yr comes from? The numbers discussed above are over 1 mm/yr for late Holocene natural rates of subsidence only.

- l. 800-802: worth mentioning other non-parametric methods such as Empirical Mode Decomposition or Singular Spectrum Analysis for computing time-varying rates of RSL change.

- l. 827: the range given for projected MSL here of 21-100 cm by 2100 should be framed into the corresponding RCPs. Otherwise can be wrongly interpreted. The climate scenario is one of the major sources of uncertainty in projections by 2100, so I think it would be better to state the numbers for the scenarios considered: 32-62 cm under RCP2.6 up to 58-110 cm under RCP8.5, including subsidence. This is important because it helps to interpret that the likelihood of the lower bound is different from than in the upper bound.

---

## Referee Comment (RC2) · Anonymous Referee #2 · 17 Dec 2020

Review of "Sea-level rise in Venice: historic and future trends"

This paper is a review paper about sea-level changes around the city of Venice, Italy. It discusses the observed changes, its relation to land motion and atmospheric forcing, and it provides projections of future sea-level changes for various scenarios.

I enjoyed reading the parts on sea-level observations and land movements. These sections give a clear overview on the subject and I think that these sections are an important contribution to the existing literature. Especially the derivation of long-term land motion, its connection to shorter records from GPS and inSAR, and its temporal variability are very insightful and can function as a blueprint for similar studies in other regions.

[Figure]

However, in my opinion, the sections on sea-level variations, atmospheric forcing and ocean dynamics are sub-par compared to the other sections and need a lot of work before they are in a publishable state. These sections introduce a lot of different processes, but it feels like the coherence between these processes is missing. Also, the mutual consistency between the studies is not discussed (notably missing around L589). At the end, I wonder what is the relative importance of each process. Also, a lot of processes are introduced, but the physics needed to understand how these processes has been left out, while insight in these physics is necessary to understand the role and spatial coherence of these processes. That leads to odd situations, such as with the NAO, which is listed as one of the main processes, but which is not connected to wind and pressure fluctuations, while in reality, these processes are intrinsically linked. Also, there's a disconnect between the processes discussed in Section 5 and the projections in Section 6.

A possible starting point to re-structure and clarify these sections could be to first discuss basin-wide fluctuations in the Mediterranean Sea at various time scales and how they are linked to sea-level variations in the North Atlantic Ocean. These changes and their linkage to the Atlantic Ocean are for example discussed in Fukumori et al (2007), Calafat et al. (2012), Volkov et al. (2019), Landerer & Volkov (2013). As far as I'm aware, almost all decadal and multi-decadal dynamic sea-level variability, as well as longer-term trends in Venice can be explained by basin-wide fluctuations in the Mediterranean, which are in turn linked to alongshore wind forcing along the East coast of the North Atlantic. With such a link established, it will be much easier to couple this knowledge to projections of sterodynamic sea-level changes from CMIP-style climate models. The second step then would be to determine which processes cause significant sea level anomalies in Venice relative to Mediterranean-averaged changes, for example due to local atmospheric forcing or local sterodynamic effects.

Some papers that could be useful as examples for a more structured description of dynamic sea-level variations on various temporal and spatial scales are Calafat et al.

[Figure]

2012 who explicitly shows for each process the amount of explained variability, or Wahl et al. 2013, Dangendorf et al. 2013, 2014, Frederikse et al. 2016, 2019 for papers investigating the dynamics that affect sea-level variability in the North Sea. Another recent example is Piecuch et al. (2019).

Therefore, I recommend to thoroughly revise and rewrite the sections on atmosphere and ocean dynamics before the paper can be published.

Another possible way forward could be to just limit this review to the sections on vertical land motion and subsidence, which on their own would already be a very useful contribution to the literature.

I have brought up a lot of points for sections 4,5, and 6, but nevertheless, I hope the authors find them useful, despite their sheer number.

Detailed and line-by-line comments:

At first, I'd recommend to double-check the definitions from Gregory et al. (2019) throughout the paper. For example, there's an awkward separation between MSL and RSL throughout the paper. I'd recommend getting rid of MSL altogether to avoid any ambiguity and use RSL for sea-level changes relative to land (tide gauge observations) and use GSL (geocentric sea level) for sea-level changes observed by altimetry and tide gauges corrected for vertical land motion. Also please check the definitions of the inverse barometer effect. That effect only includes the static response to atmospheric pressure variations (1 HPa of pressure drop causes a 10 mm sea-level rise), but does not fully represent the sea-level response to pressure changes (Wunsch & Stammer, 1997).

Also a point on the significance on numbers, but I admit this is a bit a matter of personal taste: in an expression like 1.23±0.13, the last numbers are not significant, so something like 1.2±0.1 avoids a false sense of accuracy.

Finally, I encourage the authors to deposit all scripts and data resulting from this paper

in a public repository, such as Zenodo, Github, or Figshare.

Abstract

L39 "An unresolved issue": do you just want to say here that altimetry doesn't measure sea level in the Venice Lagoon because there's no track overlapping ground track?

L40 Water mass exchange: I think this gives the false impression that the issue lies within understanding what's going on at Gibraltar. The real issue here is what's happening to the Northeastern Atlantic Ocean, which directly drives this water mass exchange, see for example (Volkov et al, Calafat et al.).

L42: Subsidence and regional. . . and beyond. These sentences are valid for each and every coastal location, so it's rather trivial. In fact, there's no single known process that causes a true spatially homogeneous sea-level rise. What would be useful here is to explain which processes will cause large deviations from GMSL on various time scales.

L45: "non-negligible differential trends": this is strange wording.

Section 1

L52: Remove 'critically'

L65: As said above, please avoid the term 'MSL' here and throughoput the manuscript, and just distinguish between GSL (not ASL, Gregory et al. 2019) and RSL.

Section 2

L129: The altimetry era actually started one year earlier with ERS1 (and even before that with SEASAT).

L140. One question that is hanging over this section is, to which extent will sea level variations in Venice deviate from sea level a few km off the coast? My guess would be that that effect is rather minimal, except for very short temporal scales (like tides, waves and surges). This is especially relevant, as a few lines further down the paper,

altimetry is directly validated with tide-gauge data, under the assumption that both would measure the same sea-level variability. The answer to this question also circles back to the open challenge described in the abstract, about the challenge of getting altimetry observations as close as possible to the coastline. If these variations are about the same as variations a few km's offshore, why worry about getting closer to the coast?

Section 3

L197: For non-paleo readers, please add an approximate date of MIS 5.5

L269: Your current definition of GIA encompasses both the response to past and contemporary ice mass changes. Following Gregory et al. 2019, it might be a better idea to use GIA for the response to past ice changes, and use 'contemporary GRD effects' for the local sea-level response to contemporary ice-mass changes.

L365: Please define 'MOSE'.

Paragraph 4.1

General: in this section, the authors try to determine the secular trend in sea level in Venice and compare this trend to the global mean. There are a few fundamental problems with this section:

1. What do the authors mean by 'secular trend'? Is there some linear background trend? And if so, which processes are meant to be represented by this trend? To my knowledge, except GIA and maybe some other long-term geologic processes, no single process could cause such a trend. Processes like ice mass loss and thermal expansion are far from linear over 100-year time scales. The idea that this trend has to do with climate change is reinforced by the 'climate component' label in Table 6. I also wonder whether the different numbers found in Table 5 are just caused by computing linear trends over different time spans. Given that sea-level changes often show a lot of multi-decadal variability, even small changes of the period over which the trends are

computed can lead to differences in the range of the differences shown in Table 5. In line 400, the authors make a statement about the non-linearity due to subsidence. This argument is valid for most other processes as well.

2. The comparison to GMSL. The authors compare the trend in Venice to GMSL. This is generally not a good comparison for multiple reasons: due to its proximity to the Greenland Ice Sheet and many glaciated regions, their contributions (which together explain more than half of observed GMSL 1900-2018, Frederikse et al. (2020)) to Venice RSL will be much smaller than to GMSL. On the other hand, the Atlantic Ocean is accumulating heat at a higher rate than the Pacific and Indian Oceans (e.g. Zanna et al. 2019), so the steric component won't follow the global mean either. Therefore, the fact that Venetian and global sea level show similar trends is merely a coincidence, and you cannot expect these numbers to be comparable. Thus, expecting consistency with GMSL doesn't make sense. In fact, it may lead to the false assumption that future sea level rise in Venice will be comparable to GMSL rise. Since various contributors to GMSL rise vary over time as well, this coincidence may not hold. The consistency noted in line 436-437 reinforces this issue: the contributions from Greenland and glaciers in the second half of the 20th century are much smaller than for the first half of the 20th century. Therefore, they do not explain the deviation to be especially large in the second half of the 20th century. In contrary: if it were due to the aforementioned ice melt, one would expect a large difference around the 1930s and a smaller one over the second half of the 20th century.

Line 405: What are 'secular tide-gauge records'? Also, the stations listed here might not be affected by large local subsidence the way Venice is, but I don't think they're unaffected by GIA-induced VLM. Most of these stations have some GNSS records available as well, which can be used to assess whether these stations don't show any VLM.

Line 429. The trends denoted in the IPCC report and Hay and al. refer to global-mean relative sea level, and not global-mean geocentric sea level (See Gregory et al. 2019),

although the difference between both is probably not very large. However, on local scale, the difference can be substantial, even when ignoring local VLM effects. See for example Lickley et al. 2018. Therefore, comparing local geocentric sea level to global-mean relative sea level is not fair.

Paragraph 4.2:

'Multidecadal trends': what do the authors refer to when talking about 'multidecadal trends'? Just 'linear trends over 1993-present'?

L445: Sorry, but I really disagree on this. One can fill whole bookshelves with papers looking into regional sea level from altimetry.

L464: Altimetry also observed the seasonal cycle in sea level. So, when comparing both, why do you need to correct for the seasonal cycle in tide-gauge observations?

Paragraph 4.3 In general, this section seems to lack focus. It's a mixture between variabilities of seasonal sea level, some remarks about peaks in the sea-level spectrum, a wavelet analysis, and reference to some correlation with sunspot cycles.

I'm guessing here, but it might be the case that the authors have mixed up 'periodicity', oscillations that occur with a fixed frequency (such as the seasonal cycle or the M2 tide) versus 'low-frequency variability', variability that occurs on some typical time scales, but cannot be described as (a sum of) periodic functions, such as ENSO or the NAO. The former will cause a clear peak in the spectrum, while the latter is more associated with the behavior discussed in this section, such as intermittent signals in wavelet analyses, and correlations that vary over time. This low-frequency variability is common in global and local sea-level observations. Peaks will show up when computing a spectrum from tide-gauge data, but I wonder about the significance of any of these peaks. Beyond the conclusion that sea level in Venice shows decadal and multi-decadal variability, what should we distill from these peaks? How did the authors determine the significance of the peaks relative to a signal with a red spectrum? An alternative approach here could

be to determine which processes act on with time scales.

L490: The sunspot cycle. This seems a far-fetched link to me. How does the 11-year sunspot cycle cause significant local sea-level variations? I guess that this link is merely coincidence.

L494: What is a 'statistically significant fluctuation'? Significant with respect to what? White noise? A linear trend? Why is a wavelet analysis a good tool to study this?

L500: This stationarity here has been attributed to Greenland and glaciers above, and here it is due to atmospheric processes. What is it?

L505: What is "the integral of the absolute trend differences for bidecadal and shorter periods"?

Sections 5.1 and 5.2

General: like section 4.3, these sections lack focus, tend to introduce a lot of different processes, but at the end, I still don't understand the relative importance of each process. Furthermore, a general introduction of the physics behind the processes is necessary here to understand what's going on. For example, something like: "The North Atlantic Oscillation causes large-scale atmospheric pressure variations on inter-annual scales. The geostrophic winds caused by these variations drive a barotropic sea-level response in Venice, especially in winter."

L511: Steric effects. What about steric effects within the Mediterranean? Do they play an important role? They are discussed in the projections section, but what about observed changes? One could estimate the size of this effect over the last few decades from gridded hydrographic observations, such as EN4 (Good et al. 2013) or Ishii et al. (2017).

L520: What do the authors want to say in this paragraph?

L531: Geostrophic wind is something different than large-scale wind

(https://en.wikipedia.org/wiki/Geostrophic_wind)

L534: "Mass exchange can dominate...": Isn't NAO forcing just a local barotropic response to wind forcing and as such, just added on top of basin-wide fluctuations?

L539: What is "explained linearly"? L540: They...EAWR. This is vague. What happens under the hood here?

L547 This link is also discussed above, but here, the notion that is link is not very strong is omitted. I'd just leave the sunspot studies out.

L553. I don't see this strong correlation from the wavelet coherence plot. The correlation looks weak to me: it does not hold throughput time, goes in and out of phase. From this result, I'd make the opposite conclusions, namely that there's only a weak correlation between winter sea level and the NAO. In for example Piecuch et al. (2019), a wavelet coherence plot that looks similar to this one (their Figure 1) is used to argue for a weak correlation.

L555 In autumn... how significant is this statement? Given the difference between interannual NAO variability and multidecadal subsidence variability, I doubt whether you're just looking for an explanation of insignificant changes.

L558. The contribution of the IBE effect to sea level has been quantified way before 2009 as $\sim$ 1cm of sea-level rise to 1 mbar of pressure drop (see Wunsch & Stammer, 1997). Using regression, you might find other correlation coefficients between sea-level pressure and sea level, but that's because sea-level pressure and sea level interact in many more ways than just the inverse barometer effect. See for example Woodworth et al. 2010. It may also be a good idea to repeat the conclusions from Calafat et al. (2012) that the IB effect only explains a marginal fraction of variability (see their Figure 4).

L563-574: What is the exact point of this paragraph? Hard to follow.

L574ff: Please carefully re-read Calafat et al. 2012: they discuss alongshore coastallytrapped wave propagation along the Atlantic coast, affecting Mediterranean Sea level as a whole. They do not look into the Adriatic Sea in particular.

L584-585: Accordingly -variability. This is a vague sentence. What is an atmospheric bridge in this context, and what does "constitute a potential precursor to multidecadal variability" mean?

L586: "This could contribute to explaining the statistical connection between bidecadal variability of Venetian RSL". Or it could not. This statement needs some evidence, as it now looks like guesswork.

L588: What is "multi-scale acceleration analysis"?

L589ff: This section contrasts with many of the cited papers above, or at least, it needs more explanation. It's namely very unlikely that ice melt can explain 1.3 mm/yr in the Mediterranean, due to the magnitude of past ice melt and GRD effects, causing the Mediterranean to be affected much less than the global mean. How about large-scale changes in the Atlantic Ocean that propagate into the Mediterranean?

L603: Lots of terms that need explanation: "amplitude modulation of water transport", "migration of the eastern hydraulic control".

Section 5.2 In its current setup, this section reads like it has been written without a clear focus in mind. What message do the authors want to tell with this section? It now reads like an unconnected collection of papers that each describe an individual problem, but an overarching story is missing. What models are available, reanalyses, operational forecasts? A good starting point may be the model results from Fukumori et al. (2007).

Section 6 Similar to the previous sections, this section is also somewhat unorganized and lacks focus. After reading it, I am unable to determine a conclusion from the section. Why don't the authors just use the SROCC projections? They should contain all the processes discussed here, except for the vertical land motion part. There is

also a serious lack of information on the methods, and where methods are named, it feels like a bit of a grab-bag of individual estimates. I see SRES scenarios, AR5, SROCC, local projections... Are they combined in a consistent way? Where does the atmospheric forcing come from? Therefore, please thoroughly revise this section with a consistent treatment of scenarios and processes, together with a clear methods section.

L728: This statement is not true. There's no single process that causes a uniform sea-level rise. Therefore, each single process, from ice mass loss to GIA and sterodynamic effects causes a local deviation from GMSL. You might reach the conclusion that the resulting local changes are close to the global mean changes, but that's something different.

Section 7 L736: Similar as in the previous sections: except for VLM, how different are sea-level variations at the coast and just off the coasts as measured by altimetry? Or in other words? Why would we need altimetry closer to the coast?

L795: The method to determine the trend and uncertainties critically depends on the purpose: what should the trend encompass? That should set the method you want to use. For example, neither method gives you a number that should be extrapolated into the past or future.

L800: Similar to above: what does "the shape of the local RSL rise" encompass?

L803: What is "energetic variability"?

L809: This is not the first attempt to create regional sea-level projections for Venice. They are for example in the AR5 and SROCC report, Kopp et al. 2014 and Slangen et al. 2012. All provide local RSL projections with uncertainties. In this respect, Kopp et al. (2014) should be discussed here, since it uses a statistical model to estimate and project land motion not related to GIA.

Figures

There is no reference to Figure 3.

Figure 6: How have confidence intervals been determined? What noise model has been used?

Figure 7 as well: how have all the errors been computed? For panel B? Why use model data from CMIP3, while we have had CMIP5 and now CMIP6 has become available as well?

Figure 8. Middle row: how should I interpret this plot?

References Calafat, F. M., Chambers, D. P., & Tsimplis, M. N. (2012). Mechanisms of decadal sea level variability in the eastern North Atlantic and the Mediterranean Sea. Journal of Geophysical Research: Oceans, 117(C9). https://doi.org/10.1029/2012JC008285

Dangendorf, S., Calafat, F. M., Arns, A., Wahl, T., Haigh, I. D., & Jensen, J. (2014). Mean sea level variability in the North Sea: Processes and implications. Journal of Geophysical Research: Oceans, 119(10), 6820–6841. https://doi.org/10.1002/2014JC009901

Dangendorf, S., Mudersbach, C., Wahl, T., & Jensen, J. (2013). Characteristics of intra-, inter-annual and decadal sea-level variability and the role of meteorological forcing: The long record of Cuxhaven. Ocean Dynamics, 63(2–3), 209–224. https://doi.org/10.1007/s10236-013-0598-0

Frederikse, T., & Gerkema, T. (2018). Multi-decadal variability in seasonal mean sea level along the North Sea coast. Ocean Science, 14(6), 1491–1501. https://doi.org/10.5194/os-14-1491-2018

Frederikse, T., Landerer, F., Caron, L., Adhikari, S., Parkes, D., Humphrey, V. W., Dangendorf, S., Hogarth, P., Zanna, L., Cheng, L., & Wu, Y.-H. (2020). The causes of sea-level rise since 1900. Nature, 584(7821), 393–397. https://doi.org/10.1038/s41586-020-2591-3

Frederikse, T., Riva, R., Kleinherenbrink, M., Wada, Y., van den Broeke, M., & Marzeion, B. (2016). Closing the sea level budget on a regional scale: Trends and variability on the Northwestern European continental shelf. Geophysical Research Letters, 43(20), 10,864-10,872. https://doi.org/10.1002/2016GL070750

Fukumori, I., Menemenlis, D., & Lee, T. (2007). A Near-Uniform Basin-Wide Sea Level Fluctuation of the Mediterranean Sea. Journal of Physical Oceanography, 37(2), 338–358. https://doi.org/10.1175/JPO3016.1

Good, S. A., Martin, M. J., & Rayner, N. A. (2013). EN4: Quality controlled ocean temperature and salinity profiles and monthly objective analyses with uncertainty estimates. Journal of Geophysical Research: Oceans, 118(12), 6704–6716. https://doi.org/10.1002/2013JC009067

Gregory, J. M., Griffies, S. M., Hughes, C. W., Lowe, J. A., Church, J. A., Fukimori, I., Gomez, N., Kopp, R. E., Landerer, F., Cozannet, G. L., Ponte, R. M., Stammer, D., Tamisiea, M. E., & van de Wal, R. S. W. (2019). Concepts and Terminology for Sea Level: Mean, Variability and Change, Both Local and Global. Surveys in Geophysics. https://doi.org/10.1007/s10712-019-09525-z

Ishii, M., Fukuda, Y., Hirahara, S., Yasui, S., Suzuki, T., & Sato, K. (2017). Accuracy of Global Upper Ocean Heat Content Estimation Expected from Present Observational Data Sets. SOLA, 13(0), 163–167. https://doi.org/10.2151/sola.2017-030

Kopp, R. E., Horton, R. M., Little, C. M., Mitrovica, J. X., Oppenheimer, M., Rasmussen, D. J., Strauss, B. H., & Tebaldi, C. (2014). Probabilistic 21st and 22nd century sea-level projections at a global network of tide-gauge sites. Earth's Future, 2(8), 383–406. https://doi.org/10.1002/2014EF000239

Landerer, F. W., & Volkov, D. L. (2013). The anatomy of recent large sea level fluctuations in the Mediterranean Sea. Geophysical Research Letters, 40(3), 553–557. https://doi.org/10.1002/grl.50140

Lickley, M. J., Hay, C. C., Tamisiea, M. E., & Mitrovica, J. X. (2018). Bias in Estimates of Global Mean Sea Level Change Inferred from Satellite Altimetry. Journal of Climate, 31(13), 5263–5271. https://doi.org/10.1175/JCLI-D-18-0024.1

Piecuch, C. G., Huybers, P., Hay, C. C., Kemp, A. C., Little, C. M., Mitrovica, J. X., Ponte, R. M., & Tingley, M. P. (2018). Origin of spatial variation in US East Coast sea-level trends during 1900–2017. Nature, 564(7736), 400–404. https://doi.org/10.1038/s41586-018-0787-6

Piecuch, C. G., Dangendorf, S., Gawarkiewicz, G. G., Little, C. M., Ponte, R. M., & Yang, J. (2019). How is New England coastal sea level related to the Atlantic meridional overturning circulation at 26° N? Geophysical Research Letters, 2019GL083073. https://doi.org/10.1029/2019GL083073

Slangen, A. B. A., Katsman, C. A., van de Wal, R. S. W., Vermeersen, L. L. A., & Riva, R. E. M. (2012). Towards regional projections of twenty-first century sea-level change based on IPCC SRES scenarios. Climate Dynamics, 38(5–6), 1191–1209. https://doi.org/10.1007/s00382-011-1057-6

Volkov, D. L., Baringer, M., Smeed, D., Johns, W., & Landerer, F. W. (2019). Teleconnection between the Atlantic Meridional Overturning Circulation and Sea Level in the Mediterranean Sea. Journal of Climate, 32(3), 935–955. https://doi.org/10.1175/JCLI-D-18-0474.1

Wahl, T., Haigh, I. D., Woodworth, P. L., Albrecht, F., Dillingh, D., Jensen, J., Nicholls, R. J., Weisse, R., & Wöppelmann, G. (2013). Observed mean sea level changes around the North Sea coastline from 1800 to present. Earth-Science Reviews, 124, 51–67. https://doi.org/10.1016/j.earscirev.2013.05.003

Woodworth, P. L., Pouvreau, N., & Wöppelmann, G. (2010). The gyre-scale circulation of the North Atlantic and sea level at Brest. Ocean Science, 6(1), 185–190. https://doi.org/10.5194/os-6-185-2010

[Figure]

Wunsch, C., & Stammer, D. (1997). Atmospheric loading and the oceanic "inverted barometer" effect. Reviews of Geophysics, 35(1), 79–107. https://doi.org/10.1029/96RG03037

Zanna, L., Khatiwala, S., Gregory, J. M., Ison, J., & Heimbach, P. (2019). Global reconstruction of historical ocean heat storage and transport. Proceedings of the National Academy of Sciences, 201808838. https://doi.org/10.1073/pnas.1808838115
* * *

---

## Author Comment (AC1) · 4 Feb 2021

**Reviewer 1**

We thank Reviewer 1 for their appreciation of our manuscript and for the useful comments and suggestions. In the following, we respond to the general as well as specific comments by the Reviewer, which are reported in italics (our response in normal font). We also describe how we would implement the associated changes in the manuscript during the revision.

*This is a comprehensive paper reviewing the present knowledge of sea level changes in Venice, including observations and mechanisms and also with some discussion about regional projections in the Northern Adriatic. The manuscript is complete and includes the state of the art in regional sea level in the Mediterranean basin, in addition to provide new computations that include the most recent data for Venice. One strength is the detailed description of vertical land movements, a major driver of relative sea level changes, at many different temporal scales and with emphasis of the distinct acting mechanisms. Another one is the summary of some of the main results in table format. In my opinion, this manuscript deserves publication and will likely become a main reference of sea level variations in Venice. Some parts of the paper are, however, confusing and I think should be reorganised. I am giving details on this latter comment below.*

Thank you for the overall positive evaluation of our review. We will improve the manuscript based on your recommendations as described in our responses to the specific comments below.

*General: The way section 5 is organised is confusing and, in my opinion, the separation into subsections is misleading. I am giving more details below but, essentially, I do not think that if the section is discussing climate forcing of MSL variability, it is convenient to separate the physical mechanisms from numerical modelling. I would suggest focussing on the mechanisms (namely the effect of atmospheric pressure and winds, the water mass exchanges through Gibraltar, surface fluxes) and discuss their spatial and temporal scales, together with the origin of the data, whether observations or models. Note that barotropic models are mentioned in 5.1.1 but separated from the section on numerical modelling. Overall, I think that this section would benefit from rewriting.*

We would substantially restructure section 5 in the revised manuscript and merge the subsections regarding the physical mechanisms underlying Venetian sea-level variability and numerical modelling. We plan to have an introductory general part and then three separate subsections dealing with the three most important aspects of climate forcing of Mediterranean and Venetian sea level as they emerge from the screened literature, namely:

5.1 Lateral boundary forcing at the Strait of Gibraltar

5.2 Air-sea interaction within the Mediterranean basin

5.3 Linkage with the NAO and other teleconnection patterns

Each section/subsection will embed relevant aspects regarding the associated numerical modelling. The restructuring will benefit from an expanded list of references built on the Reviewers' suggestion.

*Section 7 discusses many of the gaps of knowledge in Mediterranean physical oceanography. In particular, around lines 766-794, the focus is on the limited knowledge on ocean circulation and the impacts that thermohaline changes in the lateral Atlantic boundary. However, the impact of these processes on basin-scale sea level and in Venice is small in comparison to other effects. This is especially true when climate projections are concerned. Indeed, in section 6.2, the authors include a nice summary of regional projections and uncertainty ranges which are by far much larger than the impacts of regional circulation changes.*

In the revision of the manuscript we will homogenize sections 5 and 6 in terms of the relevance of the different processes responsible for Venetian sea-level variations and better highlight, where necessary, why different focus is given in different parts of the manuscript. In the restructured Section 5 we will clearly separate processes that lead to Mediterranean basin-scale sea level changes from those that lead to spatial heterogeneity in sea-level anomalies within the Mediterranean, with a focus on the Adriatic Sea and Venice. We would better confront thermosteric and mass contributions to sea-level, also in relation to the considered timescales. The restructuring will be facilitated by an expanded list of references, which builds on suggestions by both Reviewers.

*Throughout the paper, MSL is used to refer to geocentric MSL in contrast to RSL. But MSL can be computed from RSL from tide gauges as well as from geocentric sea surface height, following the definition here and in Gregory et al (2019). I suggest to add geocentric when referring to RSL corrected for vertical land movements (e.g. section 4.1) to avoid confusion.*

The subsidence correction of RSL series in Zerbini et al. (2017) was mainly based on time series of benchmark heights, obtained during levelling surveys, and are referred to the Zero of the Italian altimetric network in Genoa. Therefore, despite the Altimetric Zero in Genoa being stable (hence at a rather constant geocentric height), the corrected sea level is not strictly speaking geocentric. We would therefore avoid using GSL to refer to VLM-corrected RSL. We propose to update the nomenclature regarding sea level and use the following acronyms in the revised manuscript:

- Relative Sea Level (RSL) change: change in local sea level from the local solid surface (Gregory et al., 2019);

- Geocentric Sea Level (GSL) change: change in local sea level with respect to a geocentric reference, namely a Terrestrial Reference Frame or, equivalently, a reference ellipsoid (Gregory et al., 2019). The acronym GSL is therefore used for satellite altimetry sea-level data;

- Subsidence: land surface sinking (UNESCO, 2020; see also: Gregory et al., 2019);

- VLM-corrected RSL: local sea level derived from tide-gauge RSL data corrected for vertical land movements;

- Global-mean sea level (GMSL): spatially averaged sea level over the World Ocean.

*Specific comments: - Line 90: "from the open ocean to the coastal zone": strickly speaking this depends on the definition of coastal zone, as altimetry measurements are often only valid tenths of km offshore, where the land signal does not contaminate the observations.*

We totally agree with the Reviewer that satellite altimetry becomes challenging when approaching the coasts, as this is also discussed extensively in our manuscript for the case of the Venice lagoon. To avoid confusion in this specific sentence, we would change the quoted part to "with an almost global coverage" in the revised manuscript.

*- l. 118-119: how was the vertical datum determined to an accuracy of 0.1 mm?*

The vertical datum is the one reported in Dorigo (1961). It was defined from the computation of a 25-yr mean tide level and we have no information about the reason for adopting the 0.1-mm precision; it was not an accuracy because it did not come from a measurement. Perhaps (but it is just a speculation) the choice was made for consistency with the 0.1-mm accuracy, typical of benchmark levelling.

In order to clarify this point in the revised manuscript we would modify the text as follows:

"According to Dorigo (1961), Mati established the tide gauge datum at Santo Stefano at 1.50 m below the CM of 1825. In 1910, a new datum was adopted, namely the mean tide level (MTL) of 1884-1909 (central year 1897), computed from the high and low waters measured at Santo Stefano. According to Dorigo (1961), it turned out to be 1.2754 m above the old tide gauge datum, corresponding to 0.2246 m below the CM of 1825. The new reference was named the "Zero Mareografico Punta Salute" (ZMPS)."

*- l. 171: determine->determines*

Thanks, this would be corrected in the revised manuscript.

*- l. 225: b.s.l. -> below sea level (I guess)*

Yes, this would be explicitly reported in the revised manuscript.

*- table 2: shouldn't GIA rates from section 3.1.3 be included in this table, for completion?*

Table 2 reports the time evolution of natural land subsidence in the Venetian region. GIA is just one of the components that contributes to vertical land motions, therefore we believe that it would be misleading to report GIA rates in the table.

*- fig. 5: please add a-b-c-d labels to subplots*

Thanks, labels would be added in the revised manuscript.

*- l. 357: wrong reference to fig. 3*

Thanks, the correct reference is to Fig. 5, to be amended in the revised manuscript.

*- table 3: the value of subsidence reported for the period 2008-2020 is notably larger than 2003-2010 and 2014-2020 (the latter also from a GPS station), but no comments are provided in the text. Given the differences and the likely overestimation of the GPS VEN1 I think it is worth to mention it in the paper.*

This comment refers to the following estimates reported in Table 3:

| Time span | Value (mm/yr) | Technique |
|-----------|---------------|-----------|
| 2003-2010 | 1.0±0.7 | SAR |
| 2008-2020 | 1.7±0.5 | GPS station VEN1(Riva dei Sette Mari) |
| 2014-2020 | 0.9±0.6 | GPS station PSAL (Punta della Salute) |

These estimates result from different techniques and are not only referred to different periods of time, but also to different locations. As reported in the table's caption, the uncertainty associated with the SAR-derived rate represents the ground motion variability at the city scale. Therefore, the rates of the two GPS estimates are actually both consistent with the average value reported over the whole city. SAR has proven capable of detecting ground displacements at the single-building scale and the measured rates range between - 10 and 2 mm/year (see lines 342-343 of the original manuscript). Besides that, one of the main purposes of Section 3 is to clarify that the subsidence behavior in Venice is definitely non linear. Therefore, temporal variations are to be expected even at short time scales.

We would better discuss differences between estimates reported in Table 3 in the revised manuscript, along the line of this response to the Reviewer.

*- l. 402-404: please provide a reference for the value in Ravenna.*

The reference is the same used for Venice, i.e., Zerbini et al. (2017). We would amend the sentence accordingly in the revised manuscript.

*- Fig. 8 and l. 550-555: it is surprising the differences between fall and winter wavelets and coherence RSL-NAO. Generally, extended winter (Dec-March) NAO is used to correlated with RSL, since it is during these months when the signal is stronger. It is probably worth to comment on the differences found here.*

We agree that the NAO is more stable in winter and therefore the connection between NAO and Venetian sea level is more significant during this period of the year, whereas other modes of large-scale atmospheric variability become more important than the NAO in other seasons. In fact, this was also shown in Zanchettin et al. (2009). In our manuscript, the inclusion of an analysis of autumn series was mainly motivated due to the relevance of this season for storm surges in the Venice lagoon. Also following a comment by Reviewer #2, we would better stress the seasonal difference in the NAO imprint on Venetian sea level. Our responses to Reviewer #2 provide more details about how we plan to revise the manuscript regarding the NAO.

As mentioned above, we would have a specific subsection dedicated to the NAO in the revised manuscript (section 5.3 "Linkage with the NAO"), which will also contain general introductory information about the NAO, and a more critical discussion about the results from the wavelet coherence analysis. We also plan to add a figure illustrating maps of sea-level pressure and near-surface wind anomalies linked to different phases of the NAO, to aid the

discussion about the atmospheric mechanical forcing of Venetian sea level associated with the NAO. Overall, we expect the new section 5.3 in the revised manuscript to provide a much clearer presentation of what is known about the connection between Mediterranean and Venetian sea level and the NAO.

*- section 5.1.1 discusses the effect of atmospheric pressure and winds. Yet, the title "atmospheric forcing" is misleading as it might also include heat/water flux exchanges. These processes are then discussed in section 5.1.2 instead. I suggest merging both sub-sections.*

As outlined above, we would carefully revise the manuscript toward a better structure of section 5. The subsections mentioned in this comment will be removed, and their content merged. We will clearly present local atmospheric mechanical forcing (IBE and wind forcing) and ocean-atmosphere surface fluxes as different aspects in a dedicated revised subsection within section 5 (new section 5.2, see above). To better present local atmospheric forcing of Venetian sea level, we also plan to have an additional figure illustrating the sea-level pressure and near-surface wind anomalous patterns over the Euro-Mediterranean region associated to Venetian sea-level anomalies.

*- section 5.2 on numerical modelling of the Mediterranean sea level is definitely too short. The authors should also describe other numerical simulations that are available and that provide sea surface height as an output, as well as the community effort developed by MedCORDEX in which the ocean component is very strong. For the former case, one prominent example is the recent 3D reanalysis by Simoncelli et al (2017) available through CMEMS (https://resources.marine.copernicus.eu/?option=com_csw&view=details&product_id=MEDS EA_REANALYSIS_PHYS_006_Also, section 5.2 refers only to 3D numerical modelling, while does not mention vertically integrated numerical models that have contributed in the past to unveil the role of atmospheric pressure and wind. Some of these works are discussed in section 5.1.1.*

*The term coastal sea level in the title is not particularly addressed either.*

As replied above to the general comment by the Reviewer, in the restructuring of Section 5 we would get rid of the original subsection on numerical modelling. In the revised manuscript we would avoid having a complete list of model types and initiatives regarding the Mediterranean Sea and instead embed the relevant parts of section 5.2 of the original manuscript along the discussion of the physical mechanisms responsible for variations in Venetian sea level. Reference to models of different complexity should then become more straightforward.

We would still mention MedCORDEX and the availability of regional reanalysis in Section 6 (gaps of knowledge and opportunities for progress) as follows: "Accordingly, over recent years, considerable efforts have been invested into developing and applying regional climate and ocean circulation models approaching the issue of dynamical downscaling from different perspectives (e.g., Somot et al., 2008; Sannino et al., 2009; Artale et al., 2010; Naranjo et al., 2014; Sein et al., 2015; Turuncoglu and Sannino, 2017; Androsov et al., 2019; Palma et al., 2020), also in the context of coordinated international activities (e.g., MedCORDEX) including the use of data assimilation techniques to obtain regional reanalysis for the Mediterranean Sea (Simoncelli et al., 2014)."

*- L. 632-634: GMSL and regional deviations from the global mean, instead of regional sea level changes. This distinction would come up naturally if the mechanisms are described in terms of their spatio-temporal variability in section 5.*

Agreed.

*- l. 641: where does the range 0.6-1 mm/yr comes from? The numbers discussed above are over 1 mm/yr for late Holocene natural rates of subsidence only.*

The smaller rate was proposed in old studies (e.g., Carbognin et al., 1976), whereas more recent estimates are more consistently in the order of 1 mm/yr. We would amend this in the revised version and only refer to the current estimates (hence1 mm/yr).

*- l. 800-802: worth mentioning other non-parametric methods such as Empirical Mode Decomposition or Singular Spectrum Analysis for computing time-varying rates of RSL change.*

We would mention both techniques in the revised manuscript: "Further research (e.g., exploiting techniques such as empirical mode decomposition and singular spectrum analysis) is therefore needed …".

*- l. 827: the range given for projected MSL here of 21-100 cm by 2100 should be framed into the corresponding RCPs. Otherwise can be wrongly interpreted. The climate scenario is one of the major sources of uncertainty in projections by 2100, so I think it would be better to state the numbers for the scenarios considered: 32-62 cm under RCP2.6 up to 58-110 cm under RCP8.5, including subsidence. This is important because it helps to interpret that the likelihood of the lower bound is different from than in the upper bound.*

We would explicit the scenarios in the revised manuscript, as follows: "Projected climatically-induced Venetian sea-level rise from estimates for the GMSL corrected for further uncertainty associated with the regional redistribution of different mass contribution components such as glacier, ice-sheet and groundwater is in the range from 22 to 52 (from 48 to 100) centimeters by 2100 for the RCP2.6 (RCP5.8) concentration scenarios."

---

## Author Comment (AC2) · 4 Feb 2021

**Reviewer 2**

We thank Reviewer 2 for their appreciation of our manuscript and for the useful comments and suggestions. In the following, we respond to the general as well as specific comments by the Reviewer, which are reported in italics (our response in normal font). We also describe how we would implement the associated changes in the manuscript during the revision.

*Review of "Sea-level rise in Venice: historic and future trends"*

*This paper is a review paper about sea-level changes around the city of Venice, Italy. It discusses the observed changes, its relation to land motion and atmospheric forcing, and it provides projections of future sea-level changes for various scenarios. I enjoyed reading the parts on sea-level observations and land movements. These sections give a clear overview on the subject and I think that these sections are an important contribution to the existing literature. Especially the derivation of long-term land motion, its connection to shorter records from GPS and inSAR, and its temporal variability are very insightful and can function as a blueprint for similar studies in other regions.*

*However, in my opinion, the sections on sea-level variations, atmospheric forcing and ocean dynamics are sub-par compared to the other sections and need a lot of work before they are in a publishable state. These sections introduce a lot of different processes, but it feels like the coherence between these processes is missing. Also, the mutual consistency between the studies is not discussed (notably missing around L589). At the end, I wonder what is the relative importance of each process. Also, a lot of processes are introduced, but the physics needed to understand how these processes has been left out, while insight in these physics is necessary to understand the role and spatial coherence of these processes. That leads to odd situations, such as with the NAO, which is listed as one of the main processes, but which is not connected to wind and pressure fluctuations, while in reality, these processes are intrinsically linked. Also, there's a disconnect between the processes discussed in Section 5 and the projections in Section 6.*

In retrospective, we agree that sections 5 and 6 can be substantially improved. The change to be implemented to both sections during the revision would include their general restructuring. We would better emphasize consistencies and inconsistencies between published results, and better link statistical connections with associated physical mechanisms.

For the specific comment on the results illustrated around line 589 (Scarascia and Lionello, 2013) and their apparent inconsistency with those illustrated around line 500 (Marcos and Tsimplis, 2008), we plan to present both studies together within section 4.3 and clarify the reasons for the different conclusions they reach. Specifically, Marcos and Tsimplis (2008) computes the expansion of the water column missing the contribution of the redistribution of mass, while the approach of Scarascia and Lionello (2013) accounts for it. Further, Scarascia and Lionello (2013) consider explicitly the Adriatic Sea, which is not present in the maps of Marcos and Tsimplis (2008), whose approach produces negligible values of sea level rise in shallow water areas.

Regarding the connection between NAO and Venetian sea level, we would restructure Section 5 to have one specific subsection on the NAO, after the physical processes have been introduced. The new structure of Section 5 would be:

5. Climatic drivers of Venetian sea-level fluctuations

5.1 Lateral boundary forcing at the Strait of Gibraltar

5.2 Air-sea interaction within the Mediterranean  basin

5.3 Linkage with the NAO and other teleconnection patterns

The role of IBE and wind setup for the NAO connection was already mentioned in the original manuscript (lines 531-537). To better support this connection, in addition to changes in the structure of the section we plan to add a figure showing surface wind and sea-level pressure anomalies over the Euro-Mediterranean region linked to Venetian sea level variations as well as with different phases of the NAO.

Regarding the connection between processes discussed in section 5 and their relevance for section 6, in the revised version we would include an introductory paragraph and enhance references to section 5 in section 6 as far as possible, making sure that comparable relevance is given to each process in section 5 and 6.

*A possible starting point to re-structure and clarify these sections could be to first discuss basin-wide fluctuations in the Mediterranean Sea at various time scales and how they are linked to sea-level variations in the North Atlantic Ocean. These changes and their linkage to the Atlantic Ocean are for example discussed in Fukumori et al (2007), Calafat et al. (2012), Volkov et al. (2019), Landerer & Volkov (2013). As far as I'm aware, almost all decadal and multi-decadal dynamic sea-level variability, as well as longer-term trends in Venice can be explained by basin-wide fluctuations in the Mediterranean, which are in turn linked to alongshore wind forcing along the East coast of the North Atlantic. With such a link established, it will be much easier to couple this knowledge to projections of sterodynamic sea-level changes from CMIP-style climate models. The second step then would be to determine which processes cause significant sea level anomalies in Venice relative to Mediterranean-averaged changes, for example due to local atmospheric forcing or local sterodynamic effects.*

Thanks for the suggestion and for pointing at these papers, which we would include in the revised manuscript. As mentioned above, we plan to substantially restructure section 5 following the Reviewer's recommendation.

*Some papers that could be useful as examples for a more structured description of dynamic sea-level variations on various temporal and spatial scales are Calafat et al. 2012 who explicitly shows for each process the amount of explained variability, or Wahl et al. 2013, Dangendorf et al. 2013, 2014, Frederikse et al. 2016, 2019 for papers investigating the dynamics that affect sea-level variability in the North Sea. Another recent example is Piecuch et al. (2019). Therefore, I recommend to thoroughly revise and rewrite the sections on atmosphere and ocean dynamics before the paper can be published.*

Thanks for pointing at these papers, which we looked at carefully and got inspiration for the revision of our manuscript. We are confident that a revised manuscript with a substantially restructured and rewritten section 5 as outlined in this response would be suitable for publication.

*Another possible way forward could be to just limit this review to the sections on vertical land motion and subsidence, which on their own would already be a very useful contribution to the literature. I have brought up a lot of points for sections 4, 5, and 6, but nevertheless, I hope the authors find them useful, despite their sheer number.*

We believe that sections 4, 5 and 6 are a necessary part of this literature review. We are confident that the changes we plan to implement in the revised manuscript and illustrated in this response will convince the Reviewer that the paper is worth being published with all its components.

*Detailed and line-by-line comments:*

*At first, I'd recommend to double-check the definitions from Gregory et al. (2019) throughout the paper. For example, there's an awkward separation between MSL and RSL throughout the paper. I'd recommend getting rid of MSL altogether to avoid any ambiguity and use RSL for sea-level changes relative to land (tide gauge observations) and use GSL (geocentric sea level) for sea-level changes observed by altimetry and tide gauges corrected for vertical land motion. Also please check the definitions of the inverse barometer effect. That effect only includes the static response to atmospheric pressure variations (1 HPa of pressure drop causes a 10 mm sea-level rise), but does not fully represent the sea-level response to pressure changes (Wunsch & Stammer, 1997).*

The subsidence correction of the RSL series in Zerbini et al. (2017) was mainly based on time series of benchmark heights, obtained during levelling surveys, and are referred to the Zero of the Italian altimetric network in Genoa. Therefore, despite the Altimetric Zero in Genoa being stable (hence at a rather constant geocentric height), the corrected sea level is not strictly speaking geocentric. We would therefore avoid using GSL to refer to VLM-corrected RSL. We propose to update the nomenclature regarding sea level and use the following acronyms:

- Relative Sea Level (RSL) change: change in local sea level relative to the local solid surface (Gregory et al., 2019);

- Geocentric Sea Level (GSL) change: change in local sea level with respect to a geocentric reference, namely a Terrestrial Reference Frame or, equivalently, a reference ellipsoid (Gregory et al., 2019). The acronym GSL is therefore used for satellite altimetry sea-level data;

- Subsidence: land surface sinking (UNESCO, 2020; see also: Gregory et al., 2019);

- VLM-corrected RSL: local sea level derived from tide-gauge RSL data corrected for vertical land movements;

- Global-mean sea level (GMSL): spatially averaged sea level over the World Ocean.

*Also a point on the significance on numbers, but I admit this is a bit a matter of personal taste: in an expression like 1.230.13, the last numbers are not significant, so something like 1.20.1 avoids a false sense of accuracy.*

We understand this comment to refer in particular to section 2, where indeed some reported values prospect a high accuracy (as also reported by Reviewer #1). We clarify that when reporting results from published papers, we kept the original number provided by the authors. In the revised manuscript we would amend the text so that it is clear that this is the case. Also, the numbers provided by our own calculations will be carefully checked in that they provide a true sense of accuracy.

*Finally, I encourage the authors to deposit all scripts and data resulting from this paper in a public repository, such as Zenodo, Github, or Figshare.*

We agree to publish relevant data and scripts in a public repository, or even as a supplement to the paper upon final publication.

*Abstract*

*L39 "An unresolved issue": do you just want to say here that altimetry doesn't measure sea level in the Venice Lagoon because there's no track overlapping ground track?*

The issue is the contrast between tide gauge and satellite altimetry data. As far as the Venice Lagoon is concerned, the issue is twofold. First, the altimeter may not cross the lagoon depending on the ground track configuration design. TOPEX/Poseidon and Jason series do not cross the lagoon. Sentinel-2b instead crosses the lagoon, so data could be recovered if we improve the processing. As detailed in the main text, the Venice Lagoon is a challenging target due to several factors, among which are presence of land and specular reflections, and efforts to retrieve data are ongoing. To avoid misinterpretation or confusion, in the revised manuscript we would change the sentence in the abstract as: "An unresolved issue is the contrast between the observational capacity of tide gauges and satellite altimetry, with the latter tool not providing reliable data within the Venice Lagoon yet. It is therefore currently not possible to take advantage of the full potential of along-track altimetric data in the inter-technique comparison."

*L40 Water mass exchange: I think this gives the false impression that the issue lies within understanding what's going on at Gibraltar. The real issue here is what's happening to the Northeastern Atlantic Ocean, which directly drives this water mass exchange, see for example (Volkov et al, Calafat et al.).*

In fact, to better estimate Mediterranean sea- level variations under future global warming scenarios it is important to better understand and simulate both, the water mass exchange within the Strait of Gibraltar and precursors of its variability. To better stress this in the abstract, in the revised manuscript we would change the sentence as follows: "Water mass exchange through the Strait of Gibraltar and its drivers currently constitute a source of substantial uncertainty for estimating future deviations of the Mediterranean mean sea-level trend from the global-mean value."

*L42: Subsidence and regional: : : and beyond. These sentences are valid for each and every coastal location, so it's rather trivial. In fact, there's no single known process that causes a true spatially homogeneous sea-level rise. What would be useful here is to explain which processes will cause large deviations from GMSL on various time scales.*

We agree this is a rather general sentence. We would drop this in the revised version.

*L45: "non-negligible differential trends": this is strange wording.*

We agree. We would skip the quoted sentence.

*Section 1*

*L52: Remove 'critically'*

Agreed

*L65: As said above, please avoid the term 'MSL' here and throughout the manuscript, and just distinguish between GSL (not ASL, Gregory et al. 2019) and RSL.*

Agreed

*Section 2*

*L129: The altimetry era actually started one year earlier with ERS1 (and even before that with SEASAT).*

We agree that the United States were the first to fly a satellite-borne altimeter, with the Skylab and Geos3 missions, Seasat in 1978 (the first satellite to provide data) and Geosat in 1985. Seasat had only 110-day lifetime (end of mission due to a malfunction). Topex/Poseidn was launched in August 1992. Ers-1 was launched in 1991 but the revisiting was initially 3 days and not exploitable for sea-level studies the ground track coverage is too coarse. The 35-day phase started in 1992. The continuous altimeter era is considered starting on1992. We would avoid reference to the beginning of the satellite era and modify the sentence as follows in the revised manuscript: "Since the first satellite altimetry missions in the mid-1970s, the accuracy of sea-surface height measurement has increased considerably until high-precision and routinely measured altimetric data were made available in the early 1990s with the launch of the TOPEX/Poseidon mission"

*L140. One question that is hanging over this section is, to which extent will sea level variations in Venice deviate from sea level a few km off the coast? My guess would be that that effect is rather minimal, except for very short temporal scales (like tides, waves and surges). This is especially relevant, as a few lines further down the paper, altimetry is directly validated with tide-gauge data, under the assumption that both would measure the same sea-level variability. The answer to this question also circles back to the open challenge described in the abstract, about the challenge of getting altimetry observations as close as possible to the coastline. If these variations are about the same as variations a few km's offshore, why worry about getting closer to the coast?*

We agree that sea level in the Northern Adriatic is rather homogeneous. The non-trivial point is to determine how sea level signals in the Northern Adriatic propagate into and then within the Venice Lagoon, for which satellite altimetry data are not available yet. As detailed later in section 4.2, the deviation between sea level in Venice measured by tide gauges and in the open sea in the vicinity to the lagoon measured by satellite altimetry provide estimates of interdecadal sea-level trends that do overlap indeed within uncertainties, but they do so only marginally after the tide-gauge data are corrected for vertical land movements, for which the estimate is about half that from satellite data. This motivates the open issue pointed out in the abstract. In the revised manuscript, we would take advantage of a newly published paper (De Biasio et al., 2020) to better assert that the best approach to obtain robust estimates is not to assess each site independently from the others, rather to use multiple sites and exploit synergies between available measuring systems (namely altimetry and tide gauges).

In the revised manuscript we plan to add a sentence like the following one at the end of the last paragraph of section 4.2: "So, despite satellite- and tide gauge-based trend estimates still overlap

within their uncertainties after tide-gauge data have been corrected for subsidence, they do so only marginally. The underlying causes remain to be understood, which motivates the search for improved approaches to integrate both measuring systems (De Biasio et al., 2020)."

*Section 3*

*L197: For non-paleo readers, please add an approximate date of MIS 5.5*

Agreed, in the revised manuscript we would specify "subsidence using the Marine Isotope Stage (MIS) 5.5 event between 130 and 120 kyr ago as a reference to separate geologically older and newer RSL changes"

*L269: Your current definition of GIA encompasses both the response to past and contemporary ice mass changes. Following Gregory et al. 2019, it might be a better idea to use GIA for the response to past ice changes, and use 'contemporary GRD effects' for the local sea-level response to contemporary ice-mass changes.*

We agree that there could be some confusion. We propose to differentiate the terminology for past and contemporary ice-mass changes by using "GIA" and "GIA including contemporary ice mass changes", respectively, since the two phenomena are governed by the same physics.

*L365: Please define 'MOSE'.*

We would add the definition of MOSE in the revised manuscript: "(so-called "MOdulo Sperimentale Elettromeccanico" or MOSE, see Lionello et al., 2020a)"

*Paragraph 4.1*

*General: in this section, the authors try to determine the secular trend in sea level in Venice and compare this trend to the global mean. There are a few fundamental problems with this section:*

*1. What do the authors mean by 'secular trend'? Is there some linear background trend? And if so, which processes are meant to be represented by this trend? To my knowledge, except GIA and maybe some other long-term geologic processes, no single process could cause such a trend. Processes like ice mass loss and thermal expansion are far from linear over 100-year time scales. The idea that this trend has to do with climate change is reinforced by the 'climate component' label in Table 6. I also wonder whether the different numbers found in Table 5 are just caused by computing linear trends over different time spans. Given that sea-level changes often show a lot of multi-decadal variability, even small changes of the period over which the trends are computed can lead to differences in the range of the differences shown in Table 5. In line 400, the authors make a statement about the non-linearity due to subsidence. This argument is valid for most other processes as well.*

Following the aim of this literature review, this section summarizes published estimates of the observed rate of sea-level rise in Venice, and highlights and attempts to explain differences between them. We certainly agree that such estimates should not imply that the underlying process is necessarily linear, despite estimates being often obtained from the application of linear statistical techniques. In the revised version of the manuscript we would change "secular trend" with "Average

rates of sea-level rise over centennial periods" and would add some clarification in this regard, by adding a text along the following lines in the introductory part of section 4: " Average rates of sea-level rise are often calculated using some linear fit to the available data. Given the variety and non-linearity of the processes known to contribute to sea-level trends on the considered centennial and multidecadal time scales – some explicitly accounted for in the trend calculation -, such estimates should not be intended as necessarily representing a linear process (see Section 6)."

We also certainly agree that there is substantial multidecadal variability in the sea-level records, which can explain differences between estimates of the secular rate of sea level rise provided by different authors (e.g., the apparent discrepancy between the conclusions of Marcos and Tsimplis, 2008, and Scarascia and Lionello, 2013, quoted above). We would better state this in the discussion by adding something along the following lines: "The presence of substantial variations in Venetian RSL and VLM-corrected RSL multidecadal trends contributes to explain, together with methodological aspects in the calculation, the different estimates of the average rate of sea-level rise obtained by different authors considering different periods."

*2. The comparison to GMSL. The authors compare the trend in Venice to GMSL. This is generally not a good comparison for multiple reasons: due to its proximity to the Greenland Ice Sheet and many glaciated regions, their contributions (which together explain more than half of observed GMSL 1900-2018, Frederikse et al. (2020)) to Venice RSL will be much smaller than to GMSL. On the other hand, the Atlantic Ocean is accumulating heat at a higher rate than the Pacific and Indian Oceans (e.g. Zanna et al. 2019), so the steric component won't follow the global mean either. Therefore, the fact that Venetian and global sea level show similar trends is merely a coincidence, and you cannot expect these numbers to be comparable. Thus, expecting consistency with GMSL doesn't make sense. In fact, it may lead to the false assumption that future sea level rise in Venice will be comparable to GMSL rise. Since various contributors to GMSL rise vary over time as well, this coincidence may not hold. The consistency noted in line 436-437 reinforces this issue: the contributions from Greenland and glaciers in the second half of the 20th century are much smaller than for the first half of the 20th century. Therefore, they do not explain the deviation to be especially large in the second half of the 20th century. In contrary: if it were due to the aforementioned ice melt, one would expect a large difference around the 1930s and a smaller one over the second half of the 20th century.*

We agree with this comment. In fact, we did not expect consistency to be necessary between GMSL and sea-level variations in Venice. Our analysis was indeed motivated by the fact that available sea-level rise projections for Venice are in some cases directly based upon estimates of the GMSL rise (see, for instance, Troccoli et al., 2012, and Carbogning et al., 2010). Possibly, this is better explained in the accompanying editorial to the special issue (Lionello et al., 2020, discussion paper available at: https://nhess.copernicus.org/preprints/nhess-2020-367/), where Venetian sea level is more appropriately compared to the midlatitude eastern North Atlantic sea level.

We would still keep the comparison between GMSL and Venetian sea level trends in the revised manuscript, but would add an introductory sentence to explain the rationale of the comparison along the following lines: "As illustrated in Sections 5 and 6, Mediterranean sea level, hence Venetian sea-level variations are tightly connected to sea-level variations in the midlatitude eastern North Atlantic, whose underlying processes differ from those in other oceanic basins.  Therefore, any statistical consistency between historical Venetian RSL/VLM-corrected RSL and GMSL rise should not give the false impression that both variables are interchangeable and that any consistency in the historical period necessarily holds in the future. Still, it is instructive to compare estimates of Venetian RSL/VLM-corrected RSL and GMSL rise during the 20th Century."

We would then refer again to the comparison in the restructured Section 5 and provide an explanation along the one provided by the Reviewer in this comment.

Moreover, we would add a sentence clarifying the implications of associating trends in GMSL and Venetian sea level in the revised abstract, along the one that follows: "Even if consistent with each other, Venetian and global-mean sea-level trends are caused by a different combination of processes, whose individual contribution varies through time, hence future projections of Venetian sea-level rise should not build on global-mean estimates."

*Line 405: What are 'secular tide-gauge records'? Also, the stations listed here might not be affected by large local subsidence the way Venice is, but I don't think they're unaffected by GIA-induced VLM. Most of these stations have some GNSS records available as well, which can be used to assess whether these stations don't show any VLM.*

By "secular tide-gauge records" we meant records spanning over one century or more. The precise time spans covered by each time series are indicated in the text (at lines 400 for Venice and 404 for M. di Ravenna). The wording would be revised by replacing "secular" with formulations like "centennial" or "spanning over a century".

The stability of the tide gauges included in this discussion is provided in the references reported at line 408 of the original manuscript. These include (also) GPS-based results, see in particular Sanchez et al., 2018 (their Figures 8 and 14).

*Line 429. The trends denoted in the IPCC report and Hay and al. refer to global-mean relative sea level, and not global-mean geocentric sea level (See Gregory et al. 2019), although the difference between both is probably not very large. However, on local scale, the difference can be substantial, even when ignoring local VLM effects. See for example Lickley et al. 2018. Therefore, comparing local geocentric sea level to global-mean relative sea level is not fair.*

We definitely agree that there might be substantial difference between local and global-mean sea-level estimates. Obviously, we also agree that geocentric and relative estimates of sea-level variations can differ, this is actually the rationale for the whole discussion provided in Sections 2 and 3. However, the following points should be considered:

1) The subsidence correction of RSL series in Zerbini et al. (2017) was mainly based on time series of benchmark heights, obtained during levelling surveys, and are referred to the Zero of the Italian altimetric network in Genoa. Therefore, despite the Altimetric Zero in Genoa is stable (hence at a rather constant geocentric height), the corrected sea level is not strictly speaking geocentric;
2) In fact, the focus of Zerbini et al. (2017) is regional rather than local as six tide-gauges characterized by centennial time series along the coasts of the Mediterranean Sea were considered;
3) hopefully, the impact of VLM on GMSL from tide-gauge records is limited. Over the altimetric period, in fact, the GMSL rise derived from tide-gauge records is consistent with altimetric measurements (e.g. IPCC 5th report, Chapt. 13, Fig. 13.7).

In the revised manuscript, we plan to modify the text as follows: "[...] full-period trends in both the original RSL (2.53 +/- 0.14 mm/year) and the VLM-corrected RSL (1.23 +/- 0.13 mm/year)"

Moreover, for clarity, on line 389 we would replace 'relative MSL' with 'RSL'.

Overall, in the revised manuscript we would change the last paragraph of section 4.1 along the line of the following paragraph:

"As illustrated in Sections 5 and 6, Mediterranean sea level, hence Venetian sea-level variations are tightly connected to sea-level variations in the midlatitude eastern North Atlantic, whose underlying processes differ from those in other oceanic basins. Therefore, any statistical consistency between historical Venetian RSL/VLM-corrected RSL and GMSL rise should not give the false impression that both variables are interchangeable and that any consistency in the historical period necessarily holds in the future. Still, it is instructive to compare estimates of Venetian RSL/VLM-corrected RSL and GMSL rise during the 20th Century. Venetian sea-level trends are smaller than GMSL trends reported in the fifth assessment report of the Intergovernmental Panel on Climate Change (IPCC-AR5), quantified as 1.7 [1.5 to 1.9] mm/year (likelihood >90%, period from 1901 to 2010, see: Church et al., 2013). They are, however, consistent with revisited estimates of historical GMSL rise that include significantly slower rates than reported by the IPCC-AR5 for the pre-altimetry period, e.g., 1.2±0.2 mm/year (90% confidence interval, Hay et al., 2015), 1.1 ± 0.3 mm/year (99% confidence interval, Dangendorf et al., 2017) and 1.56 ± 0.33 mm/year (90% confidence interval, Frederikse et al., 2020). Figure 6 revisits the connection between Venetian RSL/VLM-corrected RSL and GMSL trends on time scales ranging from interannual to centennial. Clearly, the significant difference between centennial trends in Venetian RSL and GMSL is strongly damped when the contribution of subsidence is removed, confirming the critical role of vertical land motions in determining local RSL variations. Nonetheless, the Venetian VLM-corrected RSL appears to rise at a lower rate than the GMSL over the second half of the 20th century (Fig. 6a). Note that the GMSL-Venetian sea-level discrepancy observed in the first portion of the record is resolved when uncertainty in GMSL estimate is considered (not shown)"

*Paragraph 4.2:*

*'Multidecadal trends': what do the authors refer to when talking about 'multidecadal trends'? Just 'linear trends over 1993-present'?*

Similar to what we plan to do for paragraph 4.1 we would change the title to "Rates of sea-level rise during the satellite altimetry era".

*L445: Sorry, but I really disagree on this. One can fill whole bookshelves with papers looking into regional sea level from altimetry.*

In the revised manuscript we would rephrase the sentence as: "An overall GMSL trend of about 3 mm/year during the satellite altimetry period is consistently reported by several studies (Hay et al., 2015; Chen et al., 2016; Dangendorf et al., 2017; Quartly et al., 2017). Regional trends can deviate considerably from the global mean (e.g., Scharroo et al., 445 2013; Legeais et al., 2018; Cazenave et al., 2019)."

*L464: Altimetry also observed the seasonal cycle in sea level. So, when comparing both, why do you need to correct for the seasonal cycle in tide-gauge observations?*

In fact the removal of the seasonal signal is just a common procedure. We would rephrase the statement to clarify this in the revised manuscript, also citing the reference paper by Carrère and Lyard (2003).

Ref: Carrère L., Lyard F (2003) : Modeling the barotropic response of the global ocean to atmospheric wind and pressure forcing – comparisons with observations. Geophys Res Lett 30(6):1275.doi:10.1029/2002GL016473

*Paragraph 4.3 In general, this section seems to lack focus. It's a mixture between variabilities of seasonal sea level, some remarks about peaks in the sea-level spectrum, a wavelet analysis, and reference to some correlation with sunspot cycles. I'm guessing here, but it might be the case that the authors have mixed up 'periodicity', oscillations that occur with a fixed frequency (such as the seasonal cycle or the M2 tide) versus 'low-frequency variability', variability that occurs on some typical time scales, but cannot be described as (a sum of) periodic functions, such as ENSO or the NAO. The former will cause a clear peak in the spectrum, while the latter is more associated with the behavior discussed in this section, such as intermittent signals in wavelet analyses, and correlations that vary over time. This low-frequency variability is common in global and local sea-level observations. Peaks will show up when computing a spectrum from tide-gauge data, but I wonder about the significance of any of these peaks. Beyond the conclusion that sea level in Venice shows decadal and multi-decadal variability, what should we distill from these peaks? How did the authors determine the significance of the peaks relative to a signal with a red spectrum? An alternative approach here could be to determine which processes act on with time scales.*

As correctly stated by the Reviewer, in this case the term "periodicity" was intended in the sense of a spectral peak at a certain frequency. To avoid confusion, in the revised version we would replace the term "periodicity" with "variability mode and significant spectral component". For instance, we would rephrase the sentence at lines 477-478 as follows: "Hereafter, we indicate detection of a statistically significant spectral component around a period of XX years with $O_{XX_{years}}$, where O means order of magnitude."

We remark that, in the spirit of a literature review, we report the presence of significant spectral components detected by previous studies in the tide-gauge record of Venice. We provided the analysis of the updated seasonal tidal records for autumn and winter in order to update published results and to confirm/revise their conclusions. Similarly, we illustrate possible linkages with large-scale atmospheric and oceanic phenomena as they are discussed in published papers. An unambiguous association of each variability mode with a well-defined process has not been done yet, and it is out of the scope of this review. In fact, we would better highlight in the revised manuscript that variations in the phase lag and intermittent significance in the wavelet coherence spectra indicate that caution should be taken when establishing connections between local sea-level variability and large-scale climatic modes, as they may not be robust over multidecadal or centennial time scales.

Concerning the determination of significance of the spectral peaks, the spectral peaks were tested against a red-noise (lag-1 autoregressive model) null-hypothesis. A lag-1 autoregressive model characterized by high power spectral density at lower frequencies. It is the most widely used model for geophysical purposes since a large class of geophysical processes produce output statistically compatible with red noise hypothesis (Allen and Smith 1996 and references therein). The theoretical power spectrum of a red-noise process is known and the parameters of the model are deduced from the analysed record. The power spectrum of the tide-gauge data was compared with that of the null-hypothesis, at a certain confidence level, and those peaks with power higher than that expected from the null-hypothesis have been defined as significant. Details are provided in chapter 10 of Alessio (2016).

Ref: Allen M, Smith L. 1996. Monte Carlo SSA: Detecting irregular oscillations in the presence of colored noise. Journal of Climate, 9(12): 3373–3404

Ref: Alessio SM. 2016. Digital signal processing and spectral analysis for scientists: concepts and applications. Springer.

*L490: The sunspot cycle. This seems a far-fetched link to me. How does the 11- year sunspot cycle cause significant local sea-level variations? I guess that this link is merely coincidence.*

In the spirit of a literature review, in the original manuscript we opted for including as much details as possible from the references quoted in the sentence. In the revised manuscript we would omit the link to 11-year solar variations (which is anyway dealt with in detail in the companion paper Lionello et al., 2020b) and only refer to the presence of significant decadal variability in the record of Venetian storm surges. The revised sentence would read as follows: "Lionello (2005), Barriopedro et al. (2010), Troccoli et al. (2012) and Martínez-Asensio et al. (2016) consistently identify significant decadal variability in the time series of autumn Venetian surge events for the period 1948-2008 (for an updated analysis see Lionello et al., 2020b, in this special issue)."

*L494: What is a 'statistically significant fluctuation'? Significant with respect to what? White noise? A linear trend? Why is a wavelet analysis a good tool to study this?*

Fluctuation refers to the amplitude of spectral components. Significance of observed spectral peaks was tested against a red-noise hypothesis, namely a lag-1 autoregressive model characterized by high power spectral density at lower frequencies. It is the most widely used model for geophysical purposes since a large class of geophysical processes produce output statistically compatible with the red noise hypothesis (Allen and Smith 1996 and references therein). According to Grinsted et al. (2004) the significance is computed through the distribution of an ensemble of surrogate series describing a red noise process with the lag-1 parameter and variance estimated from the analysed time series.

Wavelet transform is commonly applied since it is an evolutionary spectral method which allows to examine the spectral evolution of the analysed record in a time-frequency(period) domain and to find localized intermittent periodicities (Grinsted et al., 2004). Thanks to the multivariate version of this method, namely the Wavelet Coherence, also the coherence between two time series, at certain frequency bands, can be detected as done in this study.

For the sake of clarity and completeness, we would add the first paragraph of this response in the revised manuscript and modify the caption of figure 8 as follows:

"Shading (thick black contour) is the portion of the spectrum exceeding 90% (95%) confidence against red noise (lag-1 autoregressive model) hypothesis (see Grinsted et al., 2004, for details)"

Ref: Allen M, Smith L. 1996. Monte Carlo SSA: Detecting irregular oscillations in the presence of colored noise. Journal of Climate, 9(12): 3373–3404

*L500: This stationarity here has been attributed to Greenland and glaciers above, and here it is due to atmospheric processes. What is it?*

We explain the apparent discrepancy between the trend estimates and the conclusions obtained by Marcos and Tsimplis (2008) and Scarascia and Lionello (2013) partly as a result of the different focus region, partly to different period considered in the two studies and of the strong interdecadal variability observed in sea-level records in the Adriatic and Mediterranean seas, and partly due to the different consideration of the processes contributing to sea-level variability (see our response to the general comment above). It is among the conclusions of our review that there substantial

differences in the rate of sea-level change in Venice can be obtained if the analysis is conducted on periods of a few (say up to 5) decades. It is true that different trends in different periods can be determined by different mechanisms being active or anyway predominant. We are convinced that this will be clear in the revised manuscript, not only in reference to different periods, but also as far as the relation between Venetian sea level and the GMSL is concerned, as outlined in our responses above to the specific comments by the Reviewer in this regard.

*L505: What is "the integral of the absolute trend differences for bidecadal and shorter periods"?*

Thanks for pointing at this convoluted sentence. We referred to the sea-level difference, in absolute values, obtained by integrating (or summing up) through time the linear trend estimates for the GMSL and Venetian sea level over the given period of time, subtracting them for each other, and keeping the absolute value of such difference. It is meant to quantify the Venetian sea-level anomaly associated with the local linear trend with respect to the GMSL trend. We would revise the sentence as follows: "Accordingly, the linear trend can yield a local sea-level anomaly in Venice from the GMSL of about 10 cm (but up to about 20 cm occasionally) over bidecadal and shorter periods, and a rather small anomaly (generally <5 cm) over interdecadal and longer periods (not shown)."

*Sections 5.1 and 5.2*

*General: like section 4.3, these sections lack focus, tend to introduce a lot of different processes, but at the end, I still don't understand the relative importance of each process. Furthermore, a general introduction of the physics behind the processes is necessary here to understand what's going on. For example, something like: "The North Atlantic Oscillation causes large-scale atmospheric pressure variations on interannual scales. The geostrophic winds caused by these variations drive a barotropic sea-level response in Venice, especially in winter."*

In the revised version we would improve the description of physical mechanisms involved in sea-level variability, in a widely restructured Section 5. For the NAO, we plan to have a dedicated subsection, which will benefit from some introductory statements along the one proposed by the Reviewer. Please see our response to the more general comments by the Reviewer above, our response to the specific comments on the NAO below as well as our responses to the comments by Reviewer #1.

*L511: Steric effects. What about steric effects within the Mediterranean? Do they play an important role? They are discussed in the projections section, but what about observed changes? One could estimate the size of this effect over the last few decades from gridded hydrographic observations, such as EN4 (Good et al. 2013) or Ishii et al. (2017).*

We plan to better illustrate and discuss steric effects on Mediterranean sea-level change based on available literature (for instance Jordà and Gomis, 2013; Carillo et al., 2012; Scarascia and Lionello, 2012), also taking advantage of the substantial restructuring of Sections 5 and 6 in the revised manuscript.

Concerning the additional analyses suggested by the Reviewer, whereas we provide updated estimates for some components to sea-level variations, it is beyond the scope of this literature review paper to estimate the size of all individual effects to Mediterranean sea-level variability. We acknowledge that the availability of updated oceanic datasets as those reported by the Reviewer can

be valuable for better constraining the different contributions to sea-level variability within the Mediterranean Sea, and we plan to discuss this opportunity in Section 7.

*L520: What do the authors want to say in this paragraph?*

This part would be removed in the revised manuscript.

*L531: Geostrophic wind is something different than large-scale wind (https://en.wikipedia.org/wiki/Geostrophic_wind)*

We certainly agree. The sentence would be removed in the revised manuscript as it was confusing.

*L534: "Mass exchange can dominate: : :": Isn't NAO forcing just a local barotropic response to wind forcing and as such, just added on top of basin-wide fluctuations?*

In fact, the NAO has been associated with Mediterranean sea-level variability both as large-scale driver of winds over the Strait of Gibraltar, hence influencing water-mass exchange between Atlantic Ocean and Mediterranean Sea, and therefore basin-mean sea-level fluctuations in the Mediterranean, as well as driver of local wind anomalies within the Mediterranean, thereby contributing to spatial heterogeneity in sea-level variations within the basin. We plan to restructure the section and have a subsection devoted to the NAO-Venetian and Mediterranean sea-level connection, possibly with the addition of a figure illustrating changes in the pressure and wind fields over the Mediterranean region associated with different NAO phases to support the text.

*L539: What is "explained linearly"? L540: They: : :EAWR. This is vague. What happens under the hood here?*

The sentence would be removed from the revised manuscript

*L547 This link is also discussed above, but here, the notion that is link is not very strong is omitted. I'd just leave the sunspot studies out.*

Agreed.

*L553. I don't see this strong correlation from the wavelet coherence plot. The correlation looks weak to me: it does not hold throughput time, goes in and out of phase. From this result, I'd make the opposite conclusions, namely that there's only a weak correlation between winter sea level and the NAO. In for example Piecuch et al. (2019), a wavelet coherence plot that looks similar to this one (their Figure 1) is used to argue for a weak correlation.*

We overall agree that the NAO-Venetian sea level connection does not emerge as clearly  for both seasons as just briefly discussed in the quoted lines of the original manuscript. In fact the lack of a robust connection with the NAO especially in autumn agrees with Zanchettin et al. (2009), where other climatic modes such as the SCA and the EAWR appeared to be predominant. We would cite

Piecuch et al. (2019) and Zanchettin et al. (2016) to put caveats on the interpretation of coherences with variable phase through time. We would also discuss the results in the light of a new figure showing atmospheric circulation anomalies over the Euro-Mediterranean region around different NAO phases.

Ref: Zanchettin, D., Bothe, O., Rubino, A., and Jungclaus,J. H.: Multi-model ensemble analysis of Pacific and Atlantic SST variability in unperturbed climate simulations. Clim. Dyn., 47(3),1073-1090, doi:10.1007/s00382-015-2889-2, 2016.

*L555 In autumn: : : how significant is this statement? Given the difference between interannual NAO variability and multidecadal subsidence variability, I doubt whether you're just looking for an explanation of insignificant changes.*

As explained in the response to the previous comment, we overall agree with the Reviewer. We would therefore restructure the section substantially in the revised manuscript, also including an additional figure to illustrate the connection between NAO and Venetian sea level in autumn and winter. Our revised conclusion is that the NAO has a significant impact in winter whereas it exerts a less clear influence in autumn. In both autumn and winter, the patterns of sea-level pressure and wind anomalies under different NAO states superpose well on those that correspond to variations in Venetian sea level, including large-scale low pressures enhanced over the northern Mediterranean Sea and Sirocco-like northeastward wind anomalies over the Ionian Sea. However, amplitude and spatial extent of the significant anomalies in the atmospheric forcing fields are larger in winter compared to autumn, confirming the weak imprint of the NAO on Venetian sea level in the latter season.

*L558. The contribution of the IBE effect to sea level has been quantified way before 2009 as 1cm of sea-level rise to 1 mbar of pressure drop (see Wunsch & Stammer, 1997). Using regression, you might find other correlation coefficients between sea-level pressure and sea level, but that's because sea-level pressure and sea level interact in many more ways than just the inverse barometer effect. See for example Woodworth et al. 2010. It may also be a good idea to repeat the conclusions from Calafat et al. (2012) that the IB effect only explains a marginal fraction of variability (see their Figure 4).*

In the substantially restructured Section 5, we would briefly illustrate IBE with a paragraph along the one that follows: "Local atmospheric mechanical forcing is primarily exerted through local pressure anomalies, associated with the so-called Inverse Barometer Effect (IBE), and wind anomalies. The IBE is quantified by the hydrostatic equation in about 1 mm of sea-level rise per 1 mbar of sea-level pressure drop. Calafat et al. (2012) quantify in 25% the IBE contribution to decadal winter sea-level variability in Trieste for the period 1950-2009. The highest IBE contributions to seasonal Venetian sea-level variability over the period 1872-2003 in autumn (about 32%) and winter (41.5%) estimated by Zanchettin et al. (2009) can be explained by the regression model between local sea-level and local sea-level pressure which could embed also other contributions than IBE alone (e.g., Woodworth et al., 2010)."

*L563-574: What is the exact point of this paragraph? Hard to follow.*

This paragraph would be removed from the revised manuscript, with relevant parts of it embedded in different other paragraphs where appropriate.

*L574ff: Please carefully re-read Calafat et al. 2012: they discuss alongshore coastally-trapped wave propagation along the Atlantic coast, affecting Mediterranean Sea level as a whole. They do not look into the Adriatic Sea in particular.*

Indeed, thanks for catching this. The sentence would be amended in the revised manuscript.

*L584-585: Accordingly -variability. This is a vague sentence. What is an atmospheric bridge in this context, and what does "constitute a potential precursor to multidecadal variability" mean?*

The sentence refers to the possibility that the remote connection between multidecadal variability in Venetian sea-levels and North Atlantic climate is determined by atmospheric teleconnections, following the analysis between Euro-Mediterranean sea level pressure and the Atlantic Multidecadal Oscillation by Mariotti and Dell'Aquila (2012). Accordingly, atmospheric circulation would act as a bridge.

We plan to rephrase this in the revised manuscript within a restructured section 5, as follows: "In addition to the NAO, statistical connections identified in the literature between Venetian RSL and climatic modes include the atmospheric patterns known as Scandinavian and East Atlantic Western Russia (Zanchettin et al., 2009), showing prominent variability at interannual to decadal time scales in the autumn and winter seasons, and the Atlantic Multidecadal Oscillation or AMO (Scafetta (2014), describing multidecadal fluctuations in North Atlantic sea-surface temperature and influencing atmospheric variability over the Euro-Mediterranean region (e.g., Mariotti and Dell'Aquila, 2012; Maslova et al., 2017)."

*L586: "This could contribute to explaining the statistical connection between bidecadal variability of Venetian RSL". Or it could not. This statement needs some evidence, as it now looks like guesswork.*

We agree and our sentence was not meant to provide any conclusive evidence, rather point at a possibility. We plan to rephrase this part as illustrated in our response to the preceding comment.

*L588: What is "multi-scale acceleration analysis"?*

MSAA is the technique used by Scafetta (2014). We would omit this detail in the revised manuscript as non relevant.

*L589ff: This section contrasts with many of the cited papers above, or at least, it needs more explanation. It's namely very unlikely that ice melt can explain 1.3 mm/yr in the Mediterranean, due to the magnitude of past ice melt and GRD effects, causing the Mediterranean to be affected much less than the global mean. How about large-scale changes in the Atlantic Ocean that propagate into the Mediterranean?*

Scarascia and Lionello (2013) explain interannual VLM-corrected RSL by the combination of steric effect and the mechanical action of the atmosphere. The authors observe that these factors have no net trend for the period 1940-2005 and therefore cannot be used for explaining the SL trends. In this respect, there is no contradiction with other studies that are cited in section 5, which mostly describe the same variability with different perspectives. The new structure of section 5 will better separate the different factors and clarify the  lack of contradiction among studies. We agree that the

word "remote" is confusing and it should be replaced with "external to the Mediterranan Sea". We agree that the specific attribution to ice-cap melting is not supported by the cited study and it will be dropped in the revised version.

*L603: Lots of terms that need explanation: "amplitude modulation of water transport", "migration of the eastern hydraulic control".*

We would rephrase the first quoted text as "modulation of the water transport".

Then, there are two hydraulic controls in Gibraltar. One is located over the Camarinal Sill, the other is a moving hydraulic control that basically is locked in-phase with the bore propagation within the Tarifa Narrow. These details are not necessary, therefore we would simplify the sentence as follows: "Local dynamics are strongly influenced by tides, which are responsible for the modulation of the water transport and the hydraulic control (Armi and Farmer, 1988), as well as for the substantial vertical mixing that has been observed (Garcìa-Lafuente et al., 2013)."

*Section 5.2 In its current setup, this section reads like it has been written without a clear focus in mind. What message do the authors want to tell with this section? It now reads like an unconnected collection of papers that each describe an individual problem, but an overarching story is missing. What models are available, reanalyses, operational forecasts? A good starting point may be the model results from Fukumori et al. (2007).*

We will substantially restructure section 5 during the revision, and merge the sections regarding the physical mechanisms underlying Venetian sea-level variability and numerical modelling. As detailed above, we plan to have an introductory part and then three separate subsections dealing with the most important aspects of climate forcing of Mediterranean and Venetian sea level, which will embed any relevant aspect regarding associated numerical modelling issues.

*Section 6 Similar to the previous sections, this section is also somewhat unorganized and lacks focus. After reading it, I am unable to determine a conclusion from the section. Why don't the authors just use the SROCC projections? They should contain all the processes discussed here, except for the vertical land motion part. There is also a serious lack of information on the methods, and where methods are named, it feels like a bit of a grab-bag of individual estimates. I see SRES scenarios, AR5, SROCC, local projections... Are they combined in a consistent way? Where does the atmospheric forcing come from? Therefore, please thoroughly revise this section with a consistent treatment of scenarios and processes, together with a clear methods section.*

The feeling that this section - as others in this manuscript - is a collection of estimates from different authors, based on different data and methods, stems from the fact that this is meant to be a literature review. Our aim is to collect all relevant information and try to make sense of the differences in the conclusions reached in the various studies. We would carefully revise section 6 concerning content, structure and presentation, as outlined by our responses to the specific comments below. In particular, we would have a closer and clearer focus on projections for the local sea-level change in Venice, also taking advantage of the references suggested below by the Reviewer.

We will take extreme care in detailing the different contributions to Venetian relative sea-level rise and how they are combined in our estimate. For the mentioned atmospheric forcing, in particular,

we rely on published estimates indicating that associated sea-level variations are generally less than but up to about 10 cm (as stated in line 677-679 of the original manuscript). As there is no quantitative precise estimate for this contribution for the case of Venice or the Northern Adriatic, we would amend the text highlighting that further research is required in this regard and avoid including this specific contribution to our total estimate.

*L728: This statement is not true. There's no single process that causes a uniform sealevel rise. Therefore, each single process, from ice mass loss to GIA and sterodynamic effects causes a local deviation from GMSL. You might reach the conclusion that the resulting local changes are close to the global mean changes, but that's something different.*

We agree that, in its present form, the second part of this paragraph is confusing. An important conclusion of this review is that the Mediterraean mean sea level follows that of the midlatitude eastern North Atlantic on multidecadal time scales. However, changes over interdecadal periods can distort the detection of forced trends over rather long periods of time (e.g., Jordà, 2014). Further, regional atmospheric patterns can determine differences up to 10 cm between the RSL in the Mediterranean and in the midlatitude eastern North Atlantic and within different parts of the Mediterranean basin itself. Finally, RSL in Venice can differ as land movements (subsidence) could provide additional and important contributions at local scale. The text will be revised accordingly.

*Section 7 L736: Similar as in the previous sections: except for VLM, how different are sea-level variations at the coast and just off the coasts as measured by altimetry? Or in other words? Why would we need altimetry closer to the coast?*

Essentially, availability of altimetry data very close to the coast would be important for a more direct comparison with tide-gauge data. This would allow us to better understand why both tools provide different - though not mutually inconsistent - statistics, such as trends (see Table 4). As a further scientific motivation, altimetry is the only tool currently capable of measuring sea level from the open ocean to the coasts, hence allowing to assess whether coastal sea level is rising at the same rate as open-ocean sea level. An example is sea level near the coast of western Africa, where observed trends are significantly different than offshore (Marti et al., 2019). We would better clarify this in the revised manuscript and also better report the issue in the abstract as follows: "An unresolved issue is the contrast between the observational capacity of tide gauges and satellite altimetry, with the latter tool not providing reliable data within the Venice Lagoon yet. It is therefore currently not possible to take advantage of the full potential of along-track altimetric data in the inter-technique comparison."

Ref: Marti, F., Cazenave, A., Birol, F., Passaro, M., Léger, F., Niño, F., ... & Legeais, J. F. (2019). Altimetry-based sea level trends along the coasts of western Africa. Advances in Space Research, doi: 10.1016/j.asr.2019.05.033.

*L795: The method to determine the trend and uncertainties critically depends on the purpose: what should the trend encompass? That should set the method you want to use. For example, neither method gives you a number that should be extrapolated into the past or future.*

Here, we refer to the trend within its statistical definition. Following the International Statistical Institute (ISI, 2003), this would be the long-term movement in a time series, which may be regarded, together with the oscillation and random component, as generating the observed values. The issue, as stated in the manuscript, is meant to be exquisitely practical, as often in time series analysis the

long-term trend is removed. We are convinced that some general guidance is needed in this regard, hence this part of our manuscript.

In the spirit of a literature review, we highlight the determination of the statistical trend as an open issue, and only provide two exemplary statistical models that could be used to calculate the trend. We do this for two time series - the raw RSL series and the VLM-corrected RSL series - to highlight differences and possible caveats due to the different considered processes.

We agree that neither trend should be extrapolated, and we see no reference to such extrapolation in our manuscript.

In the revised manuscript we would better clarify the aim of this paragraph, first of all by determining what is meant here for "trend" as outlined above and, then by adding a sentence like the following: "The presence of substantial variations in Venetian RSL and VLM-corrected RSL multidecadal trends contributes to explain, together with methodological aspects in the calculation, the different estimates of secular average rate of sea-level rise obtained by different authors considering different periods."

Ref: ISI (The International Statistical Institute): The Oxford Dictionary of Statistical Terms. Yadolah Dodge (ed.), Oxford University Press, Oxford, 506 pp, ISBN 0-19-850994-4, 2003.

*L800: Similar to above: what does "the shape of the local RSL rise" encompass?*

We would rephrase this as "the shape of the trend in sea level", hoping that the meaning is sufficiently clear in the light of the changes to be implemented in the preceding sentences, as outlined in the response to the previous comment.

*L803: What is "energetic variability"?*

We would change this to "significant variability"

*L809: This is not the first attempt to create regional sea-level projections for Venice. They are for example in the AR5 and SROCC report, Kopp et al. 2014 and Slangen et al. 2012. All provide local RSL projections with uncertainties. In this respect, Kopp et al. (2014) should be discussed here, since it uses a statistical model to estimate and project land motion not related to GIA.*

We acknowledge the fundamental contribution of the Slangen et al. (2012), AR5 and SROCC to study sea-level changes and the implications of sea-level rise for low-lying islands, coasts and communities. Despite there is no explicit reference to sea-level projections for Venice in these references (while Venice is indeed mentioned in the SROCC in a few instances regarding analysis of the associated tide-gauge data), we agree that they are relevant for this study as they provide regional and local relative sea-level, from which, somehow, one could "pick-up" sea-level projection for Venice. In this sense, Slangen et al. (2012) is indeed a pioneering paper, and we would cite this in the revised manuscript.

Also, thanks for pointing at Kopp et al. (2014), which is indeed very relevant for our paper as it contains explicit projections for Venice. They use the global tide-gauge PSMSL dataset and assume that the recorded sea level is represented as the sum of three Gaussian processes, including (1) a globally uniform process, (2) a regionally varying, temporally linear process, and (3) a regionally

varying, temporally autocorrelated non-linear process for each tide-gauge site. The process (2) is retained as the "background non-climatic local sea-level change" corresponding to GIA, tectonics, and other non-climatic local effects. They then use this background linear estimates and its uncertainty for projections. In Venice, they estimate a background subsidence of 0.72 +/- 0.33 mm/yr with their technique (see their supplementary material, Table 8). The subsidence rates provided in Table 4 of our manuscript, which stems from various measurement methods, are overall higher than that by Kopp et al., which stems from a statistical method applied to tide-gauges record. Furthermore, Kopp et al. (2014) quote as a note of caution : "Third, our background rate estimates are the result of an algorithm applied to a global database of tide-gauge data, with different sites having been subjected to different degrees of quality control. Some tide-gauge sites may have experienced datum shifts or other local sources of errors not identified by the analysis. We recommend that users of projections for practical applications in specific regions scrutinize local tide-gauge records for such effects".

We would refer to Kopp et al. (2014) in the revised manuscript as follows: "In particular, Kopp et al. (2014) provide probabilistic sea-level projections for Venice as part of a global set of local sea-level projections for three different representative concentration pathways. Their projections build on the decomposition of the recorded historical sea level into a number of processes, among which is the "background non-climatic local sea-level change" corresponding to GIA, tectonics, and other non-climatic local effects. This background linear estimate and its uncertainty (0.72±0.33 mm/year for Venice, see supplementary material, Table 8) is then included in the projections, together with the other components. The 5th-95th percentile range of sea level change from their projections at year 2100 is 29-79 cm for RCP2.6 and 41-107 cm for RCP8.5."

Our projections largely overlap with those by Kopp et al. (2014), suggesting that they are overall consistent despite some differences that reflect the fact that methods, models and assumptions differ between the two studies. We would mention the need to understand such differences as an opportunity for progress in the revised manuscript.

We would use the following arguments to better motivate our own projections.

- firstly, in published literature the sterodynamic component in the Mediteranean basin is derived from CMIP3 or CMIP5 multi-model ensembles while it has been shown that CMIP model results in the Mediterranean basin have strong biases (see for instance a discussion in Thieblemont et al., 2019, notably their Figure 2). Accordingly, we rely on the lateral boundary forcing at Gibraltar as explained at line 710 of the original manuscript. We would better clarify this in the revised manuscript.

- second, as far as we understand, the papers reported by the Reviewer include regional/local estimates of vertical land motions in the sense that they account for some GIA contribution, intended as present-day viscoelastic response of the Earth's crust to changes in ice masses throughout the last glacial cycle, hence they do not provide a full characterization of vertical land motions in Venice, which is instead a salient point in the present study.

We would also revise the text by removing any reference to ours being the first attempt to develop local RSL change scenarios for Venice, and discuss differences between our projections and those obtained by Kopp et al. (2014) as outlined above.

*Figures*

*There is no reference to Figure 3.*

Thanks for picking this. Figure 3 is relevant for section 2.2, and we would refer to it as follows: "The results show that a reasonable increase in quantity and quality of data can be achieved compared to standard products up to a few kilometers from the coastline. Figure 3 illustrates the example of the Gulf of Trieste, where three missions cross the area and a data gap exists with standard products. In this case, the number of outliers along the Jason-1 and Jason-2 tracks is almost always less than the standard product and the improvement is clearly evident until 6 km from the coast."

*Figure 6: How have confidence intervals been determined? What noise model has been used?*

The trend and associated confidence intervals are obtained by linear regression analysis. The matlab function "regress" is used for the calculation. We plan to publish the script in a repository so the procedure should be clear. We are open to include more details in the revised manuscript in case the following clarification in the caption of the figure is deemed insufficient: "black contours illustrate where the GMSL and Venetian sea-level trend estimates do not overlap within 95% confidence intervals obtained from the linear regression [...]"

*Figure 7 as well: how have all the errors been computed? For panel B? Why use model data from CMIP3, while we have had CMIP5 and now CMIP6 has become available as well?*

Concerning panel A, the errors are 90% confidence level obtained from linear least-squares regression analysis. We would clarify this in the revised manuscript.

Concerning panel B, the plot is a replica of the original figure in Slangen et al. (2016). The uncertainties are the spread among the simulations with only differing Atlantic boundary conditions (blue) and the spread among the simulations with differing socio-economic scenarios (red). We would clarify this in the revised manuscript.

Concerning panel C, the uncertainties correspond to the combined uncertainty of each sea-level component (i.e. glaciers, ice-sheets, sterodynamic, ...) calculated as the square root of the sum of the squares of each component uncertainty. Note however that contributions that correlate with global air temperature have correlated uncertainties and are therefore added linearly (this concerns sterodynamic and ice-sheet surface mass balance components). See Church et al. (2013) for more details. We would clarify this in the revised manuscript.

We agree that the figure needs to be carefully described as it encompasses different aspects of the issue and different panels refers to different data and/or publications. We plan to amend this with improved caption and improved description in the text. We are aware that CMIP6 data are being made available, but an updated analysis on such data deserves a dedicated study beyond the current literature review. We would highlight this in the revised manuscript with a sentence along the following lines at the end of section 7: "Updated scenarios of Venetian sea-level future change are expected as output from the 6th phase of the Coupled Model Intercomparison Project is made available."

Ref: Church, J.A.; Clark, P.U.; Cazenave, A.; Gregory, J.M.; Jevrejeva, S.; Levermann, A.; Merrifield, M.A.; Milne, G.A.; Nerem, R.S.; Nunn, P.D.; et al. Sea Level Change. In Climate Change 2013: The Physical Science Basis; Contribution of Working Group I to the Fifth Assessment Report of the Intergovernmental Panel on Climate Change ed.; Cambridge University Press: Cambridge, UK, 2013.

*Figure 8. Middle row: how should I interpret this plot?*

The panels represent the wavelet spectra for the autumn (left) and winter (right) Venetian sea-level time series, after correction for subsidence. Since the interest is on portions of the wavelet spectrum where wavelet amplitude exceeds statistical significance (against a lag-1 red noise), we only show this and omit the representation of wavelet amplitude. Specifically, "Shading (thick black contour) is the portion of the wavelet spectrum exceeding 90% (95%) confidence against red noise (lag-1 autoregressive model) hypothesis (see Grinsted et al., 2004, for details)". We would report this in the caption.

*References to be added in revised manuscript*

*De Biasio, F., Baldin, G., and Vignudelli, S.: Revisiting Vertical Land Motion and Sea Level Trends in the Northeastern Adriatic Sea Using Satellite Altimetry and Tide Gauge Data. Journal of Marine Science and Engineering, 8(11), 949, 2020.*

*Calafat, F. M., Chambers, D. P., & Tsimplis, M. N. (2012). Mechanisms of decadal sea level variability in the eastern North Atlantic and the Mediterranean Sea. Journal of Geophysical Research: Oceans, 117(C9). https://doi.org/10.1029/2012JC008285*

*Dangendorf, S., Calafat, F. M., Arns, A., Wahl, T., Haigh, I. D., & Jensen, J. (2014). Mean sea level variability in the North Sea: Processes and implications. Journal of Geophysical Research: Oceans, 119(10), 6820–6841. https://doi.org/10.1002/2014JC009901*

*Dangendorf, S., Mudersbach, C., Wahl, T., & Jensen, J. (2013). Characteristics of intra-, inter-annual and decadal sea-level variability and the role of meteorological forcing: The long record of Cuxhaven. Ocean Dynamics, 63(2–3), 209–224. https://doi.org/10.1007/s10236-013-0598-0*

*Frederikse, T., & Gerkema, T. (2018). Multi-decadal variability in seasonal mean sea level along the North Sea coast. Ocean Science, 14(6), 1491–1501. https://doi.org/10.5194/os-14-1491-2018*

*Frederikse, T., Landerer, F., Caron, L., Adhikari, S., Parkes, D., Humphrey, V. W., Dangendorf, S., Hogarth, P., Zanna, L., Cheng, L., & Wu, Y.-H. (2020). The causes of sealevel rise since 1900. Nature, 584(7821), 393–397. https://doi.org/10.1038/s41586-020-2591-3*

*Frederikse, T., Riva, R., Kleinherenbrink, M., Wada, Y., van den Broeke, M., & Marzeion, B. (2016). Closing the sea level budget on a regional scale: Trends and variability on the Northwestern European continental shelf. Geophysical Research Letters, 43(20), 10,864-10,872. https://doi.org/10.1002/2016GL070750*

*Fukumori, I., Menemenlis, D., & Lee, T. (2007). A Near-Uniform Basin-Wide Sea Level Fluctuation of the Mediterranean Sea. Journal of Physical Oceanography, 37(2), 338–358. https://doi.org/10.1175/JPO3016.1*

*Good, S. A., Martin, M. J., & Rayner, N. A. (2013). EN4: Quality controlled ocean temperature and salinity profiles and monthly objective analyses with uncertainty estimates. Journal of Geophysical Research: Oceans, 118(12), 6704–6716. https://doi.org/10.1002/2013JC009067*

*Gregory, J. M., Griffies, S. M., Hughes, C. W., Lowe, J. A., Church, J. A., Fukimori, I., Gomez, N., Kopp, R. E., Landerer, F., Cozannet, G. L., Ponte, R. M., Stammer, D., Tamisiea, M. E., & van de Wal, R. S. W. (2019). Concepts and Terminology for Sea Level: Mean, Variability and Change, Both Local and Global. Surveys in Geophysics. https://doi.org/10.1007/s10712-019-09525-z*

*Ishii, M., Fukuda, Y., Hirahara, S., Yasui, S., Suzuki, T., & Sato, K. (2017). Accuracy of Global Upper Ocean Heat Content Estimation Expected from Present Observational Data Sets. SOLA, 13(0), 163–167. https://doi.org/10.2151/sola.2017-030*

*Kopp, R. E., Horton, R. M., Little, C. M., Mitrovica, J. X., Oppenheimer, M., Rasmussen, D. J., Strauss, B. H., & Tebaldi, C. (2014). Probabilistic 21st and 22nd century sea level projections at a global network of tide-gauge sites. Earth's Future, 2(8), 383–406. https://doi.org/10.1002/2014EF000239*

*Landerer, F. W., & Volkov, D. L. (2013). The anatomy of recent large sea level fluctuations in the Mediterranean Sea. Geophysical Research Letters, 40(3), 553–557. https://doi.org/10.1002/grl.50140*

*Lickley, M. J., Hay, C. C., Tamisiea, M. E., & Mitrovica, J. X. (2018). Bias in Estimates of Global Mean Sea Level Change Inferred from Satellite Altimetry. Journal of Climate, 31(13), 5263–5271. https://doi.org/10.1175/JCLI-D-18-0024.1*

*Piecuch, C. G., Huybers, P., Hay, C. C., Kemp, A. C., Little, C. M., Mitrovica, J. X., Ponte, R. M., & Tingley, M. P. (2018). Origin of spatial variation in US East Coast sea-level trends during 1900–2017. Nature, 564(7736), 400–404. https://doi.org/10.1038/s41586-018-0787-6*

*Piecuch, C. G., Dangendorf, S., Gawarkiewicz, G. G., Little, C. M., Ponte, R. M., & Yang, J. (2019). How is New England coastal sea level related to the Atlantic meridional overturning circulation at 26 N? Geophysical Research Letters, 2019GL083073. https://doi.org/10.1029/2019GL083073*

*Slangen, A. B. A., Katsman, C. A., van de Wal, R. S. W., Vermeersen, L. L. A., & Riva, R. E. M. (2012). Towards regional projections of twenty-first century sea-level change based on IPCC SRES scenarios. Climate Dynamics, 38(5–6), 1191–1209. https://doi.org/10.1007/s00382-011-1057-6*

*Volkov, D. L., Baringer, M., Smeed, D., Johns, W., & Landerer, F. W. (2019). Teleconnection between the Atlantic Meridional Overturning Circulation and Sea Level in the Mediterranean Sea. Journal of Climate, 32(3), 935–955. https://doi.org/10.1175/JCLID-18-0474.1*

*Wahl, T., Haigh, I. D.,Woodworth, P. L., Albrecht, F., Dillingh, D., Jensen, J., Nicholls, R. J., Weisse, R., & Wöppelmann, G. (2013). Observed mean sea level changes around the North Sea coastline from 1800 to present. Earth-Science Reviews, 124, 51–67. https://doi.org/10.1016/j.earscirev.2013.05.003*

*Woodworth, P. L., Pouvreau, N., & Wöppelmann, G. (2010). The gyre-scale circulation of the North Atlantic and sea level at Brest. Ocean Science, 6(1), 185–190. https://doi.org/10.5194/os-6-185-2010*

*Wunsch, C., & Stammer, D. (1997). Atmospheric loading and the oceanic "inverted barometer" effect. Reviews of Geophysics, 35(1), 79–107. https://doi.org/10.1029/96RG03037*

*Zanna, L., Khatiwala, S., Gregory, J. M., Ison, J., & Heimbach, P. (2019). Global reconstruction of historical ocean heat storage and transport. Proceedings of the National Academy of Sciences, 201808838. https://doi.org/10.1073/pnas.1808838115*

We would add all relevant references in the revised manuscript.

---

## Author Response (AR1)

**Response to reviewers**

**Reviewer 1**

We thank Reviewer 1 for their appreciation of our manuscript and for the useful comments and suggestions. In the following, we respond to the general as well as specific comments by the Reviewer, which are reported in italics (our response in normal font). We also describe how we implemented the associated changes in the manuscript during the revision.

*This is a comprehensive paper reviewing the present knowledge of sea level changes in Venice, including observations and mechanisms and also with some discussion about regional projections in the Northern Adriatic. The manuscript is complete and includes the state of the art in regional sea level in the Mediterranean basin, in addition to provide new computations that include the most recent data for Venice. One strength is the detailed description of vertical land movements, a major driver of relative sea level changes, at many different temporal scales and with emphasis of the distinct acting mechanisms. Another one is the summary of some of the main results in table format. In my opinion, this manuscript deserves publication and will likely become a main reference of sea level variations in Venice. Some parts of the paper are, however, confusing and I think should be reorganised. I am giving details on this latter comment below.*

Thank you for the overall positive evaluation of our review. We have improved the manuscript based on your recommendations as described in our responses to the specific comments below.

*General: The way section 5 is organised is confusing and, in my opinion, the separation into subsections is misleading. I am giving more details below but, essentially, I do not think that if the section is discussing climate forcing of MSL variability, it is convenient to separate the physical mechanisms from numerical modelling. I would suggest focussing on the mechanisms (namely the effect of atmospheric pressure and winds, the water mass exchanges through Gibraltar, surface fluxes) and discuss their spatial and temporal scales, together with the origin of the data, whether observations or models. Note that barotropic models are mentioned in 5.1.1 but separated from the section on numerical modelling. Overall, I think that this section would benefit from rewriting.*

We have substantially restructured section 5 in the revised manuscript and merged the subsections regarding the physical mechanisms underlying Venetian sea-level variability and numerical modelling. The section now includes an introductory general part and, then, three separate subsections dealing with the three most important aspects of climate forcing of Mediterranean and Venetian sea level as they emerge from the screened literature, namely:

5.1 Lateral boundary forcing at the Strait of Gibraltar

5.2 Air-sea interaction within the Mediterranean basin

5.3 Linkage with the NAO and other teleconnection patterns

Each section/subsection embeds relevant aspects regarding the associated numerical modelling. We have expanded the list of references building on the Reviewers' suggestion. Please refer to the revised manuscript with tracked changes for an overall appreciation of the extensive changes made to the section 5.

*Section 7 discusses many of the gaps of knowledge in Mediterranean physical oceanography. In particular, around lines 766-794, the focus is on the limited knowledge on ocean circulation and the impacts that thermohaline changes in the lateral Atlantic boundary. However, the impact of these processes on basin-scale sea level and in Venice is small in comparison to other effects. This is especially true when climate projections are concerned. Indeed, in section 6.2, the authors include a nice summary of regional projections and uncertainty ranges which are by far much larger than the impacts of regional circulation changes.*

The revised manuscript have better homogenized sections 5 and 6 in terms of the relevance of the different processes responsible for Venetian sea-level variations and better highlight, where necessary, why different focus is given in different parts of the manuscript. In the restructured Section 5 we separate processes that lead to Mediterranean basin-scale sea level changes from those that lead to spatial heterogeneity in sea-level anomalies within the Mediterranean, with a focus on the Adriatic Sea and Venice. We better confront thermosteric and mass contributions to sea-level, also in relation to the considered timescales. Again, we have expanded the list of references building on the Reviewers' suggestion. Please refer to the revised manuscript with tracked changes for an overall appreciation of the extensive changes made to the section 5.

*Throughout the paper, MSL is used to refer to geocentric MSL in contrast to RSL. But MSL can be computed from RSL from tide gauges as well as from geocentric sea surface height, following the definition here and in Gregory et al (2019). I suggest to add geocentric when referring to RSL corrected for vertical land movements (e.g. section 4.1) to avoid confusion.*

The subsidence correction of RSL series in Zerbini et al. (2017) was mainly based on time series of benchmark heights, obtained during levelling surveys, and are referred to the Zero of the Italian altimetric network in Genoa. Therefore, despite the Altimetric Zero in Genoa being stable (hence at a rather constant geocentric height), the corrected sea level is not strictly speaking geocentric. In the revised manuscript we therefore avoid using GSL to refer to VLM-corrected RSL. We have updated the nomenclature regarding sea level and use the following acronyms in the revised manuscript:

- Relative Sea Level (RSL) change: change in local sea level from the local solid surface (Gregory et al., 2019);

- Geocentric Sea Level (GSL) change: change in local sea level with respect to a geocentric reference, namely a Terrestrial Reference Frame or, equivalently, a reference ellipsoid (Gregory et al., 2019). The acronym GSL is therefore used for satellite altimetry sea-level data;

- Subsidence: land surface sinking (UNESCO, 2020; see also: Gregory et al., 2019);

- VLM-corrected RSL: local sea level derived from tide-gauge RSL data corrected for vertical land movements;

- Global-mean sea level (GMSL): spatially averaged sea level over the World Ocean.

*Specific comments: - Line 90: "from the open ocean to the coastal zone": strickly speaking this depends on the definition of coastal zone, as altimetry measurements are often only valid tenths of km offshore, where the land signal does not contaminate the observations.*

We totally agree with the Reviewer that satellite altimetry becomes challenging when approaching the coasts, as this is also discussed extensively in our original manuscript for the case of the Venice lagoon. To avoid confusion in this specific sentence, we have changed the quoted part to "with an almost global coverage" in the revised manuscript.

*- l. 118-119: how was the vertical datum determined to an accuracy of 0.1 mm?*

The vertical datum is the one reported in Dorigo (1961). It was defined from the computation of a 25-yr mean tide level and we have no information about the reason for adopting the 0.1-mm precision; it was not an accuracy because it did not come from a measurement. Perhaps (but it is just a speculation) the choice was made for consistency with the 0.1-mm accuracy, typical of benchmark levelling.

To clarify this point in the revised manuscript we modified the text as follows:

"According to Dorigo (1961), Mati established the tide gauge datum at Santo Stefano at 1.50 m below the CM of 1825. In 1910, a new datum was adopted, namely the mean tide level (MTL) of 1884-1909 (central year 1897), computed from the high and low waters measured at Santo Stefano. According to Dorigo (1961), it turned out to be 1.2754 m above the tide gauge datum and 0.2246 m below the CM of the year 1825. This new reference was named the "Zero Mareografico Punta Salute" (ZMPS)."

*- l. 171: determine->determines*

Thanks, corrected.

*- l. 225: b.s.l. -> below sea level (I guess)*

Yes, this is explicitly reported in the revised manuscript.

*- table 2: shouldn't GIA rates from section 3.1.3 be included in this table, for completion?*

Table 2 reports the time evolution of natural land subsidence in the Venetian region. GIA is just one of the components that contributes to vertical land motions, therefore we believe that it would be misleading to report GIA rates in the table. Therefore, we did not modify the table in this sense.

*- fig. 5: please add a-b-c-d labels to subplots*

Thanks, labels have been added in the revised manuscript.

*- l. 357: wrong reference to fig. 3*

Thanks, the correct reference is to Fig. 5, amended in the revised manuscript.

*- table 3: the value of subsidence reported for the period 2008-2020 is notably larger than 2003-2010 and 2014-2020 (the latter also from a GPS station), but no comments are provided in the text. Given the differences and the likely overestimation of the GPS VEN1 I think it is worth to mention it in the paper.*

This comment refers to the following estimates reported in Table 3:

| Time span | Value (mm/yr) | Technique |
|---|---|---|
| 2003-2010 | 1.0±0.7 | SAR |
| 2008-2020 | 1.7±0.5 | GPS station VEN1(Riva dei Sette Mari) |
| 2014-2020 | 0.9±0.6 | GPS station PSAL (Punta della Salute) |

These estimates result from different techniques and are not only referred to different periods of time, but also to different locations. As reported in the table's caption, the uncertainty associated with the SAR-derived rate represents the ground motion variability at the city scale. Therefore, the rates of the two GPS estimates are both consistent with the average value reported over the whole city. SAR has proven capable of detecting ground displacements at the single-building scale and the measured rates range between -10 and 2 mm/year (see lines 342-343 of the original manuscript). Besides that, one of the main purposes of Section 3 is to clarify that the subsidence behavior in Venice is nonlinear. Therefore, temporal variations are to be expected even at short time scales.

We discuss differences between estimates reported in Table 3 in the revised manuscript, along the line of this response to the Reviewer. Specifically, the revised text now reads:

"Table 3 presents the evolution of subsidence rates in the historical center of Venice, as measured by geodetic instruments over the last century. The estimates refer to different periods of time as well as different locations, and they result from different techniques, for each of which associated uncertainty is available in the literature. Precise leveling allows measuring height differences with a mean error ranging from 0.3 to 1 mm in a line of 1 km (Torge and Müller, 1980). The average uncertainty for the vertical component of the GNSS trends is in the order of 0.3 mm/year for time series spanning a decade or more (Santamaría-Gómez et al., 2017). This estimate increases to 0.4 mm/year when reference frame uncertainties are considered in the error budget (Santamaría-Gómez et al., 2017). Finally, Tosi et al. (2013) propose to present the results of the SAR technique over a selected area in terms of average rate and standard deviation of the spatial variability. By doing so, the technique provides insights on the representativeness of the estimated trend at the investigated spatial scale. Therefore, the two GPS-based trend estimates reported in Table 3 are both consistent with the average value obtained by the SAR technique over the whole city."

*- l. 402-404: please provide a reference for the value in Ravenna.*

The reference is the same used for Venice, i.e., Zerbini et al. (2017). The sentence has been amended accordingly in the revised manuscript as follows:

"In Zerbini et al. (2017), the application of the same procedure to the neighboring tide gauge of Marina di Ravenna, also located in a rapidly subsiding area, provided a consistent estimate of 1.22±0.32 mm/year (period 1896-2012)."

*- Fig. 8 and l. 550-555: it is surprising the differences between fall and winter wavelets and coherence RSL-NAO. Generally, extended winter (Dec-March) NAO is used to correlated with RSL, since it is during these months when the signal is stronger. It is probably worth to comment on the differences found here.*

We agree that the NAO is more stable in winter and therefore the connection between NAO and Venetian sea level is more significant during this period of the year, whereas other modes of large-scale atmospheric variability become more important than the NAO in other seasons. In fact, this was also shown in Zanchettin et al. (2009). In our manuscript, the inclusion of an analysis of autumn series was mainly motivated due to the relevance of this season for storm surges in the Venice lagoon. Also following a comment by Reviewer #2, we better stress in the revised manuscript the seasonal difference in the NAO imprint on Venetian sea level. Our responses to Reviewer #2 provide more details about how we have revised the manuscript regarding the NAO.

In particular, the revised manuscript includes a specific subsection dedicated to the NAO in the revised manuscript (section 5.3 "Linkage with the NAO and other teleconnection patterns"), which contains general introductory information about the NAO, and a more critical discussion about the results from the wavelet coherence analysis. The revised manuscript includes a new figure (Figure 8) illustrating maps of sea-level pressure and near-surface wind anomalies linked to different phases of the NAO, which aids the discussion about the atmospheric mechanical forcing of Venetian sea level associated with the NAO. Overall, we are convinced that the new section 5.3 in the revised manuscript provides a much clearer presentation of what is known about the connection between Mediterranean and Venetian sea level and the NAO.

Please see the revised manuscript for the extensive changes made to section 5.

*- section 5.1.1 discusses the effect of atmospheric pressure and winds. Yet, the title "atmospheric forcing" is misleading as it might also include heat/water flux exchanges. These processes are then discussed in section 5.1.2 instead. I suggest merging both sub-sections.*

As outlined above, we have revised the manuscript to better structure section 5. The subsections mentioned in this comment by the Reviewer have been removed, and their content merged. The revised section 5 presents local atmospheric mechanical forcing (IBE and wind forcing) and ocean-atmosphere surface fluxes as different aspects in a dedicated subsection within section 5 (new section 5.2, see above). To better present local atmospheric forcing of Venetian sea level, we have added a figure (Figure 8) illustrating the Mediterranean sea-level pressure and near-surface wind anomalous patterns associated to Venetian sea-level anomalies.

*- section 5.2 on numerical modelling of the Mediterranean sea level is definitely too short. The authors should also describe other numerical simulations that are available and that provide sea surface height as an output, as well as the community effort developed by MedCORDEX in which the ocean component is very strong. For the former case, one prominent example is the recent 3D reanalysis by Simoncelli et al (2017) available through CMEMS (https://resources.marine.copernicus.eu/?option=com_csw&view=details&product_id=MEDS EA_REANALYSIS_PHYS_006_Also, section 5.2 refers only to 3D numerical modelling,*

*while does not mention vertically integrated numerical models that have contributed in the past to unveil the role of atmospheric pressure and wind. Some of these works are discussed in section 5.1.1.*

*The term coastal sea level in the title is not particularly addressed either.*

As replied above to the general comment by the Reviewer, in the restructuring of Section 5 we got rid of the original subsection on numerical modelling. We avoid in the revised manuscript to list model types and initiatives regarding the Mediterranean Sea and instead embed the relevant parts of section 5.2 of the original manuscript along the discussion of the physical mechanisms responsible for variations in Venetian sea level. Reference to models of different complexity should then become more straightforward.

We still mention MedCORDEX and the availability of regional reanalysis in Section 6 (gaps of knowledge and opportunities for progress) as follows: "Concerning the simulation of Mediterranean oceanic circulation, considerable efforts have been invested over recent years into developing and applying regional climate and ocean circulation models approaching the issue of dynamical downscaling from different perspectives (e.g., Somot et al., 2008; Sannino et al., 2009; Artale et al., 2010; Naranjo et al., 2014; Sein et al., 2015; Turuncoglu and Sannino, 2017; Androsov et al., 2019; Palma et al., 2020). This included coordinated international regional climate modeling activities (e.g., MedCORDEX) and regional reanalysis for the Mediterranean Sea obtained through numerical simulations with data assimilation (Simoncelli et al., 2014)."

*- L. 632-634: GMSL and regional deviations from the global mean, instead of regional sea level changes. This distinction would come up naturally if the mechanisms are described in terms of their spatio-temporal variability in section 5.*

This text regarding this comment have been changed during the revision of the manuscript.

*- l. 641: where does the range 0.6-1 mm/yr comes from? The numbers discussed above are over 1 mm/yr for late Holocene natural rates of subsidence only.*

The smaller rate was proposed in old studies (e.g., Carbognin et al., 1976), whereas more recent estimates are more consistently in the order of 1 mm/yr. We have amended this in the revised version and only refer to the current estimates (hence1 mm/yr).

*- l. 800-802: worth mentioning other non-parametric methods such as Empirical Mode Decomposition or Singular Spectrum Analysis for computing time-varying rates of RSL change.*

We mention both techniques in the revised manuscript: "Further research (e.g., exploiting techniques such as empirical mode decomposition and singular spectrum analysis) is therefore needed …".

*- l. 827: the range given for projected MSL here of 21-100 cm by 2100 should be framed into the corresponding RCPs. Otherwise can be wrongly interpreted. The climate scenario is one of the major sources of uncertainty in projections by 2100, so I think it would be better to state the numbers for the scenarios considered: 32-62 cm under RCP2.6 up to 58-110 cm under RCP8.5, including subsidence. This is important because it helps to interpret that the likelihood of the lower bound is different from than in the upper bound.*

We have explicited the scenarios in the revised manuscript, as follows: "Projected climatically-induced Venetian sea-level rise from estimates for the GMSL corrected for the regional redistribution of mass contribution components such as glacier, ice-sheet and groundwater is in the range from 21 to 52 (from 48 to 100) centimeters by 2100 for the RCP2.6 (RCP5.8) socio-economic scenario. Ice-sheet melting provides a highly uncertain contribution. A RSL rise in Venice exceeding 180 cm by 2100 is an unlikely but plausible high-end change under strong melting of Greenland and Antarctica. […]"

**Reviewer 2**

We thank Reviewer 2 for their appreciation of our manuscript and for the useful comments and suggestions. In the following, we respond to the general as well as specific comments by the Reviewer, which are reported in italics (our response in normal font). We also describe how we have implemented the associated changes in the manuscript during the revision.

*Review of "Sea-level rise in Venice: historic and future trends"*

*This paper is a review paper about sea-level changes around the city of Venice, Italy. It discusses the observed changes, its relation to land motion and atmospheric forcing, and it provides projections of future sea-level changes for various scenarios. I enjoyed reading the parts on sea-level observations and land movements. These sections give a clear overview on the subject and I think that these sections are an important contribution to the existing literature. Especially the derivation of long-term land motion, its connection to shorter records from GPS and inSAR, and its temporal variability are very insightful and can function as a blueprint for similar studies in other regions.*

*However, in my opinion, the sections on sea-level variations, atmospheric forcing and ocean dynamics are sub-par compared to the other sections and need a lot of work before they are in a publishable state. These sections introduce a lot of different processes, but it feels like the coherence between these processes is missing. Also, the mutual consistency between the studies is not discussed (notably missing around L589). At the end, I wonder what is the relative importance of each process. Also, a lot of processes are introduced, but the physics needed to understand how these processes has been left out, while insight in these physics is necessary to understand the role and spatial coherence of these processes. That leads to odd situations, such as with the NAO, which is listed as one of the main processes, but which is not connected to wind and pressure fluctuations, while in reality, these processes are intrinsically linked. Also, there's a disconnect between the processes discussed in Section 5 and the projections in Section 6.*

In retrospective, we agree that sections 5 and 6 could be substantially improved. We have changed both sections during the revision include a general restructuring, especially for section 5. We better emphasize consistencies and inconsistencies between published results, and better link statistical connections with associated physical mechanisms.

For the specific comment on the results illustrated around line 589 (Scarascia and Lionello, 2013) and their apparent inconsistency with those illustrated around line 500 (Marcos and Tsimplis, 2008), we present both studies together in the revised manuscript within section 4.3 and clarify the reasons for the different conclusions they reach. The revised text reads as follows: "Marcos and Tsimplis (2008) estimated RSL trends to be zero (within the errors) in the 1960-2000 period at the tide-gauge stations of Trieste, Genoa and Marseille. So, stationary sea level characterized the whole Mediterranean Sea during this period but not the Atlantic Ocean, and proposed explanations include an atmospheric contribution mainly consisting of persistent high pressure and an oceanic contribution due to steric changes in deep water masses (Tsimplis and Baker, 2000; Tsimplis et al., 2005; Gomis et al., 2008). Scarascia and Lionello (2013) conclude that the mean sea-level rise of 1.3 mm/year in the Northern Adriatic in the period 1940-2005 is primarily due to ice cap melting, with no trends in atmospheric drivers, upper ocean temperature and surface layer salinity. The different estimates and attribution of multidecadal trends by Marcos and Tsimplis (2008) and Scarascia and Lionello (2013) reflect the strong interdecadal variability of Mediterranean sea level and the variety of its driving mechanisms. The former study computes the expansion of

the water column missing the contribution of the redistribution of mass and produces negligible values of sea-level rise in shallow water areas, while the latter accounts for it and explicitly considers the Adriatic Sea.".

Regarding the connection between NAO and Venetian sea level, we have restructured Section 5 to have one specific subsection on the NAO (subsection 5.3), after the physical processes have been introduced. The new structure of Section 5 is:

5. Climatic drivers of Venetian sea-level fluctuations

5.1 Lateral boundary forcing at the Strait of Gibraltar

5.2 Air-sea interaction within the Mediterranean  basin

5.3 Linkage with the NAO and other teleconnection patterns

The role of IBE and wind setup for the NAO connection was already mentioned in the original manuscript (lines 531-537). To better support this connection, in addition to changes in the structure of the section we have added a figure, new Figure 8, showing surface wind and sea-level pressure anomalies over the Euro-Mediterranean region linked to Venetian sea level variations as well as with different phases of the NAO.

Regarding the connection between processes discussed in section 5 and their relevance for section 6, in the revised version we have enhanced references to section 5 in section 6 as far as possible, making sure that comparable relevance is given to each process in section 5 and 6.

*A possible starting point to re-structure and clarify these sections could be to first discuss basin-wide fluctuations in the Mediterranean Sea at various time scales and how they are linked to sea-level variations in the North Atlantic Ocean. These changes and their linkage to the Atlantic Ocean are for example discussed in Fukumori et al (2007), Calafat et al. (2012), Volkov et al. (2019), Landerer & Volkov (2013). As far as I'm aware, almost all decadal and multi-decadal dynamic sea-level variability, as well as longer-term trends in Venice can be explained by basin-wide fluctuations in the Mediterranean, which are in turn linked to alongshore wind forcing along the East coast of the North Atlantic. With such a link established, it will be much easier to couple this knowledge to projections of sterodynamic sea-level changes from CMIP-style climate models. The second step then would be to determine which processes cause significant sea level anomalies in Venice relative to Mediterranean-averaged changes, for example due to local atmospheric forcing or local sterodynamic effects.*

Thanks for the suggestion and for pointing at these papers, which we have included in the revised manuscript. Reference to this publications was part of the substantial restructuring of section 5 following the Reviewer's recommendation.

*Some papers that could be useful as examples for a more structured description of dynamic sea-level variations on various temporal and spatial scales are Calafat et al. 2012 who explicitly shows for each process the amount of explained variability, or Wahl et al. 2013, Dangendorf et al. 2013, 2014, Frederikse et al. 2016, 2019 for papers investigating the dynamics that affect sea-level variability in the North Sea. Another recent example is Piecuch et al. (2019). Therefore, I recommend to thoroughly revise and rewrite the sections on atmosphere and ocean dynamics before the paper can be published.*

Thanks for pointing at these papers, which we looked at carefully and got inspiration for the revision of our manuscript. We are confident that the revised manuscript and especially the

substantially restructured and rewritten section 5 as outlined in this response are suitable for publication.

*Another possible way forward could be to just limit this review to the sections on vertical land motion and subsidence, which on their own would already be a very useful contribution to the literature. I have brought up a lot of points for sections 4, 5, and 6, but nevertheless, I hope the authors find them useful, despite their sheer number.*

We believe that sections 4, 5 and 6 are a necessary part of this literature review. We are confident that the changes we have implemented in the revised manuscript and illustrated in this response will convince the Reviewer that the paper is worth being published with all its components.

*Detailed and line-by-line comments:*

*At first, I'd recommend to double-check the definitions from Gregory et al. (2019) throughout the paper. For example, there's an awkward separation between MSL and RSL throughout the paper. I'd recommend getting rid of MSL altogether to avoid any ambiguity and use RSL for sea-level changes relative to land (tide gauge observations) and use GSL (geocentric sea level) for sea-level changes observed by altimetry and tide gauges corrected for vertical land motion. Also please check the definitions of the inverse barometer effect. That effect only includes the static response to atmospheric pressure variations (1 HPa of pressure drop causes a 10 mm sea-level rise), but does not fully represent the sea-level response to pressure changes (Wunsch & Stammer, 1997).*

The subsidence correction of the RSL series in Zerbini et al. (2017) was mainly based on time series of benchmark heights, obtained during levelling surveys, and are referred to the Zero of the Italian altimetric network in Genoa. Therefore, despite the Altimetric Zero in Genoa being stable (hence at a rather constant geocentric height), the corrected sea level is not strictly speaking geocentric. We would therefore avoid using GSL to refer to VLM-corrected RSL. We have updated the nomenclature regarding sea level and use the following acronyms:

- Relative Sea Level (RSL) change: change in local sea level relative to the local solid surface (Gregory et al., 2019);

- Geocentric Sea Level (GSL) change: change in local sea level with respect to a geocentric reference, namely a Terrestrial Reference Frame or, equivalently, a reference ellipsoid (Gregory et al., 2019). The acronym GSL is therefore used for satellite altimetry sea-level data;

- Subsidence: land surface sinking (UNESCO, 2020; see also: Gregory et al., 2019);

- VLM-corrected RSL: local sea level derived from tide-gauge RSL data corrected for vertical land movements;

- Global-mean sea level (GMSL): spatially averaged sea level over the World Ocean.

*Also a point on the significance on numbers, but I admit this is a bit a matter of personal taste: in an expression like 1.230.13, the last numbers are not significant, so something like 1.20.1 avoids a false sense of accuracy.*

We understand this comment to refer in particular to section 2, where indeed some reported values prospect a high accuracy (as also reported by Reviewer #1). We clarify that when reporting results from published papers, we kept the original number provided by the authors. In the revised manuscript we have amended the text so that it is clear that this is the case. Also, the numbers provided by our own calculations were carefully checked in that they provide a true sense of accuracy.

*Finally, I encourage the authors to deposit all scripts and data resulting from this paper in a public repository, such as Zenodo, Github, or Figshare.*

Most of the data used in this review article is already published and, except for figures 7b, 8 and 9, figures are redrawn. We have prepared two relevant datasets to be uploaded in a repository, as these are the key updated or new data used in the paper. These are: (i) the autumn and winter average RSL and VLM-corrected RSL for the historical record and (ii) the VLM-corrected RSL projections.

*Abstract*

*L39 "An unresolved issue": do you just want to say here that altimetry doesn't measure sea level in the Venice Lagoon because there's no track overlapping ground track?*

The issue is the contrast between tide gauge and satellite altimetry data. As far as the Venice Lagoon is concerned, the issue is twofold. First, the altimeter may not cross the lagoon depending on the ground track configuration design. TOPEX/Poseidon and Jason series do not cross the lagoon. Sentinel-2b instead crosses the lagoon, so data could be recovered if we improve the processing. As detailed in the main text, the Venice Lagoon is a challenging target due to several factors, among which are presence of land and specular reflections, and efforts to retrieve data are ongoing. To avoid misinterpretation or confusion, in the revised manuscript we have shortened this part of the abstract which now reads "Unfortunately, satellite altimetry does not provide reliable sea-level data within the Venice Lagoon, so this data source cannot be evaluated yet."

*L40 Water mass exchange: I think this gives the false impression that the issue lies within understanding what's going on at Gibraltar. The real issue here is what's happening to the Northeastern Atlantic Ocean, which directly drives this water mass exchange, see for example (Volkov et al, Calafat et al.).*

In fact, to better estimate Mediterranean sea-level variations under future global warming scenarios it is important to better understand and simulate both, the water mass exchange within the Strait of Gibraltar and precursors of its variability. To better stress this in the abstract, in the revised manuscript we have changed the sentence as follows: "Water mass exchange through the Strait of Gibraltar and its drivers currently constitute a source of substantial uncertainty for estimating future deviations of the Mediterranean mean sea-level trend from the global-mean value. Regional atmospheric and oceanic processes will likely contribute significant interannual and interdecadal future variability of Venetian sea level with a magnitude comparable to that observed in the past."

*L42: Subsidence and regional: : : and beyond. These sentences are valid for each and every coastal location, so it's rather trivial. In fact, there's no single known process that causes a true spatially homogeneous sea-level rise. What would be useful here is to explain which processes will cause large deviations from GMSL on various time scales.*

We agree this was a rather general sentence. We dropped this in the revised version.

*L45: "non-negligible differential trends": this is strange wording.*

We agree. We skipped the quoted sentence.

*Section 1*

*L52: Remove 'critically'*

Done

*L65: As said above, please avoid the term 'MSL' here and throughout the manuscript, and just distinguish between GSL (not ASL, Gregory et al. 2019) and RSL.*

Done

*Section 2*

*L129: The altimetry era actually started one year earlier with ERS1 (and even before that with SEASAT).*

We agree that the United States were the first to fly a satellite-borne altimeter, with the Skylab and Geos3 missions, Seasat in 1978 (the first satellite to provide data) and Geosat in 1985. Seasat had only 110-day lifetime (end of mission due to a malfunction). Topex/Poseidn was launched in August 1992. Ers-1 was launched in 1991 but the revisiting was initially 3 days and not exploitable for sea-level studies the ground track coverage is too coarse. The 35-day phase started in 1992. The continuous altimeter era is considered starting on 1992. In the revised manuscript we avoid reference to the beginning of the satellite era and modify the sentence as follows: "Since the first satellite altimetry missions in the mid-1970s, the accuracy of sea-surface height estimates has increased considerably until high-precision and routinely measured altimetric data were made available in the early 1990s with the launch of the TOPEX/Poseidon mission."

*L140. One question that is hanging over this section is, to which extent will sea level variations in Venice deviate from sea level a few km off the coast? My guess would be that that effect is rather minimal, except for very short temporal scales (like tides, waves and surges). This is especially relevant, as a few lines further down the paper, altimetry is directly validated with tide-gauge data, under the assumption that both would measure the same sea-level variability. The answer to this question also circles back to the open challenge described in the abstract, about the challenge of getting altimetry observations as close as possible to the coastline. If these variations are about the same as variations a few km's offshore, why worry about getting closer to the coast?*

We agree that sea level in the Northern Adriatic is rather homogeneous. The non-trivial point is to determine how sea level signals in the Northern Adriatic propagate into and then within the Venice Lagoon, for which satellite altimetry data are not available yet. As detailed later in section 4.2, the deviation between sea level in Venice measured by tide gauges and in the open sea in the vicinity to the lagoon measured by satellite altimetry provide estimates of interdecadal sea-level trends that do overlap indeed within uncertainties, but they do so only marginally after the tide-gauge data are corrected for vertical land movements, for which the estimate is about half that from satellite data. This motivated the open issue pointed out in the abstract. In the revised manuscript, we have included this sentence in section 7: "Overall, progress is expected in terms of improving the altimetry estimates closer to the coast, perhaps with different instruments, and understanding the relationship between the coastal and close-to-the coast sea-level changes in a complex environment (De Biasio et al., 2020)."

*Section 3*

*L197: For non-paleo readers, please add an approximate date of MIS 5.5*
Agreed, in the revised manuscript we specify "The following sections illustrate the evolution of the natural component of subsidence using the Marine Isotope Stage (MIS) 5.5 event (~130-120 kyr ago) as a reference to separate geologically older and newer RSL changes. Due to its relevance within geophysical studies on sea-level variations, a dedicated section on GIA is also provided."

*L269: Your current definition of GIA encompasses both the response to past and contemporary ice mass changes. Following Gregory et al. 2019, it might be a better idea to use GIA for the response to past ice changes, and use 'contemporary GRD effects' for the local sea-level response to contemporary ice-mass changes.*

We agree that there could be some confusion. In the revised manuscript we only use "GIA" to refer to responses to past ice changes.

*L365: Please define 'MOSE'.*

We added the definition of MOSE in the revised manuscript: "(so-called "MOdulo Sperimentale Elettromeccanico" or MOSE, see Lionello et al., 2020a)"

*Paragraph 4.1*

*General: in this section, the authors try to determine the secular trend in sea level in Venice and compare this trend to the global mean. There are a few fundamental problems with this section:*

*1. What do the authors mean by 'secular trend'? Is there some linear background trend? And if so, which processes are meant to be represented by this trend? To my knowledge, except GIA and maybe some other long-term geologic processes, no single process could cause such a trend. Processes like ice mass loss and thermal expansion are far from linear over 100-year time scales. The idea that this trend has to do with climate change is reinforced by the 'climate component' label in Table 6. I also wonder whether the different numbers found in Table 5 are just caused by computing linear trends over different time spans. Given that*

*sea-level changes often show a lot of multi-decadal variability, even small changes of the period over which the trends are computed can lead to differences in the range of the differences shown in Table 5. In line 400, the authors make a statement about the non-linearity due to subsidence. This argument is valid for most other processes as well.*

Following the aim of this literature review, this section summarizes published estimates of the observed rate of sea-level rise in Venice, and highlights and attempts to explain differences between them. We certainly agree that such estimates should not imply that the underlying process is necessarily linear, despite estimates being often obtained from the application of linear statistical techniques. In the revised version of the manuscript we changed "secular trend" with "Average rates of sea-level rise over centennial periods" and added the following text in the introductory part of section 4: "Average rates of sea-level rise are often calculated using some linear fit to the available data. However, given the variety and non-linearity of the processes known to contribute to sea-level trends on the considered time scales, such estimates should not be intended as necessarily representing a linear process (see Section 6)."

We also certainly agree that there is substantial multidecadal variability in the sea-level records, which can explain differences between estimates of the centennial rate of sea level rise provided by different authors (e.g., the apparent discrepancy between the conclusions of Marcos and Tsimplis, 2008, and Scarascia and Lionello, 2013, quoted above). We better state this in the revised discussion as follows: "The presence of substantial variations in multidecadal trends of sea level in Venice, together with methodological differences, explains the wide range of estimates of average sea-level rise obtained by different authors considering different periods. Still, certain aspects of the forcing of Venetian sea-level variability requires improved quantifications and characterization (for instance regarding the atmospheric forcing of local sea-level variability), and this will better constrain the range of projected future changes."

*2. The comparison to GMSL. The authors compare the trend in Venice to GMSL. This is generally not a good comparison for multiple reasons: due to its proximity to the Greenland Ice Sheet and many glaciated regions, their contributions (which together explain more than half of observed GMSL 1900-2018, Frederikse et al. (2020)) to Venice RSL will be much smaller than to GMSL. On the other hand, the Atlantic Ocean is accumulating heat at a higher rate than the Pacific and Indian Oceans (e.g. Zanna et al. 2019), so the steric component won't follow the global mean either. Therefore, the fact that Venetian and global sea level show similar trends is merely a coincidence, and you cannot expect these numbers to be comparable. Thus, expecting consistency with GMSL doesn't make sense. In fact, it may lead to the false assumption that future sea level rise in Venice will be comparable to GMSL rise. Since various contributors to GMSL rise vary over time as well, this coincidence may not hold. The consistency noted in line 436-437 reinforces this issue: the contributions from Greenland and glaciers in the second half of the 20th century are much smaller than for the first half of the 20th century. Therefore, they do not explain the deviation to be especially large in the second half of the 20th century. In contrary: if it were due to the aforementioned ice melt, one would expect a large difference around the 1930s and a smaller one over the second half of the 20th century.*

We agree with this comment. In fact, we did not expect consistency to be necessary between GMSL and sea-level variations in Venice. Our analysis was indeed motivated by the fact that available sea-level rise projections for Venice are in some cases directly based upon estimates of the GMSL rise (see, for instance, Troccoli et al., 2012, and Carbogning et al., 2010). Possibly, this is better explained in the accompanying editorial to the special issue (Lionello et al., 2020, discussion paper available at:

https://nhess.copernicus.org/preprints/nhess-2020-367/), where Venetian sea level is more appropriately compared to the midlatitude eastern North Atlantic sea level.

We kept the comparison between GMSL and Venetian sea level trends in the revised manuscript, but following the Reviewer's comment added an introductory sentence to explain the rationale of the comparison as follows:

"As illustrated in Sections 5 and 6, Mediterranean sea level, hence Venetian sea-level variations are tightly connected to sea-level variations in the midlatitude eastern North Atlantic, whose underlying processes differ from those in other oceanic basins. Therefore, any statistical consistency between historical Venetian RSL/VLM-corrected RSL and GMSL rise should not give the false impression that both variables are interchangeable and that any consistency in the historical period necessarily holds in the future. Still, it is instructive to compare estimates of Venetian RSL/VLM-corrected RSL and GMSL rise during the 20th Century to put local changes in the context of global mean changes."

We also refer again to the comparison in the restructured Section 5 (please see the revised text for the extensive changes implemented in this section).

*Line 405: What are 'secular tide-gauge records'? Also, the stations listed here might not be affected by large local subsidence the way Venice is, but I don't think they're unaffected by GIA-induced VLM. Most of these stations have some GNSS records available as well, which can be used to assess whether these stations don't show any VLM.*

By "secular tide-gauge records" we meant records spanning over one century or more. The precise time spans covered by each time series are indicated in the text (at lines 400 for Venice and 404 for M. di Ravenna). The wording was revised by replacing "secular" with "centennial" or "spanning over a century".

The stability of the tide gauges included in this discussion is provided in the references reported at line 408 of the original manuscript. These include (also) GPS-based results, see in particular Sanchez et al., 2018 (their Figures 8 and 14).

*Line 429. The trends denoted in the IPCC report and Hay and al. refer to global-mean relative sea level, and not global-mean geocentric sea level (See Gregory et al. 2019), although the difference between both is probably not very large. However, on local scale, the difference can be substantial, even when ignoring local VLM effects. See for example Lickley et al. 2018. Therefore, comparing local geocentric sea level to global-mean relative sea level is not fair.*

We agree that there might be substantial difference between local and global-mean sea-level estimates. Obviously, we also agree that geocentric and relative estimates of sea-level variations can differ, this is the rationale for the whole discussion provided in Sections 2 and 3. However, the following points should be considered:

1) The subsidence correction of RSL series in Zerbini et al. (2017) was mainly based on time series of benchmark heights, obtained during levelling surveys, and are referred to the Zero of the Italian altimetric network in Genoa. Therefore, despite the Altimetric Zero in Genoa is stable (hence at a rather constant geocentric height), the corrected sea level is not strictly speaking geocentric;
2) In fact, the focus of Zerbini et al. (2017) is regional rather than local as six tide-gauges characterized by centennial time series along the coasts of the Mediterranean Sea were considered;

3) hopefully, the impact of VLM on GMSL from tide-gauge records is limited. Over the altimetric period, in fact, the GMSL rise derived from tide-gauge records is consistent with altimetric measurements (e.g. IPCC 5th report, Chapt. 13, Fig. 13.7).

In the revised manuscript, we modified the text as follows: "Our updated estimates agree with previous results concerning the magnitude of the full-period average rates of sea-level rise in RSL (2.53 ± 0.14 mm/year) and VLM-corrected RSL (1.23 ± 0.13 mm/year)."

Moreover, for clarity, on line 389 we replaced 'relative MSL' with 'RSL'.

The last paragraph of section 4.1 was revised as follows:

"As illustrated in Sections 5 and 6, Mediterranean sea level, hence Venetian sea-level variations are tightly connected to sea-level variations in the midlatitude eastern North Atlantic, whose underlying processes differ from those in other oceanic basins. Therefore, any statistical consistency between historical Venetian RSL/VLM-corrected RSL and GMSL rise should not give the false impression that both variables are interchangeable and that any consistency in the historical period necessarily holds in the future. Still, it is instructive to compare estimates of Venetian RSL/VLM-corrected RSL and GMSL rise during the 20th Century to put local changes in the context of global mean changes. Venetian sea-level trends are smaller than GMSL trends reported in the fifth assessment report of the Intergovernmental Panel on Climate Change (IPCC-AR5), quantified as 1.7 [1.5 to 1.9] mm/year (likelihood >90%, period from 1901 to 2010, see: Church et al., 2013). They are, however, consistent with revisited estimates of historical GMSL rise that include significantly slower rates than reported by the IPCC-AR5 for the pre-altimetry period, e.g., 1.2±0.2 mm/year (90% confidence interval, Hay et al., 2015), 1.1 ± 0.3 mm/year (99% confidence interval, Dangendorf et al., 2017) and 1.56 ± 0.33 mm/year (90% confidence interval, Frederikse et al., 2020). Figure 6 revisits the connection between Venetian RSL/VLM-corrected RSL and GMSL trends on time scales ranging from interannual to centennial. Clearly, the significant difference between centennial trends in Venetian RSL and GMSL is strongly damped when the contribution of subsidence is removed, confirming the critical role of vertical land motions in determining local RSL variations. Nonetheless, the Venetian VLM-corrected RSL appears to rise at a lower rate than the GMSL over the second half of the 20th century (Fig. 6a). The GMSL-Venetian sea-level discrepancy observed in the first portion of the record is resolved when uncertainty in GMSL estimate is considered (not shown)."

*Paragraph 4.2:*

*'Multidecadal trends': what do the authors refer to when talking about 'multidecadal trends'? Just 'linear trends over 1993-present'?*

We changed the title to "Rates of sea-level rise during the satellite altimetry era".

*L445: Sorry, but I really disagree on this. One can fill whole bookshelves with papers looking into regional sea level from altimetry.*

In the revised manuscript the sentence is rephrased as: "An overall GMSL trend of about 3 mm/year during the satellite altimetry period is consistently reported by several studies (Hay et al., 2015; Chen et al., 2016; Dangendorf et al., 2017; Quartly et al., 2017). Regional trends can deviate considerably from the global mean (e.g., Scharroo et al., 2013; Legeais et al., 2018; Cazenave et al., 2019)."

*L464: Altimetry also observed the seasonal cycle in sea level. So, when comparing both, why do you need to correct for the seasonal cycle in tide-gauge observations?*

In fact the removal of the seasonal signal is just a common procedure. We rephrased the statement to clarify this in the revised manuscript as follows: "For consistency with post-processed altimetry data (Carrère and Lyard, 2003), the tide-gauge time series need to be preprocessed for removal of the local effect of atmospheric forcing (see Sect. 5.1.1)."

*Paragraph 4.3 In general, this section seems to lack focus. It's a mixture between variabilities of seasonal sea level, some remarks about peaks in the sea-level spectrum, a wavelet analysis, and reference to some correlation with sunspot cycles. I'm guessing here, but it might be the case that the authors have mixed up 'periodicity', oscillations that occur with a fixed frequency (such as the seasonal cycle or the M2 tide) versus 'low-frequency variability', variability that occurs on some typical time scales, but cannot be described as (a sum of) periodic functions, such as ENSO or the NAO. The former will cause a clear peak in the spectrum, while the latter is more associated with the behavior discussed in this section, such as intermittent signals in wavelet analyses, and correlations that vary over time. This low-frequency variability is common in global and local sea-level observations. Peaks will show up when computing a spectrum from tide-gauge data, but I wonder about the significance of any of these peaks. Beyond the conclusion that sea level in Venice shows decadal and multi-decadal variability, what should we distill from these peaks? How did the authors determine the significance of the peaks relative to a signal with a red spectrum? An alternative approach here could be to determine which processes act on with time scales.*

As correctly stated by the Reviewer, in this case the term "periodicity" was intended in the sense of a spectral peak at a certain frequency. To avoid confusion, in the revised version we replaced the term "periodicity" with "variability mode and significant spectral component". Please see the extensively revised section 5 for details.

We remark that, in the spirit of a literature review, we report the presence of significant spectral components detected by previous studies in the tide-gauge record of Venice. We provided the analysis of the updated seasonal tidal records for autumn and winter in to update published results and to confirm/revise their conclusions. Similarly, we illustrate possible linkages with large-scale atmospheric and oceanic phenomena as they are discussed in published papers. An unambiguous association of each variability mode with a well-defined process has not been done yet, and it is out of the scope of this review. In fact, we would better highlight in the revised manuscript that variations in the phase lag and intermittent significance in the wavelet coherence spectra indicate that caution should be taken when establishing connections between local sea-level variability and large-scale climatic modes, as they may not be robust over multidecadal or centennial time scales.

Concerning the determination of significance of the spectral peaks, the spectral peaks were tested against a red-noise (lag-1 autoregressive model) null-hypothesis. A lag-1 autoregressive model characterized by high power spectral density at lower frequencies. It is the most widely used model for geophysical purposes since a large class of geophysical processes produce output statistically compatible with red noise hypothesis (Allen and Smith 1996 and references therein). The theoretical power spectrum of a red-noise process is known and the parameters of the model are deduced from the analysed record. The power spectrum of the tide-gauge data was compared with that of the null-hypothesis, at a certain confidence level, and those peaks with power higher than that expected from the null-hypothesis have been defined as significant. Details are provided in chapter 10 of Alessio (2016).

Ref: Allen M, Smith L. 1996. Monte Carlo SSA: Detecting irregular oscillations in the presence of colored noise. Journal of Climate, 9(12): 3373–3404

Ref: Alessio SM. 2016. Digital signal processing and spectral analysis for scientists: concepts and applications. Springer.

*L490: The sunspot cycle. This seems a far-fetched link to me. How does the 11- year sunspot cycle cause significant local sea-level variations? I guess that this link is merely coincidence.*

In the spirit of a literature review, in the original manuscript we opted for including as much details as possible from the references quoted in the sentence. In the revised manuscript we omit the link to 11-year solar variations (which is anyway dealt with in detail in the companion paper Lionello et al., 2020b) and only refer to the presence of significant decadal variability in the record of Venetian storm surges. The revised sentence reads as follows: "Lionello (2005), Barriopedro et al. (2010), Troccoli et al. (2012) and Martínez-Asensio et al. (2016) consistently identify significant decadal variability in the time series of autumn Venetian surge events for the period 1948-2008 (for an updated analysis see Lionello et al., 2020b, in this special issue)."

*L494: What is a 'statistically significant fluctuation'? Significant with respect to what? White noise? A linear trend? Why is a wavelet analysis a good tool to study this?*

Fluctuation refers to the amplitude of spectral components. Significance of observed spectral peaks was tested against a red-noise hypothesis, namely a lag-1 autoregressive model characterized by high power spectral density at lower frequencies. It is the most widely used model for geophysical purposes since a large class of geophysical processes produce output statistically compatible with the red noise hypothesis (Allen and Smith 1996 and references therein). According to Grinsted et al. (2004) the significance is computed through the distribution of an ensemble of surrogate series describing a red noise process with the lag-1 parameter and variance estimated from the analysed time series.

Wavelet transform is commonly applied since it is an evolutionary spectral method which allows to examine the spectral evolution of the analysed record in a time-frequency(period) domain and to find localized intermittent periodicities (Grinsted et al., 2004). Thanks to the multivariate version of this method, namely the Wavelet Coherence, also the coherence between two time series, at certain frequency bands, can be detected as done in this study.

For the sake of clarity and completeness, we added the first paragraph of this response in the revised section 4.3 and modifed the caption of revised figure 9 as follows:

"[…] Shading (thick black contour) is the portion of the wavelet spectrum exceeding 90% (95%) confidence against red noise (lag-1 autoregressive model) hypothesis (see Grinsted et al., 2004, for details); […]"

Ref: Allen M, Smith L. 1996. Monte Carlo SSA: Detecting irregular oscillations in the presence of colored noise. Journal of Climate, 9(12): 3373–3404

*L500: This stationarity here has been attributed to Greenland and glaciers above, and here it is due to atmospheric processes. What is it?*

The text regarding the Scarascia and Lionello (2013) paper has been changed as follows in the revised manuscript:" Scarascia and Lionello (2013) conclude that the mean sea-level rise

of 1.3 mm/year in the Northern Adriatic in the period 1940-2005 is primarily due to ice cap melting, with no trends in atmospheric drivers, upper ocean temperature and surface layer salinity. The different estimates and attribution of multidecadal trends by Marcos and Tsimplis (2008) and Scarascia and Lionello (2013) reflect the strong interdecadal variability of Mediterranean sea level and the variety of its driving mechanisms. The former study computes the expansion of the water column missing the contribution of the redistribution of mass and produces negligible values of sea-level rise in shallow water areas, while the latter accounts for it and explicitly considers the Adriatic Sea."

*L505: What is "the integral of the absolute trend differences for bidecadal and shorter periods"?*

Thanks for pointing at this convoluted sentence. We referred to the sea-level difference, in absolute values, obtained by integrating (or summing up) through time the linear trend estimates for the GMSL and Venetian sea level over the given period of time, subtracting them for each other, and keeping the absolute value of such difference. It is meant to quantify the Venetian sea-level anomaly associated with the local linear trend with respect to the GMSL trend. We revised the sentence as follows: "Accordingly, the linear trend can yield a local sea-level anomaly in Venice from the GMSL of about 10 cm (but up to about 20 cm occasionally) over bidecadal and shorter periods, and a rather small anomaly (generally <5 cm) over interdecadal and longer periods (not shown)."

*Sections 5.1 and 5.2*

*General: like section 4.3, these sections lack focus, tend to introduce a lot of different processes, but at the end, I still don't understand the relative importance of each process. Furthermore, a general introduction of the physics behind the processes is necessary here to understand what's going on. For example, something like: "The North Atlantic Oscillation causes large-scale atmospheric pressure variations on interannual scales. The geostrophic winds caused by these variations drive a barotropic sea-level response in Venice, especially in winter."*

The revised manuscript includes, in a widely restructured Section 5, an improved description of physical mechanisms involved in sea-level variability. There is also a dedicated subsection for the NAO, which includes some some introductory statements along the one proposed by the Reviewer. Please see the extensively revised section 5 of the manuscript, our response to the more general comments by the Reviewer above, our response to the specific comments on the NAO below as well as our responses to the comments by Reviewer #1.

*L511: Steric effects. What about steric effects within the Mediterranean? Do they play an important role? They are discussed in the projections section, but what about observed changes? One could estimate the size of this effect over the last few decades from gridded hydrographic observations, such as EN4 (Good et al. 2013) or Ishii et al. (2017).*

Steric effects on Mediterranean sea-level change are better discussed based on available literature (for instance Jordà and Gomis, 2013; Carillo et al., 2012; Scarascia and Lionello, 2012), also taking advantage of the substantial restructuring of Sections 5 and 6 in the revised manuscript. Please see the extensively revised section 5 (particularly the introductory paragraph and section 5.1) for details.

Concerning the additional analyses suggested by the Reviewer, whereas we provide updated estimates for some components to sea-level variations, it is beyond the scope of this literature review paper to estimate the size of all individual effects to Mediterranean sea-level variability. We acknowledge that the availability of updated oceanic datasets as those reported by the Reviewer can be valuable for better constraining the different contributions to sea-level variability within the Mediterranean Sea, and we discuss this opportunity in revised Section 7 as follows: "Simulation-data comparison using updated hydrographic observations, such as EN4 (Good et al., 2013) or Ishii et al. (2017), could be extremely valuable."

*L520: What do the authors want to say in this paragraph?*

This part was removed in the revised manuscript.

*L531: Geostrophic wind is something different than large-scale wind (https://en.wikipedia.org/wiki/Geostrophic_wind)*

We certainly agree. The sentence was removed in the revised manuscript as it was confusing.

*L534: "Mass exchange can dominate: : :": Isn't NAO forcing just a local barotropic response to wind forcing and as such, just added on top of basin-wide fluctuations?*

In fact, the NAO has been associated with Mediterranean sea-level variability both as large-scale driver of winds over the Strait of Gibraltar, hence influencing water-mass exchange between Atlantic Ocean and Mediterranean Sea, and therefore basin-mean sea-level fluctuations in the Mediterranean, as well as driver of local wind anomalies within the Mediterranean, thereby contributing to spatial heterogeneity in sea-level variations within the basin. In the extensively restructured section 5 there is a subsection devoted to the NAO-Venetian and Mediterranean sea-level connection, also including a new figure (Figure 8) illustrating changes in the pressure and wind fields over the Mediterranean region associated with different NAO phases to support the text.

*L539: What is "explained linearly"? L540: They: : :EAWR. This is vague. What happens under the hood here?*

The sentence was removed from the revised manuscript.

*L547 This link is also discussed above, but here, the notion that is link is not very strong is omitted. I'd just leave the sunspot studies out.*

Agreed.

*L553. I don't see this strong correlation from the wavelet coherence plot. The correlation looks weak to me: it does not hold throughput time, goes in and out of phase. From this result, I'd make the opposite conclusions, namely that there's only a weak correlation between winter sea level and the NAO. In for example Piecuch et al. (2019), a wavelet*

*coherence plot that looks similar to this one (their Figure 1) is used to argue for a weak correlation.*

We overall agree that the NAO-Venetian sea level connection does not emerge as clearly for both seasons as just briefly discussed in the quoted lines of the original manuscript. In fact the lack of a robust connection with the NAO especially in autumn agrees with Zanchettin et al. (2009), where other climatic modes such as the SCA and the EAWR appeared to be predominant. In the revised manuscript we cite Piecuch et al. (2019) and Zanchettin et al. (2016) to put caveats on the interpretation of coherences with variable phase through time. We also discuss the results in the light of the new Figure 8 showing atmospheric circulation anomalies over the Euro-Mediterranean region around different NAO phases. Please see the extensively revised Section 5 for details.

Ref: Zanchettin, D., Bothe, O., Rubino, A., and Jungclaus,J. H.: Multi-model ensemble analysis of Pacific and Atlantic SST variability in unperturbed climate simulations. Clim. Dyn., 47(3),1073-1090, doi:10.1007/s00382-015-2889-2, 2016.

*L555 In autumn: : : how significant is this statement? Given the difference between interannual NAO variability and multidecadal subsidence variability, I doubt whether you're just looking for an explanation of insignificant changes.*

As explained in the response to the previous comment, we overall agree with the Reviewer. Our revised conclusion is that the NAO has a significant impact in winter whereas it exerts a less clear influence in autumn. In both autumn and winter, the patterns of sea-level pressure and wind anomalies under different NAO states superpose well on those that correspond to variations in Venetian sea level, including large-scale low pressures enhanced over the northern Mediterranean Sea and Sirocco-like northeastward wind anomalies over the Ionian Sea. However, amplitude and spatial extent of the significant anomalies in the atmospheric forcing fields are larger in winter compared to autumn, confirming the weak imprint of the NAO on Venetian sea level in the latter season. Please see new Figure 8 of the revised manuscript and associated discussion for details.

*L558. The contribution of the IBE effect to sea level has been quantified way before 2009 as 1cm of sea-level rise to 1 mbar of pressure drop (see Wunsch & Stammer, 1997). Using regression, you might find other correlation coefficients between sea-level pressure and sea level, but that's because sea-level pressure and sea level interact in many more ways than just the inverse barometer effect. See for example Woodworth et al. 2010. It may also be a good idea to repeat the conclusions from Calafat et al. (2012) that the IB effect only explains a marginal fraction of variability (see their Figure 4).*

In the substantially restructured Section 5, we briefly illustrate IBE with the following paragraph in subsection 5.2:

"Local atmospheric mechanical forcing is primarily exerted through local pressure anomalies, associated with the so-called Inverse Barometer Effect (IBE), and wind anomalies, the latter exerting a dominant effect on Mediterranean coastal sea levels especially by inducing a barotropic oceanic response (e.g., Calafat et al., 2012; Jordà et al., 2012a, 2012b). The IBE is quantified by the hydrostatic equation in about 1 cm of sea-level rise per 1 hPa of sea-level pressure drop. Calafat et al. (2012) quantify in 25% the IBE contribution to decadal winter sea-level variability in Trieste for the period 1950-2009. The highest IBE contributions to seasonal Venetian sea-level variability over the period 1872-2003 in autumn (about 32%) and winter (41.5%) estimated by Zanchettin et al. (2009) can be accounted for by the regression model between local sea-level and local sea-level pressure which could embed also other contributions than IBE alone (e.g., Woodworth et al., 2010)."

*L563-574: What is the exact point of this paragraph? Hard to follow.*

This paragraph was removed from the revised manuscript, with relevant parts of it embedded in different other paragraphs where appropriate.

*L574ff: Please carefully re-read Calafat et al. 2012: they discuss alongshore coastally-trapped wave propagation along the Atlantic coast, affecting Mediterranean Sea level as a whole. They do not look into the Adriatic Sea in particular.*

Indeed, thanks for catching this. The sentence was removed from the revised manuscript.

*L584-585: Accordingly -variability. This is a vague sentence. What is an atmospheric bridge in this context, and what does "constitute a potential precursor to multidecadal variability" mean?*

The sentence refers to the possibility that the remote connection between multidecadal variability in Venetian sea-levels and North Atlantic climate is determined by atmospheric teleconnections, following the analysis between Euro-Mediterranean sea level pressure and the Atlantic Multidecadal Oscillation by Mariotti and Dell'Aquila (2012). Accordingly, atmospheric circulation would act as a bridge.

We rephrase this in the revised manuscript within the restructured section 5, as follows: "In addition to the NAO, statistical connections identified in the literature between Venetian RSL and climatic modes include the atmospheric patterns known as Scandinavian and East Atlantic Western Russia (Zanchettin et al., 2009), showing prominent variability at interannual-to-decadal time scales in the autumn and winter seasons, and the Atlantic Multidecadal Oscillation or AMO (Scafetta (2014), describing multidecadal fluctuations in North Atlantic sea-surface temperature and influencing atmospheric variability over the Euro-Mediterranean region (e.g., Mariotti and Dell'Aquila, 2012; Maslova et al., 2017). "

*L586: "This could contribute to explaining the statistical connection between bidecadal variability of Venetian RSL". Or it could not. This statement needs some evidence, as it now looks like guesswork.*

We agree and our sentence was not meant to provide any conclusive evidence, rather point at a possibility. The sentence was removed from the revised manuscript.

*L588: What is "multi-scale acceleration analysis"?*

MSAA is the technique used by Scafetta (2014). We omit this detail in the revised manuscript as non relevant.

*L589ff: This section contrasts with many of the cited papers above, or at least, it needs more explanation. It's namely very unlikely that ice melt can explain 1.3 mm/yr in the Mediterranean, due to the magnitude of past ice melt and GRD effects, causing the Mediterranean to be affected much less than the global mean. How about large-scale changes in the Atlantic Ocean that propagate into the Mediterranean?*

The text was revised as follows: "Scarascia and Lionello (2013) conclude that the mean sea-level rise of 1.3 mm/year in the Northern Adriatic in the period 1940-2005 is primarily due to ice cap melting, with no trends in atmospheric drivers, upper ocean temperature and surface layer salinity. The different estimates and attribution of multidecadal trends by Marcos and Tsimplis (2008) and Scarascia and Lionello (2013) reflect the strong interdecadal variability of Mediterranean sea level and the variety of its driving mechanisms. The former study computes the expansion of the water column missing the contribution of the redistribution of mass and produces negligible values of sea-level rise in shallow water areas, while the latter accounts for it and explicitly considers the Adriatic Sea."

*L603: Lots of terms that need explanation: "amplitude modulation of water transport", "migration of the eastern hydraulic control".*

We would rephrase the first quoted text as "modulation of the water transport".

Then, there are two hydraulic controls in Gibraltar. One is located over the Camarinal Sill, the other is a moving hydraulic control that basically is locked in-phase with the bore propagation within the Tarifa Narrow. These details are not necessary, therefore we simplified the sentence as follows: "Local dynamics are strongly influenced by tides, which are responsible for the modulation of the water transport and the hydraulic control (Armi and Farmer, 1988), as well as for the substantial vertical mixing that has been observed (Garcìa-Lafuente et al., 2013)."

*Section 5.2 In its current setup, this section reads like it has been written without a clear focus in mind. What message do the authors want to tell with this section? It now reads like an unconnected collection of papers that each describe an individual problem, but an overarching story is missing. What models are available, reanalyses, operational forecasts? A good starting point may be the model results from Fukumori et al. (2007).*

Section 5 was substantially restructured and rewritten during the revision, with a merging of the sections regarding the physical mechanisms underlying Venetian sea-level variability and numerical modelling. Please see the revised manuscript for details.

*Section 6 Similar to the previous sections, this section is also somewhat unorganized and lacks focus. After reading it, I am unable to determine a conclusion from the section. Why don't the authors just use the SROCC projections? They should contain all the processes discussed here, except for the vertical land motion part. There is also a serious lack of information on the methods, and where methods are named, it feels like a bit of a grab-bag of individual estimates. I see SRES scenarios, AR5, SROCC, local projections... Are they combined in a consistent way? Where does the atmospheric forcing come from? Therefore, please thoroughly revise this section with a consistent treatment of scenarios and processes, together with a clear methods section.*

The feeling that this section - as others in this manuscript - is a collection of estimates from different authors, based on different data and methods, stems from the fact that this is meant to be a literature review. Our aim is to collect all relevant information and try to make sense of the differences in the conclusions reached in the various studies. We carefully revised section 6 concerning content, structure and presentation, as outlined by our responses to the specific comments below. In particular, we had a closer and clearer focus on projections for the local sea-level change in Venice, also taking advantage of the references suggested below by the Reviewer. Please see the revised manuscript for details.

We better detail how the different contributions to Venetian relative sea-level rise are are combined in our estimate. For the mentioned atmospheric forcing we rely on published estimates indicating that associated sea-level variations are generally less than but up to about 10 cm (as stated in line 677-679 of the original manuscript). As there is no quantitative precise estimate for this contribution for the case of Venice or the Northern Adriatic, we have revised the text highlighting that further research is required in this regard and avoid including this specific contribution to our total estimate. For instance, the following sentence is included in the revised section 7: "Still, certain aspects of the forcing of Venetian sea-level variability requires improved quantifications and characterization (for instance regarding the atmospheric forcing of local sea-level variability), and this will better constrain the range of projected future changes."

*L728: This statement is not true. There's no single process that causes a uniform sealevel rise. Therefore, each single process, from ice mass loss to GIA and sterodynamic effects causes a local deviation from GMSL. You might reach the conclusion that the resulting local changes are close to the global mean changes, but that's something different.*

We agree that, in its original form, the second part of this paragraph was confusing. The text has been revised as follows: "Regional effects can determine differences in the order of 10 cm between the mean Mediterranean sea level and the GMSL and within different parts of the Mediterranean basin itself as seen in the historical record. Changes over interdecadal periods can also distort the detection of forced trends over rather long periods of time (e.g., Jordà, 2014). However, scientific literature provides no evidence for a future deviation, on a centennial timescale, of the local sea level at the Venetian coastline from the GMSL that is larger than the abovementioned order of magnitude. RSL can differ by more as land movements and regional atmospheric patterns could provide additional and important contributions.".

*Section 7 L736: Similar as in the previous sections: except for VLM, how different are sea-level variations at the coast and just off the coasts as measured by altimetry? Or in other words? Why would we need altimetry closer to the coast?*

Essentially, availability of altimetry data very close to the coast would be important for a more direct comparison with tide-gauge data. This would allow us to better understand why both tools provide different - though not mutually inconsistent - statistics, such as trends (see Table 4). As a further scientific motivation, altimetry is the only tool currently capable of measuring sea level from the open ocean to the coasts, hence allowing to assess whether coastal sea level is rising at the same rate as open-ocean sea level. An example is sea level near the coast of western Africa, where observed trends are significantly different than offshore (Marti et al., 2019). We better clarify this in the revised manuscript as follows: "The extension of the satellite-based sea-level record toward the coast with measurement quality comparable to the open ocean allows detection of differences in the rate of coastal sea-level rise from the open ocean (e.g., Marti et al., 2019) and their implications for coastal hazards (Ablain et al., 2016; Benveniste et al., 2019)."

Ref: Marti, F., Cazenave, A., Birol, F., Passaro, M., Léger, F., Niño, F., ... & Legeais, J. F. (2019). Altimetry-based sea level trends along the coasts of western Africa. Advances in Space Research, doi: 10.1016/j.asr.2019.05.033.

*L795: The method to determine the trend and uncertainties critically depends on the purpose: what should the trend encompass? That should set the method you want to use.*

*For example, neither method gives you a number that should be extrapolated into the past or future.*

Here, we refer to the trend within its statistical definition. Following the International Statistical Institute (ISI, 2003), this would be the long-term movement in a time series, which may be regarded, together with the oscillation and random component, as generating the observed values. The issue, as stated in the manuscript, is meant to be exquisitely practical, as often in time series analysis the long-term trend is removed. We are convinced that some general guidance is needed in this regard, hence this part of our manuscript.

In the spirit of a literature review, we highlight the determination of the statistical trend as an open issue, and only provide two exemplary statistical models that could be used to calculate the trend. We do this for two time series - the raw RSL series and the VLM-corrected RSL series - to highlight differences and possible caveats due to the different considered processes.

We agree that neither trend should be extrapolated, and we see no reference to such extrapolation in our manuscript.

In the revised manuscript we better clarify the aim of this paragraph, first of all by determining what is meant here for "trend" as outlined above and by adding the following sentence: "The presence of substantial variations in multidecadal trends of sea level in Venice, together with methodological differences, explains the wide range of estimates of average sea-level rise obtained by different authors considering different periods."

Ref: ISI (The International Statistical Institute): The Oxford Dictionary of Statistical Terms. Yadolah Dodge (ed.), Oxford University Press, Oxford, 506 pp, ISBN 0-19-850994-4, 2003.

*L800: Similar to above: what does "the shape of the local RSL rise" encompass?*

This was rephrased as "the shape of the trend in sea level", hoping that the meaning is sufficiently clear in the light of the changes to be implemented in the preceding sentences, as outlined in the response to the previous comment.

*L803: What is "energetic variability"?*

Change to "significant variability"

*L809: This is not the first attempt to create regional sea-level projections for Venice. They are for example in the AR5 and SROCC report, Kopp et al. 2014 and Slangen et al. 2012. All provide local RSL projections with uncertainties. In this respect, Kopp et al. (2014) should be discussed here, since it uses a statistical model to estimate and project land motion not related to GIA.*

We acknowledge the fundamental contribution of the Slangen et al. (2012), AR5 and SROCC to study sea-level changes and the implications of sea-level rise for low-lying islands, coasts and communities. Despite there is no explicit reference to sea-level projections for Venice in these references (while Venice is indeed mentioned in the SROCC in a few instances regarding analysis of the associated tide-gauge data), we agree that they are relevant for this study as they provide regional and local relative sea-level, from which, somehow, one could "pick-up" sea-level projection for Venice. In this sense, Slangen et al.

(2012) is indeed a pioneering paper, and we cite this in the revised manuscript (revised section 6.2: "Pioneering work in this regard is Slangen et al. (2012).")

Also, thanks for pointing at Kopp et al. (2014), which is indeed very relevant for our paper as it contains explicit projections for Venice. They use the global tide-gauge PSMSL dataset and assume that the recorded sea level is represented as the sum of three Gaussian processes, including (1) a globally uniform process, (2) a regionally varying, temporally linear process, and (3) a regionally varying, temporally autocorrelated non-linear process for each tide-gauge site. The process (2) is retained as the "background non-climatic local sea-level change" corresponding to GIA, tectonics, and other non-climatic local effects. They then use this background linear estimates and its uncertainty for projections. In Venice, they estimate a background subsidence of 0.72 +/- 0.33 mm/yr with their technique (see their supplementary material, Table 8). The subsidence rates provided in Table 4 of our manuscript, which stems from various measurement methods, are overall higher than that by Kopp et al., which stems from a statistical method applied to tide-gauges record. Furthermore, Kopp et al. (2014) quote as a note of caution : "Third, our background rate estimates are the result of an algorithm applied to a global database of tide-gauge data, with different sites having been subjected to different degrees of quality control. Some tide-gauge sites may have experienced datum shifts or other local sources of errors not identified by the analysis. We recommend that users of projections for practical applications in specific regions scrutinize local tide-gauge records for such effects".

We extensively refer to Kopp et al. (2014) in the revised manuscript, in particular: "In particular, Kopp et al. (2014) provide probabilistic sea-level projections for Venice as part of a global set of local sea-level projections for three different representative concentration pathways. Their projections build on the decomposition of the recorded historical sea level into a number of processes, including the "background non-climatic local sea-level change" corresponding to GIA, tectonics, and other non-climatic local effects. This background linear estimate and its uncertainty (0.72±0.33 mm/year for Venice, see supplementary material, Table 8 in Kopp et al., 2014) is then included in the projections, together with the other components. The 5th-95th percentile range of the resulting sea-level change projections at year 2100 is 29-79 cm for RCP2.6 and 41-107 cm for RCP8.5."

We motivate our own projections as follows: "However, two critical aspects are needed to obtain reliable sea-level projections for Venice. First, a full characterization of the vertical land motions in Venice requires an explicit local focus beyond what is provided by a mere extrapolation on the site from regional patterns of GIA, tectonics, and other non-climatic contribution alone. In fact, the subsidence estimate by Kopp et al. (2014) lies in the lower range of the available instrumental estimates, especially as far as the most recent period is considered (see Table 4). Then, the sterodynamic component is derived from the outputs of the coupled climate-model simulations performed within the 5th phase of the Coupled Model Intercomparison Project. The rather coarse resolution of coupled climate models prevents an accurate representation of small-scale processes (e.g., water exchange at Gibraltar), which in turn affects regional sea-level estimates (Marcos and Tsimplis, 2008; Slangen et al., 2017). Both aspects are taken into consideration to build probabilistic projections of Northern Adriatic RSL for two climate scenarios (RCP2.6 and RCP8.5) and one high-end scenario following Thiéblemont et al. (2019)."

Our projections largely overlap with those by Kopp et al. (2014), suggesting that they are overall consistent despite some differences that reflect the fact that methods, models and assumptions differ between the two studies. We would mention the need to understand such differences as an opportunity for progress in the revised manuscript. In particular, the revised manuscript contains the following statement: "Our projections largely overlap with those provided by Kopp et al. (2014) for both scenarios, indicating that they are broadly consistent with each other. Differences are likely due to differences in the methods, models

and assumptions employed by both studies, which requires a dedicated investigation to be fully understood.".

Finally, the manuscript was revised by removing any reference to ours being the first attempt to develop local RSL change scenarios for Venice.

*Figures*

*There is no reference to Figure 3.*

Thanks for picking this. Figure 3 is relevant for section 2.2, and we refer to it as follows in the revised manuscript: "The results show that a reasonable increase in quantity and quality of data can be achieved compared to standard products up to a few kilometers from the coastline. Figure 3 illustrates the example of the Gulf of Trieste, where three missions cross the area and a data gap exists with standard products. In this case, the number of outliers along the Jason-1 and Jason-2 tracks is almost always less than the standard product and the improvement is clearly evident until 6 km from the coast."

*Figure 6: How have confidence intervals been determined? What noise model has been used?*

The trend and associated confidence intervals are obtained by linear regression analysis, using the matlab function "regress" for the calculation. The following clarification has been included in the caption of the figure: "black contours illustrate where the GMSL and Venetian sea-level trend estimates do not overlap within 95% confidence intervals obtained from the linear regression [...]"

*Figure 7 as well: how have all the errors been computed? For panel B? Why use model data from CMIP3, while we have had CMIP5 and now CMIP6 has become available as well?*

Concerning panel A, the errors are 90% confidence level obtained from linear least-squares regression analysis. This is clarified in the revised manuscript.

Concerning panel B, the plot is a replica of the original figure in Slangen et al. (2016). The uncertainties are the spread among the simulations with only differing Atlantic boundary conditions (blue) and the spread among the simulations with differing socio-economic scenarios (red). This is clarified in the revised manuscript.

Concerning panel C, the uncertainties correspond to the combined uncertainty of each sea-level component (i.e. glaciers, ice-sheets, sterodynamic, ...) calculated as the square root of the sum of the squares of each component uncertainty. Note however that contributions that correlate with global air temperature have correlated uncertainties and are therefore added linearly (this concerns sterodynamic and ice-sheet surface mass balance components). See Church et al. (2013) for more details. This is clarified in the revised manuscript.

We are confident that the improved caption and improved description in the text allows for an unambiguous reading of the figure. We are aware that CMIP6 data are being made available, but an updated analysis on such data deserves a dedicated study beyond the current literature review. This is highlighted in the revised manuscript with a sentence along the following lines at the end of section 7: "It will be also important to compare our estimates with updated scenarios of Venetian sea-level future change that are expected from the 6th phase of the Coupled Model Intercomparison Project (Eyring et al., 2016)."

Ref: Church, J.A.; Clark, P.U.; Cazenave, A.; Gregory, J.M.; Jevrejeva, S.; Levermann, A.; Merrifield, M.A.; Milne, G.A.; Nerem, R.S.; Nunn, P.D.; et al. Sea Level Change. In Climate Change 2013: The Physical Science Basis; Contribution of Working Group I to the Fifth Assessment Report of the Intergovernmental Panel on Climate Change ed.; Cambridge University Press: Cambridge, UK, 2013.

*Figure 8. Middle row: how should I interpret this plot?*

The panels represent the wavelet spectra for the autumn (left) and winter (right) Venetian sea-level time series, after correction for subsidence. Since the interest is on portions of the wavelet spectrum where wavelet amplitude exceeds statistical significance (against a lag-1 red noise), we only show this and omit the representation of wavelet amplitude. Specifically, "Shading (thick black contour) is the portion of the wavelet spectrum exceeding 90% (95%) confidence against red noise (lag-1 autoregressive model) hypothesis (see Grinsted et al., 2004, for details)". This is reported in the revised caption. The figure has been improved also regarding title and label for each panel.

*References to be added in revised manuscript*

*De Biasio, F., Baldin, G., and Vignudelli, S.: Revisiting Vertical Land Motion and Sea Level Trends in the Northeastern Adriatic Sea Using Satellite Altimetry and Tide Gauge Data. Journal of Marine Science and Engineering, 8(11), 949, 2020.*

*Calafat, F. M., Chambers, D. P., & Tsimplis, M. N. (2012). Mechanisms of decadal sea level variability in the eastern North Atlantic and the Mediterranean Sea. Journal of Geophysical Research: Oceans, 117(C9). https://doi.org/10.1029/2012JC008285*

*Dangendorf, S., Calafat, F. M., Arns, A., Wahl, T., Haigh, I. D., & Jensen, J. (2014). Mean sea level variability in the North Sea: Processes and implications. Journal of Geophysical Research: Oceans, 119(10), 6820–6841. https://doi.org/10.1002/2014JC009901*

*Dangendorf, S., Mudersbach, C., Wahl, T., & Jensen, J. (2013). Characteristics of intra-, inter-annual and decadal sea-level variability and the role of meteorological forcing: The long record of Cuxhaven. Ocean Dynamics, 63(2–3), 209–224. https://doi.org/10.1007/s10236-013-0598-0*

*Frederikse, T., & Gerkema, T. (2018). Multi-decadal variability in seasonal mean sea level along the North Sea coast. Ocean Science, 14(6), 1491–1501. https://doi.org/10.5194/os-14-1491-2018*

*Frederikse, T., Landerer, F., Caron, L., Adhikari, S., Parkes, D., Humphrey, V. W., Dangendorf, S., Hogarth, P., Zanna, L., Cheng, L., & Wu, Y.-H. (2020). The causes of sealevel rise since 1900. Nature, 584(7821), 393–397. https://doi.org/10.1038/s41586-020-2591-3*

*Frederikse, T., Riva, R., Kleinherenbrink, M., Wada, Y., van den Broeke, M., & Marzeion, B. (2016). Closing the sea level budget on a regional scale: Trends and variability on the Northwestern European continental shelf. Geophysical Research Letters, 43(20), 10,864-10,872. https://doi.org/10.1002/2016GL070750*

*Fukumori, I., Menemenlis, D., & Lee, T. (2007). A Near-Uniform Basin-Wide Sea Level Fluctuation of the Mediterranean Sea. Journal of Physical Oceanography, 37(2), 338–358. https://doi.org/10.1175/JPO3016.1*

Good, S. A., Martin, M. J., & Rayner, N. A. (2013). EN4: Quality controlled ocean temperature and salinity profiles and monthly objective analyses with uncertainty estimates. Journal of Geophysical Research: Oceans, 118(12), 6704–6716. https://doi.org/10.1002/2013JC009067

Gregory, J. M., Griffies, S. M., Hughes, C. W., Lowe, J. A., Church, J. A., Fukimori, I., Gomez, N., Kopp, R. E., Landerer, F., Cozannet, G. L., Ponte, R. M., Stammer, D., Tamisiea, M. E., & van de Wal, R. S. W. (2019). Concepts and Terminology for Sea Level: Mean, Variability and Change, Both Local and Global. Surveys in Geophysics. https://doi.org/10.1007/s10712-019-09525-z

Ishii, M., Fukuda, Y., Hirahara, S., Yasui, S., Suzuki, T., & Sato, K. (2017). Accuracy of Global Upper Ocean Heat Content Estimation Expected from Present Observational Data Sets. SOLA, 13(0), 163–167. https://doi.org/10.2151/sola.2017-030

Kopp, R. E., Horton, R. M., Little, C. M., Mitrovica, J. X., Oppenheimer, M., Rasmussen, D. J., Strauss, B. H., & Tebaldi, C. (2014). Probabilistic 21st and 22nd century sea level projections at a global network of tide-gauge sites. Earth's Future, 2(8), 383–406. https://doi.org/10.1002/2014EF000239

Landerer, F. W., & Volkov, D. L. (2013). The anatomy of recent large sea level fluctuations in the Mediterranean Sea. Geophysical Research Letters, 40(3), 553–557. https://doi.org/10.1002/grl.50140

Lickley, M. J., Hay, C. C., Tamisiea, M. E., & Mitrovica, J. X. (2018). Bias in Estimates of Global Mean Sea Level Change Inferred from Satellite Altimetry. Journal of Climate, 31(13), 5263–5271. https://doi.org/10.1175/JCLI-D-18-0024.1

Piecuch, C. G., Huybers, P., Hay, C. C., Kemp, A. C., Little, C. M., Mitrovica, J. X., Ponte, R. M., & Tingley, M. P. (2018). Origin of spatial variation in US East Coast sea-level trends during 1900–2017. Nature, 564(7736), 400–404. https://doi.org/10.1038/s41586-018-0787-6

Piecuch, C. G., Dangendorf, S., Gawarkiewicz, G. G., Little, C. M., Ponte, R. M., & Yang, J. (2019). How is New England coastal sea level related to the Atlantic meridional overturning circulation at 26 N? Geophysical Research Letters, 2019GL083073. https://doi.org/10.1029/2019GL083073

Slangen, A. B. A., Katsman, C. A., van de Wal, R. S. W., Vermeersen, L. L. A., & Riva, R. E. M. (2012). Towards regional projections of twenty-first century sea-level change based on IPCC SRES scenarios. Climate Dynamics, 38(5–6), 1191–1209. https://doi.org/10.1007/s00382-011-1057-6

Volkov, D. L., Baringer, M., Smeed, D., Johns, W., & Landerer, F. W. (2019). Teleconnection between the Atlantic Meridional Overturning Circulation and Sea Level in the Mediterranean Sea. Journal of Climate, 32(3), 935–955. https://doi.org/10.1175/JCLID-18-0474.1

Wahl, T., Haigh, I. D., Woodworth, P. L., Albrecht, F., Dillingh, D., Jensen, J., Nicholls, R. J., Weisse, R., & Wöppelmann, G. (2013). Observed mean sea level changes around the North Sea coastline from 1800 to present. Earth-Science Reviews, 124, 51–67. https://doi.org/10.1016/j.earscirev.2013.05.003

Woodworth, P. L., Pouvreau, N., & Wöppelmann, G. (2010). The gyre-scale circulation of the North Atlantic and sea level at Brest. Ocean Science, 6(1), 185–190. https://doi.org/10.5194/os-6-185-2010

*Wunsch, C., & Stammer, D. (1997). Atmospheric loading and the oceanic "inverted barometer" effect. Reviews of Geophysics, 35(1), 79–107. https://doi.org/10.1029/96RG03037*

*Zanna, L., Khatiwala, S., Gregory, J. M., Ison, J., & Heimbach, P. (2019). Global reconstruction of historical ocean heat storage and transport. Proceedings of the National Academy of Sciences, 201808838. https://doi.org/10.1073/pnas.1808838115*

We added all relevant references in the revised manuscript.

---

## Author Response (AR2)

**Response to Reviewer #1**

We thank the Reviewer for the appreciation of our work.

We have checked the typos evidenced by the Reviewer. Note that Otranto was correct as the statement indeed refers to the Adriatic basin, of which the Otranto Strait is the southern boundary separating it from the Ionian Sea.

**Response to Reviewer #2**

We thank the Reviewer for the appreciation of our work and for the insightful comments she/he has provided about our revised manuscript. Below we provide our detailed response to the specific comments by the Reviewer (in bold and italic fonts).

*Review of "Sea-level rise in Venice: historic and future trends"*

*This is the second round of review of the paper, and the authors have substantially improved the manuscript, and I only have some minor comments left. The exception to this is the section on sea-level projections (section 6.2), which still contains a substantial number of confusing statements with regards to how sea-level projections are made.*

*When section 6.2 is properly revised and the other minor issues have been taken care of, I think the paper can be published.*

REPLY: We are confident that the newly revised version of the manuscript satisfies all the requests by the Reviewer.

*Remarks on section 6.2*

*L710: The SROCC report also provides regional sea-level projections based on CMIP5 model output. Now the text suggests that some sort of down-scaling is needed from these projections, while they are already available in gridded format. These projections form the basis of Kopp et al. (2014) and follow the methodology outlined in Slangen et al. (2012).*

*The big question, which is touched upon, but not answered in this section, is whether the CMIP5 models and other projections capture the relevant processes that cause sea-level changes in Venice. Can anything be said about this question?*

REPLY: We provide the following arguments to support the need for additional evaluation of sea-level rise scenarios for Venice.

First, an important caveat with the CMIP5 model results and the data provided in the SROCC and AR5 is their reliability for coastal assessments. For instance, the SROCC datasets contain jumps of up to 5 cm between estimates of sea-level rise at year 2100 under the RCP8.5 scenario between neighboring coastal pixels in the bay of Biscay. These jumps are not physical, rather reflect the fact that coastal pixels are not covered by all models. This issue applies also to many coastal pixels in the Mediterranean Sea.

Second, many studies revealed the possibility that General Circulation Models do not represent realistically the water exchange through the Strait of Gibraltar (see also our response to a comment below). A very recent dynamical downscaling work (EGU 2021 presentation from Chaigneau et al.) revealed very large sea-level discrepancies between a regional model and GCM simulations in the Meditearrenean Sea.

On the premise that CMIP5 projections for the Mediterranean Sea may not be reliable, our method (following Meyssignac et al., 2017 ) allows recognizing the CMIP5 issue in the Mediterranean basin and inflates accordingly the uncertainty in projections of the sterodynamics component. In other words, this increase in uncertainty is consistent with the fact that there is a low confidence in the CMIP5 projections from the 12 GCMs that cover the Mediterranean Sea (see our response below regarding the Reviewer's comment about L779). Apart from that, improvements of sea-level projections for the Mediterranean requires regional high-resolution modeling to resolve the relevant dynamical processes.

This paragraph in the revised version of the manuscript reads as follows:

"Then, the sterodynamic component is derived from the outputs of the coupled climate-model simulations performed within the 5th phase of the Coupled Model Intercomparison Project (CMIP5). The rather coarse resolution of coupled climate models prevents an accurate representation of small-scale processes (e.g., water exchange at Gibraltar), which in turn affects regional sea-level estimates (Marcos and Tsimplis, 2008; Slangen et al., 2017). Another important caveat on multi-model assessments is their reliability on coastal regions where the contributing models may differently resolve the coastline and bathymetry peculiarities, thus yielding local anomalies in the gridded multi-model output that may reflect a bias originated by heterogeneous spatial resolutions across models rather than a physical process (e.g., Landerer et al., 2014).

On this premise, we propose probabilistic projections of Northern Adriatic RSL for two climate scenarios (RCP2.6 and RCP8.5) and one high-end scenario following Meyssignac et al. (2017) and Thiéblemont et al. (2019). The method allows to inflate the uncertainty in projections of the stereodynamics component by accounting for the low confidence in projections of coastal sea-level rise obtained from the limited number of global circulation models participating in CMIP5 and covering the Mediterranean Sea (see Figure 2 in Thièblemont et al., 2019). Specifically, the Mediterranean sterodynamic sea-level projections are estimated by relying on those of the Atlantic area near Gibraltar. […]"

References

Meyssignac, B., Slangen, A. B. A., Melet, A., Church, J. A., Fettweis, X., Marzeion, B., Agosta, C., Ligtenberg, S. R. M., Spada, G., Richter, K., Palmer, M. D., Roberts, C. D. and Champollion,N.: Evaluating Model Simulations of Twentieth-Century Sea-Level Rise. Part II: Regional Sea-Level Changes. J. Climate, 30, 8565-8593, 2017

***L712: "Here "likely" corresponds to the IPCC uncertainty language, meaning that the probability of future sea-level change within this range is estimated from ≥66% to 100%, and therefore does not exclude values outside this range" What does this mean?***

REPLY: We have removed the last confusing part of the sentence

***L716: "Deep uncertainty" is a very specific term with a specific definition. Suggestion to remove it here.***

REPLY: The reference to Bakker et al. (2017) in the sentence with the quoted text specifies "deep uncertainty" in the context of West Antarctic Ice Sheet contribution. More generally, we quote a paragraph

from a recent paper of Haasnoot et al. (2020): "Despite the growth of scientific studies about Antarctica, its contribution to future rate of SLR is still highly uncertain and undergoing a strong scientific debate [Kopp et al., 2017]. In fact, the uncertainty in projected SLR increased recently [Garner et al., 2018; Bamber et al., 2019]. In decision making literature this is referred to as 'deep uncertainty' [Lempert, 2019], which occurs when experts do not have sufficient knowledge or when parties to a decision cannot agree upon the system processes and futures."

We have revised the text by adding the following clarifying sentence: "The high uncertainty and strong scientific debate on the contribution of Antarctic ice-sheet melting to the future rate of sea-level rise generates the so-called 'deep uncertainty' [Lempert, 2019], i.e., a condition where experts lack sufficient knowledge or parties to a decision cannot agree upon the system processes and futures (see also Haasnoot et al., 2020)."

References

Haasnoot, M., Kwadijk, J., van Alphen, J., Le Bars, D., van den Hurk, B., Diermanse, F, van der Spek, A., Oude Essink, G., Delsman, J. and Mens, M.: Adaptation to uncertain sea-level rise; how uncertainty in Antarctic mass-loss impacts the coastal adaptation strategy of the Netherlands. Environ. Res. Lett. 15, 034007, 2020

Lempert, R. J.: Robust Decision Making (RDM) BT—Decision Making under Deep Uncertainty: From Theory to Practice, Editors: Marchau, V. A. W. J., Walker, W.E., Bloemen, P.J.T.M., Popper, S.W. (Berlin: Springer), 23–51, 2019.

*L717: "Therefore": how does this sentence follow from the previous? And where does the "up to 2m" come from?*

REPLY: We agree, we have removed the "Therefore" and the parenthesis where the 2 m level was mentioned.

*L720: "Slangen et al. (2017) suggest": That paper doesn't "suggest" that number, it shows model results, and this number comes from model results and is not a 'suggestion'.*

REPLY: We agree, we have changed the sentence as "According to Slangen et al. (2017), the sea-level rise at the subtropical […]"

*L722: The coupled climate models used for CMIP3/AR4, CMIP5/AR5 do simulate the Mediterranean Sea directly. See for example Landerer et al. (2007, doi: 10.1175/JPO3013.1). Sterodynamic effects are computed directly by these models (the 'zos' and 'zostoga' variables in CMIP5/6 models) and do not need to be computed offline. The same holds for the associated water mass redistribution: this effect is also included in these models and is stored as variable 'pbo'.*

REPLY: We have removed the sentences "The steric effects are computed from temperature and salinity changes using a diagnostic offline computation. This computation obviously depends on the water depth and tends to zero at the coastline. Therefore, […]"

*L723: "Pioneering work in this regard is Slangen et al. (2012)" the pioneering work in this paper is not about the ocean models and statistical downscaling: it's about combining ocean sterodynamics from*

*CMIP models with GRD effects and GIA to make regional sea-level projections that include all relevant processes.*

REPLY: We agree, the position of the sentence is misleading. We have removed the sentence as Slangen et al. (2012) is correctly quoted in line 762.

*L742: As noted in the previous round: there's no reason to assume that GMSL is equal to sea level in Venice, so there's no 'consistency' when both numbers are close. Suggest to replace 'consistent with' by 'on the same order as' or 'similar to'.*

REPLY: We agree as we also explicitly state in the paragraph starting at line 440. We have changed "consistent with" with "similar to". We have also checked the usage of "consistency" throughout the text and changed where deemed necessary with "similarity" or analogs.

*L747: "which is ignored in the computation of the pure steric effect": This might be a bit of a strawman argument: to my knowledge, no projections just use the steric effect to approximate the total sterodynamic (steric + bottom pressure) effects. See also my comments for L722*

REPLY: We have removed the quoted text from the sentence.

*L764: "Their projections build on the decomposition of the recorded historical sea level into several processes, including the "background non-climatic local sea-level change" corresponding to GIA, tectonics, and other non-climatic local effects." This is not an accurate description of the Kopp et al. (2014) framework. They use the AR5 projections for most terms (or switch to an alternative projection for the ice sheets in the Kopp et al. (2017) update) and use a statistical framework to estimate the non-climatic component at each tide-gauge location.*

REPLY: We have updated the description as follows: "Their projections build on a combination of expert community assessment (the IPCC-AR5), expert elicitation (e.g., Bamber and Aspinall, 2013), and process modelling (e.g., the 5th phase of the Coupled Model Intercomparison Project or CMIP5) for most sea-level contributors. The "background non-climatic local sea-level change" corresponding to GIA, tectonics, and other non-climatic local effects was derived by applying a Gaussian process model to tide gauge records. This background linear estimate [...]"

Reference

Bamber,J. L., and Aspinall, W. P.: An expert judgement assessment of future sea level rise from the ice sheets. Nature Climate Change 3, 424–427, 2013.

*L779: "The Mediterranean sterodynamic sea-level projections are estimated by relying on those of the Atlantic area near Gibraltar." This is an interesting remark, as it's a much 'coarser' approximation than the coarse CMIP models. Is it a better approach and do the CMIP models suggest something different? From Figure 7 in Slangen et al. (2017) that doesn't seem to be the case.*

We thank the Reviewer for this comment that indeed deserves slightly more explanations. Figure R1 below (from Thiéblemont et al., 2019) shows projections of the sterodynamic component from CMIP5 models from RCP8.5 by the end of the 21st Century for several basins in Europe. The figure shows that semi-enclosed seas are not fully covered by all models—among the 21 models, only 14 and 12 cover the Baltic and Mediterranean basins, respectively. Furthermore, the central estimates and the model spread in the Mediterranean sea are

found to be lowered compared to surrounding basins. These differences between model spatial coverage result in inconsistencies when computing multi-model ensemble statistics, which in turn could significantly affect the spatial homogeneity of regional sea-level rise projections.

Beside this multi-model sampling issue, Landerer et al. (2014) detected unrealistic SSH biases in marginal seas for some CMIP models (e.g. -15 m over the Mediterranean for MIROC-ESM historical simulations). They could not identify a reason for such biases but suspect that the model resolution could play a role. Parras-Berocal et al. (2020) found that MPI-ESM-LR is not able to represent the exchange through Gibraltar. Meyssignac et al. (2017), who analyzed CMIP5 historical simulations, argued that the coarse resolution of climate models does not enable the simulation of the mesoscale processes and the water exchanges at Gibraltar, which results in a poor representation of the Mediterranean sea level in GCMs. As a consequence, they excluded the Mediterranean basin from the sea level simulations and instead use the sea level in the Atlantic, off the Strait of Gibraltar, as an approximation for the Mediterranean sea level.

Based on these different elements and studies, we followed the procedure of Meyssignac et al. (2017) in the present work rather than relying on the 12 CMIP5 models that provide sterodynamic estimates in the Med. basin.

[Figure]

**Figure R1.** CMIP5 sterodynamic projections in 2099 (ref period 1986–2005) for the North-Atlantic-N, North-Atlantic-S, Bay of Biscay, North Sea, Baltic Sea, Mediterranean-E, and Mediterranean-W Sea under the RCP8.5 scenario. Whisker boxes display the multi-model 1st quartile, median, and 3rd quartile and the dashed line shows the multi-model mean. After Thiéblemont et al. (2019)

References

Landerer, F.W., Gleckler, P. J., Lee T.: Evaluation of CMIP5 dynamic sea surface height multi-model simulations against satellite observations. Climate Dynamics 43, 1271–1283, 2014.

Parras-Berrocal, I.M., et al.: The climate change signal in the Mediterranean Sea in a regionally coupled atmosphere–ocean model. Ocean Sci., 16, 743–765, 2020

*L787: I think I know what is meant by "added linearly" but it might be clearer to write out the equation for the combination of the uncertainties.*

REPLY: The adopted equation is as follows:

$$\sigma_{tot}^2 = \left(\sigma_{sterodynamic} + \sigma_{smb-a} + \sigma_{smb-g}\right)^2 + \sigma_{Glac}^2 + \sigma_{LW}^2 + \sigma_{dyn-a}^2 + \sigma_{dyn-g}^2$$

where smb stands for Surface Mass Balance, Glac for Glaciers, LW for Landwater and dyn-a/dyn-g for Dynamic Antarctic and Greenland. It is basically the same equation as in the supplementary material of Chapter 13 of IPCC AR5, as mentioned in the manuscript. We have added this in the revised manuscript.

***L788: "The projections do not include expert elicitation and rely only on IPCC-like assessments, so the RCP2.6 is rather symmetric and RCP8.5 slightly asymmetric." The IPCC process is pretty close to an expert elicitation, and the link to the symmetry/asymmetry of the uncertainties is not related to that per se.***

REPLY: We agree that the sentence may be confusing and we have removed it from the revised manuscript.

***Other remarks***

***The definitions on L69ff:***

***"The acronym RSL is therefore used for tide-gauge data": I'd say: 'Tide gauges typically measure RSL' instead. Same for altimetry: they measure GSL.***

REPLY: We have changed "The acronym RSL is therefore used for tide-gauge data" with "Tide gauges typically measure RSL" and "The acronym GSL is therefore used for satellite altimetry sea-level data;" with "Satellite altimetry provides GSL";

***"VLM-corrected RSL" That is the same thing as GSL: GSL = RSL + VLM. Like above: tide gauges corrected for VLM measure GSL.***

REPLY: We prefer to keep our notation to make it clear that those data are measured by a tide gauge and not by an altimeter.

***GMSL is spatially-averaged RSL.***

REPLY: Agreed, we have updated the statement and have checked the text for the correct use of GMSL in the revised sense.

***L443: 'consistency'. I think the authors mean 'similarity' here. GMSL and sea level in Venice may have a similar trend, but that has nothing to do with 'consistency'.***

REPLY: Agreed, we changed to "similarity".

***L445: "to put local changes in the context of global mean changes". Vague and a circular reasoning. It now reads like it instructive to compare both, in order to see if they're comparable.***

REPLY: We change "instructive" with "relevant". As we have reported in our response to the Reviewer's public comment, this "analysis was indeed motivated by the fact that available sea-level rise projections for Venice are in some cases directly based upon estimates of the GMSL rise (see, for instance, Troccoli et al., 2012, and Carbognin et al., 2010)." We have inserted this statement in the revised manuscript.

*L451: 'connection' as said above, there' no connection. Only similarity. Check also the sentences after L450 for similar suggestions of connections/discrepancies etc.*

REPLY: Connection in this case was meant as a statistical relation between the data, not a physical connection between processes. We have rephrased by removing "the connection between".

*L570-L575: "The two-way water exchange regime…" This and the following sentence are vague and I don't understand what's being said here. What message should I get from these sentences?*

REPLY: We have rephrased as "Watermass exchanges across the Strait…"

*L582: "strengthened by steric changes since the late 1950s": what does that mean? From Frederikse et al. (2020): "Before the end of the 1950s, in situ observations are too sparse to derive unbiased steric changes". How do we know they did not play a role before the 1950s?*

REPLY: The sentence was indeed shrunk too much. We have rephrased as follows: "There, GIA was predominant over the ocean-mass contribution to determine the upward sea-level trend over the 20th Century; since the late 1950s unbiased estimates of steric changes are also available, indicating a contribution to the sea-level rise comparable to GIA (Frederikse et al., 2020)."

*L885: "would be reliable only in the basin mean tendencies": where does this conclusion come from? Same from the next sentences. A citation or an experiment to prove these statements is needed.*

REPLY: We have changed the sentence as follows: Overall, even under accurate representation of global steric and mass addition from the Atlantic, projections of Mediterranean sea-level change from current regional ocean models would hardly provide reliable local sea-level tendencies for Venice and the Northern Adriatic."

*L888: "Improved assessment and progress is hoped in this direction as well". Hope is the mother of disappointment. Suggest to remove this.*

REPLY: Agreed, sentence removed.

*L889: Good et al. (2013) and Ishii et al. 2017 are not about new observations, but they're about optimal interpolations of in-situ temperature and salinity profiles.*

REPLY: We have changed "observations" with "datasets".

*L896ff: Circling back to the first round of review. What would be the outcome of 'further research' on estimating a trend in a sea level record? I'd say that 'The shape of the local sea level trend' is just the first derivative of the time series of local sea level. You can decompose that time series into contributions from various processes (wind, subsidence, ice melt etc.), but I can't see what approaches like SSH or EMD can*

***add. While these methods often create more confusion than that they solve, see for example***
***https://npg.copernicus.org/articles/22/157/2015/.***

REPLY: We agree. Still, as we state at the beginning of the paragraph, this is aimed at solving a practical question related to data preprocessing performed in many studies where changes in the mean need to be removed before the main analysis. We have rephrased the sentence in lines 896-899 as follows: "As far as the higher rates of RSL rise observed in recent decades are concerned, the simple acceleration expressed statistically in terms of quadratic fitting seems therefore to be insufficient and further methods could be explored (but as a note of caution see, e.g., Chambers, 2015)."

References

Chambers, D.P. (2015). Evaluation of empirical mode decomposition for quantifying multi-decadal variations and acceleration in sea level records. Nonlin. Processes Geophys., 22, 157–166, 2015, https://doi.org/10.5194/npg-22-157-2015